# Engineered tRNAs suppress nonsense mutations in cells and in vivo

Suki Albers[1], Elizabeth C. Allen[2], Nikhil Bharti[1], Marcos Davyt[1], Disha Joshi[3,4], Carlos G. Perez-Garcia[2], Leonardo Santos[1], Rajesh Mukthavaram[2], Miguel Angel Delgado-Toscano[1], Brandon Molina[2], Kristen Kuakini[2], Maher Alayyoubi[2], Kyoung-Joo Jenny Park[2], Grishma Acharya[2], Jose A. Gonzalez[2], Amit Sagi[2], Susan E. Birket[5], Guillermo J. Tearney[6,7], Steven M. Rowe[5], Candela Manfredi[3,4], Jeong S. Hong[3,4], Kiyoshi Tachikawa[2], Priya Karmali[2], Daiki Matsuda[2], Eric J. Sorscher[3,4✉], Pad Chivukula[2✉] & Zoya Ignatova[1✉]

Nonsense mutations are the underlying cause of approximately 11% of all inherited genetic diseases[1]. Nonsense mutations convert a sense codon that is decoded by tRNA into a premature termination codon (PTC), resulting in an abrupt termination of translation. One strategy to suppress nonsense mutations is to use natural tRNAs with altered anticodons to base-pair to the newly emerged PTC and promote translation[2–7]. However, tRNA-based gene therapy has not yielded an optimal combination of clinical efficacy and safety and there is presently no treatment for individuals with nonsense mutations. Here we introduce a strategy based on altering native tRNAs into efficient suppressor tRNAs (sup-tRNAs) by individually fine-tuning their sequence to the physico-chemical properties of the amino acid that they carry. Intravenous and intratracheal lipid nanoparticle (LNP) administration of sup-tRNA in mice restored the production of functional proteins with nonsense mutations. LNP–sup-tRNA formulations caused no discernible readthrough at endogenous native stop codons, as determined by ribosome profiling. At clinically important PTCs in the cystic fibrosis transmembrane conductance regulator gene (CFTR), the sup-tRNAs re-established expression and function in cell systems and patient-derived nasal epithelia and restored airway volume homeostasis. These results provide a framework for the development of tRNA-based therapies with a high molecular safety profile and high efficacy in targeted PTC suppression.

Efforts to develop treatments for patients with nonsense mutations focus on using low molecular weight pharmacological compounds or sup-tRNAs that induce readthrough at PTCs and restore translation and production of full-length proteins[2–9]. Although some pharmacological approaches have been used in clinical trials, non-specific insertion of random amino acids[10], off-target effects at natural stop codons and safety in long-term applications have limited their clinical use. Pioneered four decades ago[3,6], natural sense-codon-decoding tRNAs with an altered anticodon to decode stop codons have been shown to correct PTCs, but tRNA-based therapies based on this principle have not reached clinical trials because of insufficient efficacy, inability to achieve a therapeutic threshold and insufficient safety. One impediment is that not every native tRNA can be engineered into a sup-tRNA by altering its anticodon[11], in part because the decoding at a sense codon markedly differs from the hydrolysis at a stop codon[12], which is mediated by release factor[13] (eRF1) in eukaryotes. Moreover, newly emerged PTCs activate mRNA surveillance pathways (including nonsense-mediated mRNA decay (NMD)) to degrade mutated mRNA[14–16]. An anticodon-altered sup-tRNA may not establish the ideal geometry for decoding[11] to efficiently outcompete premature translation termination and the mRNA degradation process. Harnessing functionally conserved features of natural tRNAs and modulating sequences outside the anticodon, we successfully repurposed bacterial tRNAs to incorporate exclusively alanine at the UGA stop codon with codon–anticodon interactions resembling the Watson–Crick geometry of a sense-codon-decoding tRNA[17]. We reasoned that by applying a similar strategy and modulating various sequence segments of human tRNAs that are crucial for tRNA function in translation (Fig. 1a) such as the anticodon-stem and anticodon-loop—which modulate the accuracy of decoding—and the TΨC-stem that determines binding affinity to elongation factor[18–20] (eEF1A in humans), we would enhance sup-tRNA efficacy. We also leveraged a synthetic LNP system[21] to encapsulate sup-tRNA and produce safe LNP–sup-tRNA with high efficacy for PTC suppression in vivo with a robust molecular safety profile.

[1]Institute of Biochemistry and Molecular Biology, University of Hamburg, Hamburg, Germany. [2]Arcturus Therapeutics, San Diego, CA, USA. [3]Department of Pediatrics, School of Medicine, Emory University, Atlanta, GA, USA. [4]Children's Healthcare of Atlanta, Atlanta, GA, USA. [5]Pulmonary, Allergy, and Critical Care Medicine, University of Alabama at Birmingham, Birmingham, AL, USA. [6]Wellman Center for Photomedicine, Massachusetts General Hospital, Boston, MA, USA. [7]Harvard-MIT Health Sciences and Technology, MA, Cambridge, USA. ✉e-mail: esorscher@emory.edu; pad@arcturusrx.com; zoya.ignatova@uni-hamburg.de

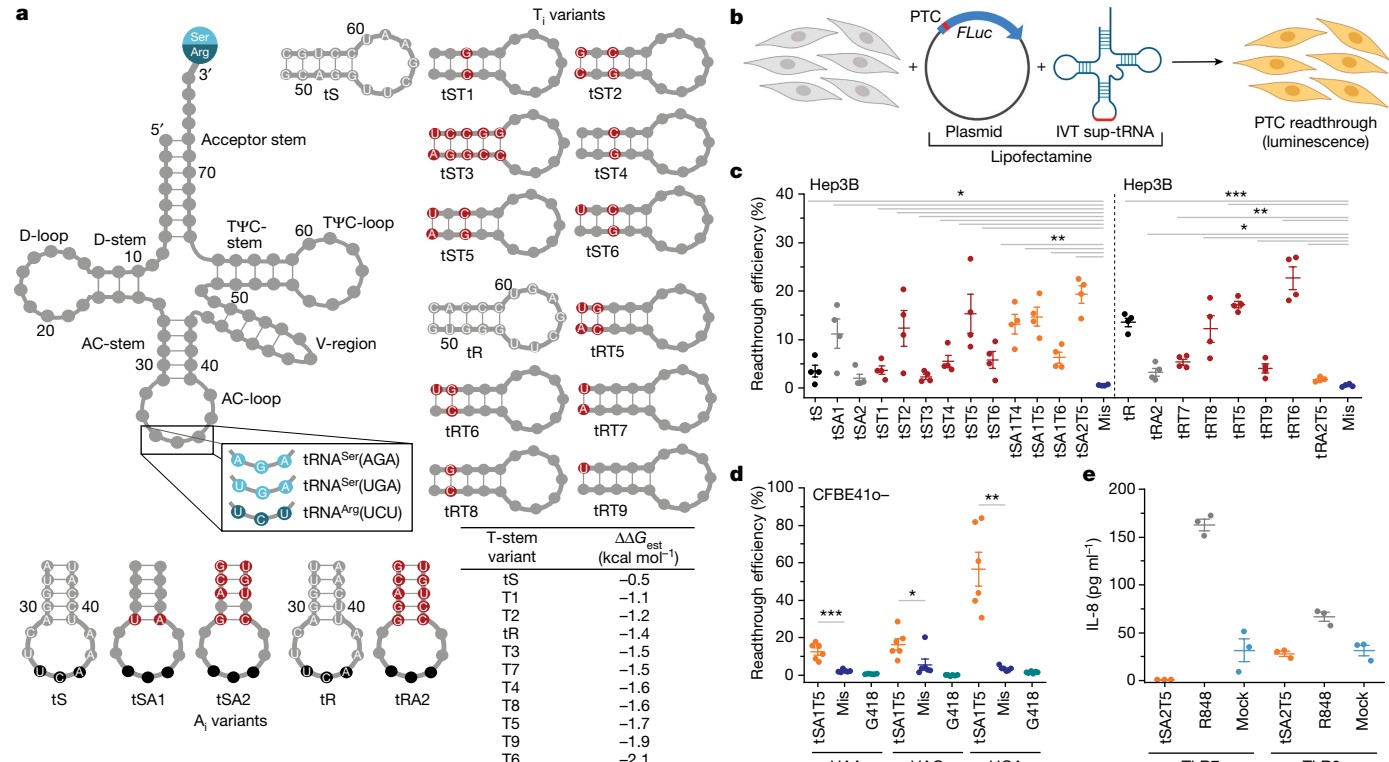

**Fig. 1 | Sup-tRNA variants suppress different PTCs at Ser and Arg codons. a**, Schematic of a generic tRNA with the natural anticodon of human tRNA^Ser^UGA, tRNA^Ser^AGA or tRNA^Arg^UCU (left). Nucleotide substitutions in the anticodon (AC)-stem (A$_i$ variants; bottom) or the TΨC-stem (T$_i$ variants; right) of tS or tR variants are highlighted in red. The table shows estimated $\Delta\Delta G$ values for binding affinities of eEF1A to the TΨC-stem. **b**, Schematic of the screening of sup-tRNAs with plasmid constructs encoding firefly luciferase (*FLuc*) with a PTC mutation (PTC-*FLuc*). IVT, in vitro-transcribed tRNA. **c**, Suppression efficacy of tS or tR variants in human liver Hep3B cells at FLuc(R208X) (in which X represents UGA), normalized to wild-type FLuc. Mis, mismatch tRNA.

Data are mean ± s.e.m. (*n* = 4 independent replicates). **d**, tSA1T5 targets PTCs with different stop codon identities, as tested with Fluc(S466X) in human bronchial epithelial CFBE41o⁻ cells and normalized to wild-type FLuc. G418 (geneticin) is a low molecular mass readthrough-promoting agent. Data are mean ± s.e.m. (*n* = 6 independent replicates). **e**, TLR-dependent activation by tSA2T5 was monitored in human TLR-transformed HEK293 cells. R848 agonist, which activates both TLR7 and TLR8, served as a positive control. Mock, mock-transfected cells. Data are mean ± s.e.m. (*n* = 3 independent replicates). *$P$ < 0.05, **$P$ < 0.01, ***$P$ < 0.001. One-sided *t*-test.

## Engineering sup-tRNAs for high efficacy

We selected three human tRNA^Ser^, tRNA^Arg^ and tRNA^Gly^ families that decode codons that are frequently mutated to PTCs[1]. We first exchanged their anticodons to pair to the UGA PTC, resulting in tS, tR and tG, respectively (Fig. 1a, Extended Data Fig. 1 and Supplementary Table 1). For the initial screen, the in vitro-transcribed sup-tRNAs with functionally homogenous 3′ ends (Extended Data Fig. 2) were co-transfected in human Hep3B cells with a plasmid-encoded PTC reporter of firefly luciferase (*FLuc*) (Fig. 1b); the coding sequence of the *FLuc* reporter was extended at its 5′ end by 15 codons representing the sequence context of the most common PTC (*FLuc^R208X^*; Supplementary Table 2) in the tripeptidyl peptidase 1 gene associated with lysosomal storage disorder. As expected, tS, tR and tG exhibited low readthrough activity, with tR showing the highest efficiency (Fig. 1c and Extended Data Fig. 1), probably because the natural tRNA^Arg^UCU—the precursor of tR—is intrinsically prone to miscoding[22]. To achieve similar decoding efficiency among all natural tRNAs, their sequences have been fine-tuned by evolution to the chemical nature of the cognate amino acid, whereby the destabilizing thermodynamic effect of some amino acids is compensated by stronger interactions of the elongation factor with the TΨC-stem and vice versa[23]. The three selected tRNA families were aminoacylated with serine, arginine or glycine, which span the entire spectrum of thermodynamic contributions (that is, glycine is a destabilizing amino acid, serine is stabilizing and arginine is nearly neutral[23]). Next, we subjected tS, tR and tG to a comprehensive

set of sequence changes, thereby preserving the recognition signals for the cognate aminoacyl-tRNA-synthetase (Fig 1a, Extended Data Fig. 1a and Supplementary Table 1). Changes in the TΨC-stem were made to stabilize (lower free energy difference ($\Delta\Delta G$) values) or destabilize (higher $\Delta\Delta G$ values) interactions with eEF1A (Fig. 1a, bottom right) and the energy contribution of different base pairs (positions 49–65, 50–64 and 51–63) were taken from ref. 23 (Methods). Position-specific changes in either anticodon-stem (A$_i$ variants) or TΨC-stem (T$_i$ variants) enhanced the readthrough efficiency of tS, and the simultaneous modulation of both anticodon-stem and TΨC-stem displayed the most robust effect (for example, variants tSA1T5 and tSA2T5; Fig 1c). The tS variants with a TΨC-stem interacting less stably with eEF1A (such as tST2 and tST5) exhibited higher suppression efficacy than tST6 (Fig.1c), which had the most stable interaction with eEF1A (Fig.1a, bottom right table). Enhancement of the suppression efficacy of sup-tRNA charged with serine (a stabilizing amino acid[23]) was achieved by modest stabilization of the interactions with eEF1A.

Arginine makes nearly no thermodynamic contribution to the aminoacyl-tRNA stability[23]. tR variants with substitutions in the TΨC-stem, which would stabilize the interactions with eEF1A, exhibited higher readthrough efficiency (that is, tRT6 the highest and tRT9 the lowest), whereas changes within the anticodon-stem alone (tRA2) or in combination with the TΨC-stem (tRA2T5) markedly reduced the PTC suppression (Fig. 1c). Glycine is a destabilizing amino acid[23], however mutations in the TΨC-stem stabilizing the interactions with eEF1A

(tGT6) marginally enhanced the tG readthrough efficiency (Extended Data Fig. 1 and Supplementary Table 1). Overall, the tG variants were less effective than the tS and tR variants, probably owing to the intrinsic hyper-accuracy of natural glycine tRNAs[22]. Yet, for some PTCs this might be still sufficient to potentially address diseases in which the therapeutic threshold is low, such as cystic fibrosis[24]. Together, our findings indicate that to engineer native tRNAs to decode PTCs, unique design principles should be established for each tRNA family to trim accuracy in decoding and affinity for eEF1A tailored to the chemical nature of the cognate amino acid.

## sup-tRNA efficacy at different PTCs

For one of the optimal tRNA variants with the highest readthrough activity (tSA1T5), which incorporated predominantly Ser at the UGA stop codon (Extended Data Fig. 3), we next assessed the suppression efficacy at UGA, UAG and UAA as pathogenic nonsense mutations lead to all three PTC identities[1] (with a frequency of 38.5% for UGA, 40.4% for UAG and 21.1% for UAA). Within the screening process, to test the sup-tRNA efficacy in another pathogenic mutation context, we considered the S466X mutation in *CFTR*, which occurs naturally with different stop codon identities (https://cftr2.org/). Since this mutation is implicated in cystic fibrosis, we considered the human bronchial epithelial CFBE41o⁻ cell line—an established model in the cystic fibrosis drug expansion pipeline[25]. At an optimal sup-tRNA concentration, which does not alter cell viability (Extended Data Fig. 4a,b), tSA1T5 suppressed both UGA and UAG PTCs with higher efficiency than UAA PTCs (Fig. 1d and Extended Data Fig. 4c,d). Notably, tSA1T5 was substantially more efficient at all PTCs with three different stop codon identities than the known readthrough-stimulating antibiotic G418 (Fig. 1d). We also observed fourfold higher suppression at lower tSA1T5 doses when the sup-tRNA was co-administered with the PTC reporter as mRNA than as DNA (Extended Data Fig. 4e). The efficacy of tSA1T5 at the UGA PTC was greater for the S466X than for R208X (compare Fig. 1d with Fig. 1c), implying that the PTC sequence context also modulates sup-tRNA efficacy—an effect that has been reported for aminoglycosides-stimulated readthrough at natural termination codons[26].

Next, we assessed whether the engineered sup-tRNA stimulates the mammalian innate immune response through activation of human Toll-like receptors (TLRs) and specifically TLR7 and TLR8, which are augmented by synthetic and viral RNA[27]. In a model system established for such analysis[28] (human TLR-transformed HEK293 cells), sup-tRNA did not activate TLR7 and only marginally activated TLR8 to the extent of the mock transfection (Fig. 1e), suggesting that it might be a non-specific effect.

## In vivo efficacy of LNP–sup-tRNA

To assess the suitability of the optimized sup-tRNA as therapeutic agents, we encapsulated them in LNPs and tested the efficacy of their suppression of PTCs and their molecular safety in vivo (Fig. 2a). For intravenous administration of the tS variants in mice, we used LUNAR LNPs, which are similar to those recently established for administration of human factor IX (*F9*) mRNA in a mouse model of liver haemophilia B[21], with a total lipid-to-RNA weight ratio of 25:1 (LUNAR_2021-1, hereafter referred to LUNAR1) (Supplementary Table 3). The LUNAR1 co-formulations of PTC akaluciferase (*aLuc^R208X*) mRNA (0.3 mg mRNA per kg) and either tS or tSA1T5 were delivered in two doses (0.6 and 1.2 mg sup-tRNA per kg). Six hours after intravenous administration, we detected a readthrough of up to 66% for tSA1T5 and 13% for tS (Fig. 2b), suggesting a rapid and efficient production of functional protein. Twenty-four hours after administration, the readthrough efficiency decreased to 40% (Fig. 2b), owing to the overall drop of the amount of *aLuc^R208X* mRNA (Extended Data Fig. 5a). By contrast, the amount of tSA1T5 remained stable for at least 72 h, as detected by tRNA-tailored

microarrays (Fig. 2c and Extended Data Fig. 5b,c). The sup-tRNA stability substantially exceeds the much shorter half-life of mRNA-based therapies and vaccines[29,30].

To access the broader applicability of sup-tRNA for treating other nonsense mutation-linked genetic disorders whose underlying protein is specifically expressed in lung, we next administered tSA1T5 into the lungs of a flox-tdTomato transgenic mouse model by intratracheal microsprayer instillation (Fig. 2a). Here we used another lung-targeting LNP formulation with a total lipid-to-RNA weight ratio of 15:1 (LUNAR-2021-2, hereafter referred to as LUNAR2) (Supplementary Table 3). tSA1T5 was co-administered with PTC-*cre* mRNA harbouring two UGA PTCs (S69X and S82X) at two equal consecutive doses (0.35 mg kg⁻¹ of each tRNA and mRNA on day 0 and day 2; Fig. 2a). We benchmarked the suitability of the PTC-*cre*–*loxP* model in cell culture (Extended Data Fig. 6). In the transgenic *flox*-tdTomato mice, the expression of the fluorescent tdTomato is interrupted by a *loxP*-flanked STOP cassette. External delivery of wild-type *cre* mRNA resulted in robust tdTomato fluorescence across the large and small epithelial airways (Fig. 2d and Extended Data Fig. 7). We administered PTC-*cre* mRNA that rendered *cre* mRNA unable to mediate recombination and consequently no tdTomato fluorescence was detected (Fig. 2e and Extended Data Fig. 7). LUNAR2-encapsulated co-delivery of tSA1T5 with the PTC-*cre* mRNA efficiently restored the tdTomato expression (Fig. 2f and Extended Data Fig. 7), with a pattern similar to that in cells expressing the wild-type *cre* (Fig. 2h and Extended Data Fig. 7); that is, tdTomato was expressed by two major epithelial cell populations in epithelial airways—ciliated (indicated by co-localization with forkhead box protein J1 (FOXJ1), a transcription factor regulating cilia gene expression and motile cilia formation) and secretory (indicated by co-localization with MUC5B, the major gel-forming mucin in lung, which is secreted by airway secretory cells). For the mismatch tRNA, we observed few recovery spots (Fig. 2g and Extended Data Fig. 7), probably owing to the reported low Cre-independent background tdTomato expression for this transgenic mouse.

## Safety of LNP–sup-tRNA in mice

To assess potential off-target effects of sup-tRNA on native stop codons, we analysed the whole lung and liver organs of mice treated with tSA1T5 intratracheally or intravenously, respectively, using ribosome profiling. The readthrough frequency at canonical stop codons was determined using the ribosome readthrough score[26] (Methods). Out of more than 10,000 transcripts (Supplementary Table 4), we detected readthrough events at canonical UGA stop codons of a small number of transcripts (that is, 23 in the lung tissue by intratracheal administration and 15 in the liver by intravenous administration; Extended Data Fig. 8a,b); however, a similar number of transcripts underwent readthrough at UGA codons in the untreated control mice and at the other two stop codons (UAA and UAG) not targeted by the sup-tRNA (Extended Data Fig. 8a,b), indicating that tSA1T5 did not enhance the readthrough at native stop codons beyond the stochastic background readthrough level. Even at transcript-internal UGA sites that are naturally selected for high readthrough efficiency[31], we detected no tSA1T5-triggered enhancement beyond the basal readthrough level (Extended Data Fig. 8c,d). Together, these data indicate efficacious suppression of readthrough in mice, with a good molecular safety profile, and reinforce the potential of sup-tRNA for correcting of PTC-triggered liver or respiratory diseases.

## sup-tRNA efficacy on protein expression

In a pilot experiment to benchmark the efficacy of the tS and tR variants in restoring translation and expression of a full-length disease protein (that is, with intron-less cDNA), we co-transfected CFBE41o⁻ cells with in vitro-transcribed sup-tRNAs and various PTC-*CFTR* variants using Lipofectamine. The sup-tRNAs derived from each tR and

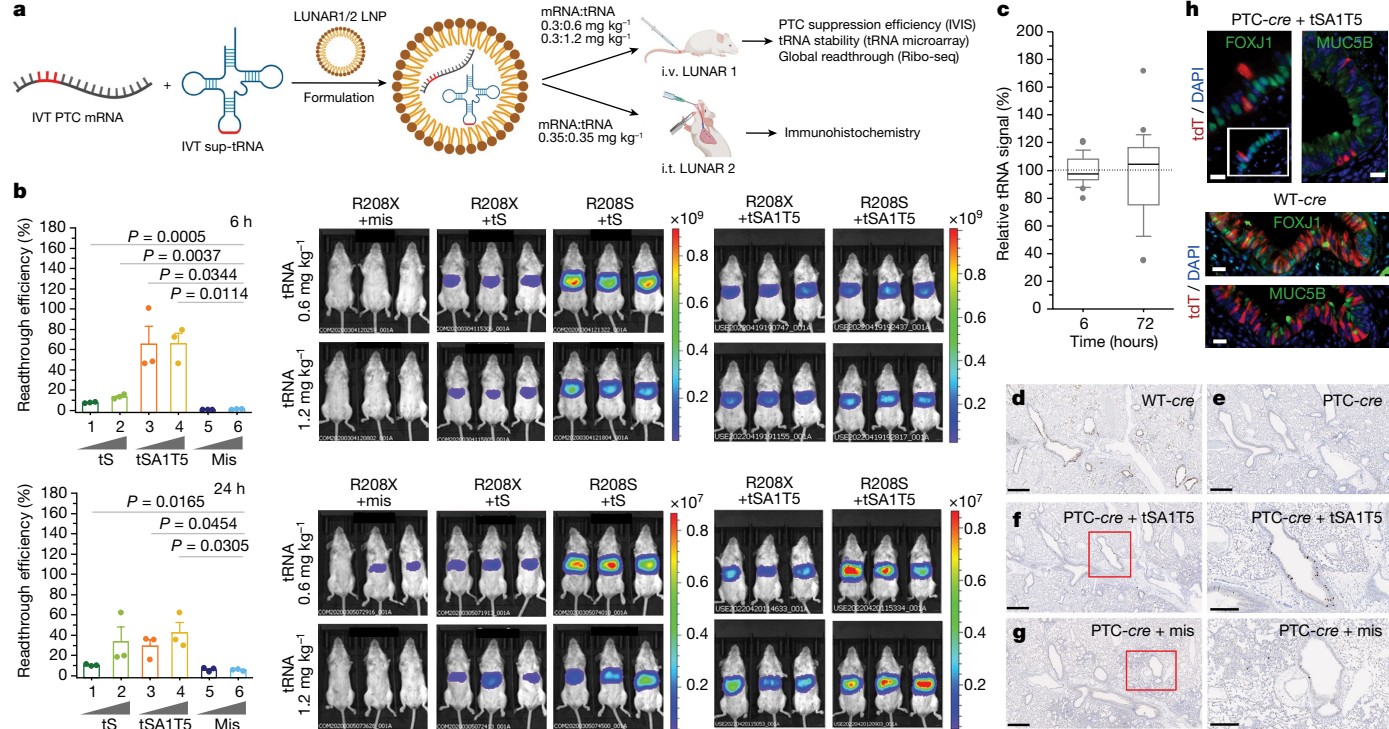

**Fig. 2 | In vivo PTC suppression by LUNAR-encapsulated sup-tRNAs in mice.**
**a**, Workflow of intravenous (i.v.) or intratracheal (i.t.) administration of
LUNAR LNPs with co-encapsulated PTC reporter mRNA and sup-tRNA. For
intravenous administration, $aLuc^{R208X}$ (X represents UGA) reporter and two
different concentrations of tS or tSA1T5 were administered in Balb/C mice.
For intratracheal administration, PTC-*cre* reporter (S69X/S82X, where X
represents UGA) and tSA1T5 were administered in transgenic flox-tdTomato
mice. Ribo-seq, ribosome sequencing. **b**, Quantification of the PTC
readthrough (left) represented as a ratio of the signal from aLuc(R208X) mice
relative to the corresponding aLuc(R208S) mice treated with sup-tRNA from
IVIS in vivo images (right). Groups 1–6 were co-administered with $aLuc^{R208X}$
mRNA. Groups 1, 3, 5: 0.6 mg kg⁻¹ sup-tRNA; groups 2, 4, 6: 1.2 mg kg⁻¹ sup-tRNA.
The mean value of the corresponding aLuc(R208S) plus sup-tRNA group is set
to 100%. Data are mean ± s.e.m. (*n* = 3 per group). One-sided *t*-test. **c**, sup-tRNA
stability in liver monitored with tRNA microarrays represented as a box plot

(36 probes on each array, *n* = 2 independent replicates) normalized to the mean
of the signal at 6 h after intravenous administration, which is set as 100%
(horizontal line). The centre line indicates the median, box edges bound 10th to
90th centiles, and whiskers represent the range of the remaining data without
exclusion of outliers. **d**–**g**, Bright-field immunohistochemistry showing
tdTomato expression (dark brown spots) in lungs from transgenic mice treated
with LUNAR2 LNPs carrying a wild-type *cre* (WT-*cre*) (**d**), PTC-*cre* (**e**) or PTC-*cre*
with tSA1T5 treatment (**f**) or mismatch tRNA (**g**). In **f**,**g**, images on the right are
magnified views of the outlined region in the left image. *n* = 4 mice in each
group (Extended Data Fig. 7). Scale bars, 500 μm; magnified images (**f**, **g**, right),
300 μm. **h**, Immunofluorescence of tdTomato (tdT) and ciliated (FOXJ1) and
secretory (MUC5B) epithelial cell markers in a mouse dosed with a LUNAR2
formulation carrying wild-type *cre* or PTC-*cre* with tSA1T5 treatment. Nuclei
were counterstained with DAPI. Scale bars, 25 μm.

tS (for example, tSA1T5, tSA2T5, tRT5 and tRT6), which showed high
readthrough efficacy in in vitro screening (Fig. 1c), displayed different
efficiencies in restoring full-length CFTR (that is, fully glycosylated,
mature CFTR (band C)), with tS variants resulting in up to 75% and tR
variants resulting in up to 27% of the expression level of cells transfected
with wild-type *CFTR* (Fig. 3a and Extended Data Fig. 9a). Overall, tSA1T5
and tSA2T5 exhibited higher restoration efficacy than tRT5 and tRT6
(Fig. 3a and Extended Data Fig. 9a); this is in stark contrast to their similar
readthrough activity with the reporter constructs (Fig. 1c), implying the
importance of the much larger PTC sequence context on the sup-tRNA
efficacy. Of note, there is a high variation in the expression levels of
wild-type CFTR (Fig. 3a), which is reported to be a consequence of its
complex biogenesis and the simultaneous degradation of endoplasmic
reticulum-retained immature CFTR forms[32]. Using ribosome profiling,
we determined that sup-tRNA uniformly restored translation, as exem-
plified by the effect of tRT5 on *CFTR*^R553X^, which restored the translation
level to 22% of the level of cells transfected with wild-type *CFTR* (Fig. 3b).
The uniform ribosome coverage (that is, the nearly equal mean coverage
of the ribosome profiling spectra) upstream and downstream of the PTC
is indicative of no substantial ribosomal drop-off at the PTC (Fig. 3b).

Nonsense mutations at arginine codons are the most common
PTC mutations in patients with cystic fibrosis (https://cftr2.org/).

Consequently, for the subsequent studies we focused on the engi-
neered tR variants and compared their efficacies in rescuing protein
expression and function of two PTC-containing *CFTR* variants, *CFTR*^R553X^
and *CFTR*^R1162X^. tRT5 effectively augmented the expression of *CFTR*^R553X^
and *CFTR*^R1162X^ by up to 27% and 10%, respectively (Fig. 3a), implying
sequence context dependence of sup-tRNA efficacy. We next measured
the activity of CFTR ion channels in Fischer rat thyroid (FRT) cells—a
standard cellular model viewed by the US Food and Drug Adminis-
tration as informative for drug label expansion of CFTR modulator
compounds[25]. FRT cells were modified to stably express intron-less
full-length *CFTR*^R553X^ or *CFTR*^R1162X^. Both tR and tRT5 restored CFTR chan-
nel activity, with similar efficacies for each PTC-CFTR variant, and aug-
mented channel activity of CFTR(R553X) and CFTR(R1162X) by up to
9% and 3%, respectively (Fig. 3c and Extended Data Fig. 9b,c), thereby
mirroring overall the protein expression levels following sup-tRNA
treatment (compare Fig. 3c with Fig. 3a).

## sup-tRNA antagonizes NMD

It is well documented that native transcripts containing nonsense
mutations are susceptible to NMD[15,33,34], but efficient readthrough with
chemical compounds or sup-tRNAs antagonize NMD on PTC-containing

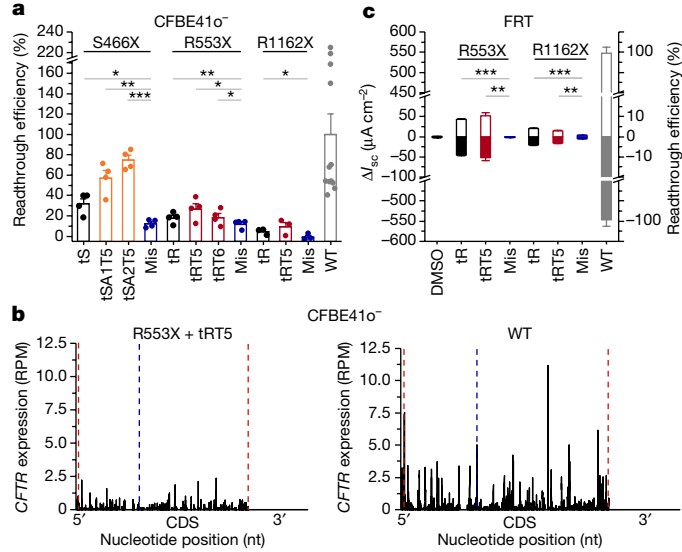

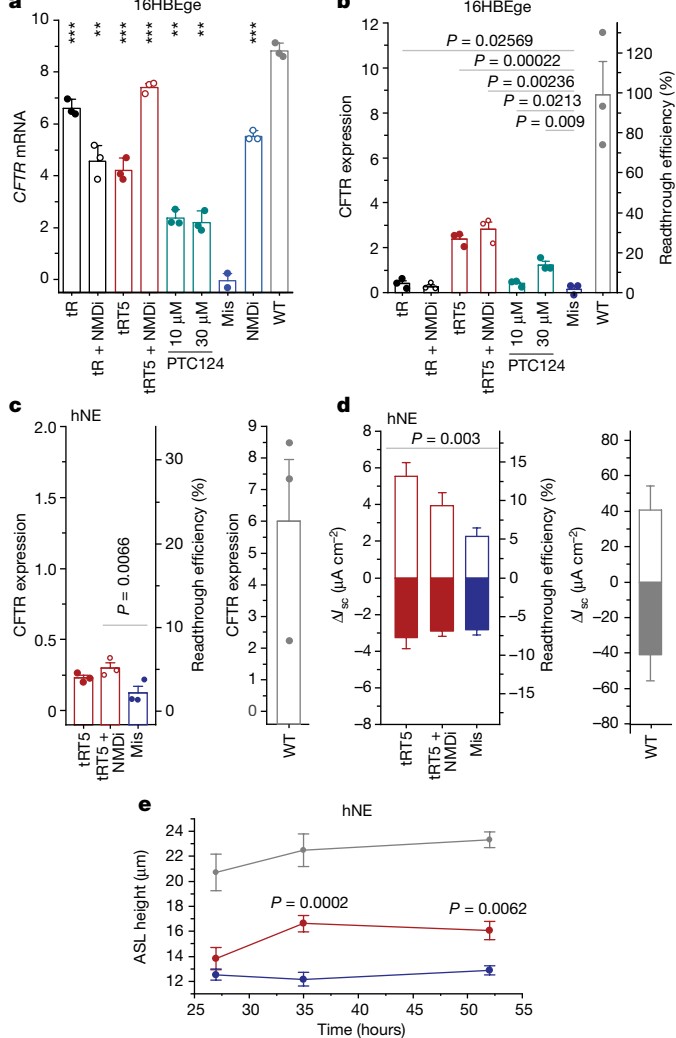

**Fig. 3 | PTC suppression and restoration of mRNA translation and protein function in cell models. a**, Efficacy of tS and tR variants in restoring expression of full-length CFTR from CFTR(S466X), CFTR(R533X) or CFTR(R1162X) (where X represents the UGA codon) in CFBE41o⁻ cells, monitored by immunoblot (Extended Data Fig. 9a). Full-length CFTR (band C) expression is normalized to that in CFBE41o⁻ cells with wild-type *CFTR*. Data are mean ± s.e.m. S466X and R553X: *n* = 4; R1162X: *n* = 3; wild type: *n* = 12 independent replicates). One-sided *t*-test. **b**, Ribosome density profile of tRT5-suppressed *CFTR*^R533X^ mRNA translation monitored by ribosome profiling (total expression, 27 reads per kilobase per million mapped reads (RPKM)) compared with wild-type *CFTR* mRNA (129 RPKM) in CFBE41o⁻ cells. Red dashed lines denote the start and stop of *CFTR* coding sequence (CDS); blue dashed line denotes the first nucleotide of the UGA PTC. RPM, reads per million mapped reads. **c**, Short-circuit current (Δ*I*~sc~) of CFTR(R533X)- and CFTR(R1162X)-expressing FRT cell monolayers transfected with tR or tRT5 compared with cells expressing wild-type CFTR. Positive values (white bars) indicate Δ*I*~sc~ following CFTR activation (with forskolin and VX-770) and negative (solid bars) indicate Δ*I*~sc~ following CFTR inhibition (with Inh-172 inhibitor). Data are mean ± s.e.m. DMSO: *n* = 5; R553X, mismatch, *n* = 5; R553X, tRT5: *n* = 4; R553X, tR: *n* = 3; R1162X, tRT5: *n* = 4; R1162X, mismatch: *n* = 4; R1162X, tR: *n* = 7; wild type: *n* = 3 independent replicates. One-sided *t*-test. For gel source data, see Supplementary Fig. 1.

mRNAs[7,26,35,36]. To assess the effect of NMD on sup-tRNA-mediated mRNA utilization, we used two systems that endogenously express full-length *CFTR*^R1162X^ (that is, with all introns and exons): (1) the gene-edited bronchial epithelial cell line 16HBEge (*CFTR*^R1162X/−^; where X represents UGA), and (2) human nasal epithelial (hNE) cells obtained by non-invasive nasal brushings from patients with cystic fibrosis harbouring the homozygous nonsense mutation *CFTR*^R1162X^, where X represents UGA. Using Lipofectamine, we transfected in vitro-transcribed tRT5 or tR into untreated 16HBEge cells or cells treated with an NMD inhibitor (5 μM NMD14) and assessed mRNA and protein expression. tR and tRT5 alone markedly stabilized the expression of full-length *CFTR* mRNA at levels similar to those achieved with the NMD inhibitor alone, with tR having a greater stabilization effect than tRT5 (Fig. 4a). A mismatched tRNA did not stabilize *CFTR* mRNA (Fig. 4a), suggesting that the effect is specific to the sup-tRNA. Combined treatment with tR or tRT5 and NMD inhibitor augmented the levels of *CFTR* mRNA, but the effect was not uniformly additive for both tRNAs (Fig. 4a), and only marginally enhanced full-length CFTR protein (band C) expression compared with tRT5 alone (Fig 4b and Extended Data Fig. 10a). For comparison, PTC124 (also known as ataluren), which is clinically approved for the treatment of Duchenne muscular dystrophy and has been shown to confer readthrough at PTCs[10,37], modestly stabilized *CFTR* mRNA and

**Fig. 4 | Restoration of CFTR expression and activity by outcompeting NMD. a**,**b**, Efficacy of tR or tRT5 with or without NMD inhibitor (NMDi, 5 μM NMD14) compared with treatment with PTC124 in augmenting *CFTR* mRNA (**a**) or CFTR(R1162X) protein (**b**; band C) expression in 16HBEge cells (cells were wild type or *CFTR*^R1162X/−^, where X represents UGA). **a**, *CFTR* mRNA level was normalized to that in untreated cells. **b**, Band C intensity was normalized to the total protein (left *y*-axis) and to wild-type CFTR (right *y*-axis). Data are mean ± s.d. of *n* = 2 independent replicates for mismatch (**a**) and mean ± s.e.m. of *n* = 3 independent replicates for all other data in **a**,**b**. One-sided *t*-test. **c**, Left, efficacy of tRT5 with and without NMD inhibitor (5 μM NMD14) for restoration of CFTR protein expression (band C is normalized to total protein (left *y*-axis)) in *CFTR*^R1162X/R1162X^ hNE cells (X represents UGA), monitored by immunoblot. Right, CFTR expression in hNE cells from non-cystic fibrosis individuals who are wild-type for the *CFTR* gene. Data are mean ± s.e.m. (*n* = 3 independent replicates) and are shown as a percentage of the mean expression in CFTR wild-type cells. One-sided *t*-test. **d**, Left, short-circuit current measurements in *CFTR*^R1162X/R1162X^ hNE cells (X represents UGA) with tRT5 alone or with tRT5 plus NMD inhibitor (0.5 μM SMG1) (right). Right, short-circuit currents in hNE cells from non-cystic fibrosis individuals who are wild-type for the *CFTR* gene. Positive values (white bars) indicate Δ*I*~sc~ following CFTR activation with forskolin and VX-770 and negative values (solid bars) indicate Δ*I*~sc~ following CFTR inhibition with Inh-172. Data are mean ± s.e.m. (tRT5 and mis: *n* = 8; tRT5 + NMDi and wild type: *n* = 4, independent replicates) and are shown as a percentage of the mean of CFTR expression in wild-type cells. One-sided *t*-test. **e**, ASL height measurement on *CFTR*^R1162X/R1162X^ hNE cells (X represents UGA) co-transfected with tRT5 or mismatch tRNA, compared with CFTR expression in wild-type cells. Data are mean ± s.e.m. (*n* = 5 independent replicates). Two-way ANOVA with Sidak's multiple comparisons. For gel source data see Supplementary Fig. 1.

enhanced protein expression to a level similar to that of tR with and without NMD inhibitor (Fig. 4a,b). In proliferating 16HBEge cells, the transfected sup-tRNA remained stable for at least 72 h (Extended Data Fig. 5c), thus corroborating the stability observed in mice (Fig. 2c).

## sup-tRNA efficacy in hNE cells

hNE cells derived from patients with cystic fibrosis who are homozygous for the R1162X mutation were grown at an air–liquid interface, differentiated into a ciliated pseudostratified epithelial monolayer and transfected with tRT5 alone or in combination with NMD inhibitor treatment. For transfection, we used Lipofectamine, as recommended when working with isolated cells[38]. LNPs are more effective in vivo, probably because endocytosis-driven uptake of LNPs is affected by gene expression changes that occur when cells are removed from their natural environment[38]. Combined treatment with tRT5 and the NMD inhibitor NMD14 slightly increased the expression of full-length CFTR over the expression level with tRT5 alone (Fig. 4c and Extended Data Fig. 10a), corroborating our results in 16HBEge cells (Fig. 4b). In hNEs obtained by non-invasive nasal brushings from individuals without cystic fibrosis, the expression and activity of wild-type CFTR varied over a fourfold range (Fig. 4c,d); in comparisons, we used a mean value across individuals. We also noted alterations of hNE viability and wild-type CFTR expression following prolonged treatment with the NMD inhibitor (Extended Data Fig. 10b), and to exclude effects driven by the NMD inhibitor, we considered another NMD inhibitor (SMG1). tRT5 alone effectively augmented ion transport by up to 14% of the wild-type activity (Fig. 4d and Extended Data Fig. 10c), which exceeds the widely used therapeutic threshold for CFTR activity[39,40] of approximately 10%. Combined treatment with NMD inhibitor weakened the effect (Fig. 4d). Considering the intrinsically low transfection efficiency of hNE cells (approximately 20% in these studies), we acknowledge that much higher efficacy of restoration of CFTR function might be achieved in vivo. Together, our results suggest that engineered sup-tRNA alone successfully outcompetes mRNA surveillance mechanisms and that combined NMD inhibitor therapy in patients may cause adverse effects[41].

Next, we tested the ability of tRT5 to restore hNE cell function, as indicated by the thickness of the airway surface liquid (ASL)—the tightly regulated thin liquid layer that has a major role in mucus clearance and lung defence against infection and is used as a predictor of the therapeutic outcome[42]. tRT5 restored the ASL height on hNE cells after 35 h (Fig. 4e), reinforcing its potential clinical benefits.

## Discussion

Here we demonstrate that native tRNAs can be modified to efficiently decode clinically important nonsense mutations. Engineering of tRNAs to decode various nonsense mutation-derived PTCs, including the most difficult to correct UAA PTC, involves altering tRNA sequences that modulate decoding accuracy[18,20] (through anticodon-stem mutations) and binding affinity to elongation factor[19] (through modulation of the TΨC-stem), framed in terms of the individual thermodynamic contribution of the amino acid to be inserted at the PTC. We showed that at PTCs that cause cystic fibrosis disease, an optimized suppressor tRNA alone restored protein expression and function, and airway volume homeostasis in a manner suggesting potential clinical benefit for individuals with cystic fibrosis. In this case, the addition of NMD inhibitors as adjuvants may not be necessary. Although NMD inhibitors may lengthen the kinetic window for PTC suppression strategies[43], the treatment and dose to block NMD must be carefully tuned to avoid widespread misregulation of gene expression[41].

This study therefore provides a framework for the development of tRNA-based PTC inhibitors for administration as lipid nanoparticles that are shaped to an individual PTC context and relevant amino acid as a means to advance the precision and individualized therapy of hereditary diseases caused by nonsense mutations.

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

# Methods

## Plasmid and RNA constructs

To test the readthrough efficiency of sup-tRNAs, we used two different constructs. First, a dual firefly luciferase–*Renilla* luciferase (*FLuc-RLuc*) reporter containing 15 codons from tripeptidyl peptidase 1 gene (codons 201–215) downstream of the AUG codon of *FLuc* (*FLuc^R208X^-RLuc*). The tripeptidyl peptidase 1 (TPP1) gene associated with the autosomal recessive progressive lysosomal disorder, late infantile neuronal ceroid lipofuscinosis (CLN2). As a control, when using the tSA1T5 suppressor charged with Ser, we replaced the UGA PTC in *FLuc^R208X^-RLuc* with the AGC codon encoding Ser yielding *FLuc^R208S^-RLuc*. The expression of RLuc is controlled by the coxsackievirus B3 internal ribosome entry site. Second, stretch of 15 codons (45 nt) from different disease-related genes centred at the respective PTC mutations (Supplementary Table 2) were inserted into pGL4.51 (Promega) harbouring the *luc2* gene, at the 5′ *luc2* CDS, directly after the AUG start codon, yielding PTC-*FLuc* variants.

For the in vivo experiments, the akaluciferase gene[44] (*aLuc*) was synthesized de novo (Genewiz) and cloned into pARM2379[45]. The coding sequence (CDS) of *aLuc* was extended 5′ upstream in-frame (after the *aLuc* ATG start codon) by 15 codons (45 nt) centered at disease-related PTCs, e.g. R208X of human tripeptidyl peptidase 1 gene TPP1 (codon positions 201–215) (Supplementary Table 2). R208 was mutated to R208X (UGA, UAG or UAA) to mimic the human PTC (*aLuc^R208X^*) or to R208S (*aLuc^R208S^*) to be used as positive control. For in-cell experiments, 15 codons flanking R208X of TPP1 and S466X (codon positions 459–473) of the human *CFTR* gene (Supplementary Table 2) were fused to *FLuc* gene yielding *FLuc^R208X^* and *FLuc^S466X^*. S466 was mutated to UAA, UAG or UGA.

The *cre* gene[46] was de novo synthesized (Genewiz) extended at its 5′ end with SV40 large T-antigen nuclear localization signal (amino acid sequence PKKKRKV) to facilitate recombination efficiency[47]. Nonsense mutations at positions Ser69 and Ser82 were introduced by PCR-based mutagenesis and de novo gene synthesis with gBlock (Integrated DNA Technology).

## Cell lines and primary cells

Hep3B (HB-8064) and Hepa1-6 (CRL-1830) cell lines were obtained from the ATCC. A Cre-*loxP* reporter was integrated in HEK293T (CRL-3216) cells (generated by R. Trelles). The immortalized cystic fibrosis bronchial epithelial cell line CFBE41o⁻ (generated by D. Gruenert) with no allelic CFTR expression was used for ectopic expression of CFTR variants. Transepithelial ion transport was measured in Fischer rat thyroid (FRT) cells stably expressing *CFTR^R553X^*, *CFTR^R1162X^* or wild-type *CFTR*.

16HBE14o⁻ cells, the immortalized version of human bronchial epithelial cells expressing wild-type *CFTR* (including all introns; generated by D. Gruenert) were obtained from M. Lalk[48]. 16HBE14o⁻ cells were gene-edited at the endogenous *CFTR* locus using CRISPR–Cas9 to create isogenic 16HBEge *CFTR^R553X/−^* and *CFTR^R1162X/−^* cell lines[49]. Both 16HBEge cell lines were obtained from the Cystic Fibrosis Foundation Therapeutics Lab.

Primary hNE cells were collected by nasal brush from an individual with cystic fibrosis who was homozygous for *CFTR^R1162X^*, and were obtained at passage 2 from the Cystic Fibrosis Foundation Therapeutics Lab.

IL-8 response was monitored in HEK293XL-TLR7 or HEK293XL-TLR8 cells (Invivogen). The cells were aliquoted at low passage number and stored in liquid nitrogen. The cells were regularly tested for mycoplasma contamination using Venor GeM PCR-based detection kit (Merck).

## tRNA design

In the isoacceptors, tRNA^Ser^UGA/tRNA^Ser^AGA, tRNA^Arg^UCU and tRNA^Gly^UCC, the anticodon was exchanged to UCA produce tS, tR and tG,

respectively (Supplementary Table 1). Previous studies have suggested that among the tRNA families, these isoacceptors (such as tRNA^Ser^(AGA) and tRNA^Ser^(UGA), tRNA^Arg^(UCU) and tRNA^Gly^(UCC)[4]) were among those with the highest readthrough following the exchange of their native anticodon to decode a stop codon.

For TΨC-stem tRNA variants (T_i variants), we estimated the $\Delta\Delta G°$ values for binding affinities to eEF1A considering the cumulative contribution of the three TΨC-stem base pairs 49–65, 50–64 and 51–63 (Fig. 1a, bottom right). The eukaryotic (eEF1A) and bacterial (EF-Tu) elongation factors share conserved sites of aminoacyl-tRNAs binding[50], thus, the $\Delta\Delta G°$ value for each nucleotide pair was taken from the bacterial EF-Tu–tRNA^Phe^ complex[23,51].

Both anticodon-stem and anticodon-loop have coevolved with the anticodon to ensure faithful decoding. To increase the decoding, U–A or A–U pairs are preferred at nucleotide positions 31 and 39 in native sup-tRNAs[52], thus, we considered them in our A1 variants (Fig. 1a and Extended Data Fig. 1). In the A2 variants, the anticodon-stem is taken from tRNA^Sec^. The identity elements—that is, nucleotides recognized by the cognate aminoacyl-tRNA synthetase to aminoacylate tRNA—were preserved. For example, in the tRNA^Ser^ family, the discriminator base G73 and the V-region[53] act as identity elements, in the tRNA^Arg^ family they are G73, A20 and C35–U/G36, and in the tRNA^Gly^ family they are A73, C2–G71[53].

## tRNA transcription

In vitro-transcribed tRNA variants were used for co-transfection in the immortalized cell culture models, patient-derived primary cells and in vivo, in mice.

tRNAs were transcribed in vitro using T7 transcription system as described[17]. In brief, two partially overlapping DNA oligonucleotides encoding the corresponding tRNA sequence with an upstream T7 promoter (5′-TAATACGACTCACTATA-3′) were used for in vitro tRNA synthesis. A 24 µM solution of both oligonucleotides was denatured for 2 min at 95 °C, and thereafter aligned for 3 min at room temperature in 20 mM Tris-HCl (pH 7.5), and 0.4 mM dNTPs were added and incubated with 4 U µl⁻¹ RevertAid Reverse Transcriptase (Thermo Fisher Scientific) for 40 min at 37 °C. This dsDNA template was purified with phenol/chloroform, washed with 80% ethanol and resuspended in DEPC-treated water. Alternatively, dsDNA templates were prepared by PCR reaction on the tSA1T5-expressing plasmid with a forward primer annealing upstream of the T7 promoter and a reverse primer (5′-TGGCGTAGTCGACGGGATTC-3′) with or without 2′-*O*-methyl modification of the first two nucleotides of the 5′ end[54]. For in vitro T7 transcription, 2 mM NTPs, 5 mM GMP, 1× transcription buffer, 0.6 U µl⁻¹ T7 RNA polymerase (Thermo Fisher Scientific) were added to the dsDNA template and incubated overnight at 37 °C. The tRNA variants were resolved by preparative denaturing polyacrylamide gel electrophoresis (PAGE) and eluted in 50 mM potassium acetate, 200 mM KCl pH 7.0 overnight at 4 °C, followed by ethanol precipitation and re-suspension in DEPC-treated water.

tRNAs for in vivo delivery were transcribed by T7 RNA polymerase from linearized plasmids under similar conditions to those described above and purified as previously described[21,45]. The integrity of the purified tRNAs was monitored by toluidine blue staining (0.4% (w/v)) in a 10% denaturing TBE-Urea gel (Thermo Fisher Scientific).

## In vitro transcription of mRNA

To generate high-quality mRNA transcripts for the in vivo experiments or for co-transfecting in vitro-transcribed tRNA and mRNA in cell systems, we adopted methods described previously[21,45]. The mRNA transcripts were purified through a silica column (Macherey-Nagel). Dual *FLuc^R208X^-RLuc* mRNA was in vitro-transcribed with unmodified nucleotides, while *aLuc^R208X^* and *cre* mRNAs were synthesized with UTPs fully substituted with N1-methyl-pseudouridine. The purified mRNAs were quantified by UV absorbance and their purity (% full-length) and

integrity verified by Fragment Analyzer (Agilent). Transcripts were stored in RNase-free water below −60 °C until in vitro transfection or formulation with LNPs for in vivo administration.

## tRNA transfection and in vitro luciferase readthrough assay

Hep3B, Hepa1-6 or CFBE41o⁻ cells were seeded in 96-well cell culture plates at $1 \times 10^4$ cells per well and grown in Dulbecco's Modified Essential Medium (DMEM, Pan Biotech or Gibco) for Hep3B and Hepa1-6 or Minimum Essential Medium (MEM, Pan Biotech) for CFBE41o⁻ cells. All medium was supplemented with 10% fetal bovine serum (FBS, Pan Biotech) and the medium for CFBE41o⁻ cells was also supplemented with 2 mM L-glutamine (Thermo Fisher Scientific). Sixteen to twenty-four hours later, Hep3B or CFBE41o⁻ cells were co-transfected in triplicate with 25 ng PTC-*FLuc* or wild-type *FLuc* plasmids and 100 ng each in vitro-transcribed tRNA variant using Lipofectamine 3000 (Thermo Fisher Scientific). After 4–6 h, medium was replaced and 24 h after transfection cells were lysed with 1× passive lysis buffer (Promega) and luciferase activity measured with luciferase assay system (Promega) and Spark microplate reader (Tecan). G418 was added to the cells at concentration 25 µg ml⁻¹ and incubated for 24 h.

Twenty-four hours after seeding, Hepa1-6 cells were transfected with 12.5 ng of 1 of the 3 in vitro-transcribed reporter mRNAs (*FLuc^R208X^-RLuc*, *FLuc^R208S^-RLuc* or *FLuc*^R208^*-RLuc* with Arg at position 208) together with 50 ng of in vitro-transcribed tRNA using MessengerMax (0.2 µl per well). For in vitro dose-response experiment (Extended Data Fig. 4), 50–0.78 ng of the in vitro-transcribed tRNA was serially diluted by twofold and co-transfected into each well with the reporter mRNA (12.5 ng). To achieve similar transfection efficiencies across different dosages of PTC-pairing tRNA, an in vitro-transcribed mismatch tRNA that does not pair to the UGA PTC was used as filler so that total tRNA of 50 ng per well was always co-transfected. Cells were incubated at 37 °C overnight, rinsed with PBS and collected by adding 20 µl per well of 1 × passive lysis buffer (Promega). Luciferase activities were measured in 10 µl lysate with the Dual-Luciferase Reporter Assay kit (Promega) on Spark microplate reader (Tecan).

## tRNA toxicity

CFBE41o⁻ cells were transfected with an in vitro-transcribed sup-tRNA in a concentration series from 4,000 to 7.81 ng per well and serially diluted by twofold as described above. Cell viability was determined using the CellTiter-Glo luminescent cell viability assay (Promega) according to the manufacturer's instructions.

## tRNA-induced stimulation of TLR7- or TLR8-expressing cells

Human TLR-transformed HEK293 cells are an established system to analyse tRNA-induced stimulation of human TLR7 and TLR8 and the IL-8 response was monitored as described in[28]. In brief, HEK293XL cells stably transfected with hTLR7 and hTLR8 (Invivogen) were seeded in 96-well cell culture plates at $5 \times 10^4$ cells per well and cultured in DMEM (Pan Biotech) supplemented with 10% FBS (Pan Biotech) and 2 mM L-glutamine (Thermo Fisher Scientific). Twenty-four hours later, cells were transfected with in vitro-transcribed tSA2T5 (1 µg ml⁻¹) or resiquimod (R848; Invivogen, 1 µg ml⁻¹) as control activators of the TLR7–TLR8 signalling pathway[55] using Lipofectin (Thermo Fisher Scientific). After 3 h, medium was replaced and 16 h post-transfections, interleukin IL-8 in supernatants was measured using human IL-8 ELISA Kit II (BD Biosciences) according to the manufacturer's instructions.

## Mass spectrometry

Hepa1-6 cells were seeded in two 10-cm dishes at $1 \times 10^6$ cells per dish. Sixteen to twenty-four hours after seeding, the cells were co-transfected with 2 µg in vitro-transcribed dual *FLuc^R208X^-RLuc* mRNA or dual *FLuc^R208^-RLuc* and 4 µg in vitro-transcribed tSA1T5 using MessengerMax (Thermo Fisher Scientific). Cells were incubated at 37 °C overnight and collected by trypsinization. Mock-transfected cells were used as a negative control. Cells were rinsed four times with PBS and analysed with 2D nano PRM liquid chromatography–tandem mass spectrometry (LC–MS/MS) by Jade Bio.

## Formulation of the LUNAR–RNA nanoparticulate liposomes

LNPs were produced using LUNAR, a proprietary lipid nanoparticle technology platform, at Arcturus Therapeutics. The LNPs were prepared as described previously[21,56]. Appropriate volumes of lipids dissolved in ethanol at the desired ratios were mixed with an aqueous phase containing RNA using a microfluidic device, followed by downstream processing. For the encapsulation of RNA, ~2 mg ml⁻¹ RNA was dissolved in 5 mM citrate buffer (pH 3.5). The molar percentage ratio for the constituent lipids is 50% ionizable amino lipids, 7% 1,2-distearoyl-*sn*-glycero-3-phosphocholine (Avanti Polar Lipids), 41.5% cholesterol (Avanti Polar Lipids), and 1.5% 1,2-dimyristoyl-*sn*-glycerol, methoxypolyethylene glycol (polyethylene glycol chain molecular mass: 2,000) (NOF America). At a flow ratio of 1:3 ethanol:aqueous phases, the solutions were combined in the microfluidic device. The total lipid-to-RNA weight ratio was ~25:1 (LUNAR1) or 15:1 (LUNAR2). The mixed material was then diluted 3 times with Tris buffer (pH 7.5) containing 50 mM NaCl and 9% sucrose after leaving the micromixer outlet, reducing the ethanol content to 6.25%. Diluted LNP formulation was concentrated and diafiltered by tangential flow filtration using hollow fibre membranes (mPES Kros membranes, Repligen) and Tris buffer (pH 7.5) containing 50 mM NaCl and 9% sucrose to remove the ethanol. Particle size and polydispersity index (PDI) were characterized using a Zen3600 (Malvern Instruments, with Zetasizer 7.1 software). Encapsulation efficiency was calculated by determining the unencapsulated RNA content by measuring fluorescence intensity (Fi) upon addition of RiboGreen (Molecular Probes) to the LNPs and comparing this value to the total fluorescence intensity (Ft) of the RNA content obtained upon lysis of the LNPs in 1% Triton X-100. The percentage encapsulation was calculated by the ratio (Ft − Fi)/Ft × 100%). All LNPs were associated with encapsulation efficiencies of >90%.

## Mouse experiments and imaging

All mice were purpose-bred and experimentally naive at the start of the study. Mice were chosen randomly for treatment with either control or experimental conditions without blinding. Mice were housed 5 per cage in a pathogen-free environment in Innovive disposable IVC rodent caging system with a 12 h light/dark cycle, at temperature between 19–22 °C and humidity 50–60%. Ad libitum access to standard diet (2018, Global 18% protein rodent diet from Envigo+++) and pre-filled acidified water from Innovive (pH 2.5–3.0) were used throughout the study period. The bedding material was hardwood chips (Sani-Chips, 7115, Envigo++++) and cages were changed biweekly. All in vivo procedures involving animals were performed at Arcturus Therapeutics in accordance with the animal use protocols and policies approved by the Institutional Animal Care and Use Committee (IACUC), protocol (EB17-004-003 from 1 February 2017 and latest amendment from 17 June 2021). The vivarium is managed by an AAALAC approved vendor Explora BioLabs (A Charles River Company).

Intravenous administration and in vivo imaging: 8–10 week old, female Balb/C mice were purchased from Charles River Laboratories. LUNAR1 formulations were administered intravenously at 0.9 or 1.5 mg kg⁻¹ (that is, 0.3 mg kg⁻¹ luciferase mRNA and 0.6 or 1.2 mg kg⁻¹ sup-tRNA) on day 0. Six and twenty-four hours after dosing, Akalumine-HCL (TokeOni, Sigma Aldrich, 808350-100MG, lot no. MKCL1624, 15 mg ml⁻¹) was injected intraperitoneally at 100 µl per mouse followed by in vivo imaging. Ten minutes after administration of the luminescent Luc substrate, live animal bioluminescence imaging was performed on mice anaesthetized with 2% isoflurane using IVIS Lumina III (Perkin Elmer). Images were quantified by the region of interest for total FLuc signal using Living Image Software (Perkin Elmer). In total, 33 mice were subjected to intravenous administrations. Naive

animals within the same age group were stratified to different treatment groups; animals were assigned to groups randomly. No statistical tests were used to predetermine the sample size. Six mice were dosed with LUNAR1 co-formulated with in vitro-transcribed $Luc^{R208X}$ mRNA and in vitro-transcribed tS at two different concentrations (each cohort comprising three mice). Six mice were dosed with LUNAR1 co-formulated with mRNA without a PTC ($Luc^{R208S}$) and tS at two different concentrations (each cohort comprising three mice). Another cohort of six mice was dosed with LUNAR1 co-formulated with in vitro-transcribed $Luc^{R208X}$ mRNA and in vitro-transcribed tSA1T5 at two different concentrations (each cohort of three mice). Six mice were dosed with LUNAR1 co-formulated with mRNA without a PTC ($Luc^{R208S}$) and tSA1T5 at two different concentrations (each cohort of three mice). To another cohort of six mice LUNAR1 co-formulated with in vitro-transcribed $Luc^{R208X}$ mRNA and in vitro-transcribed mismatch tRNA not pairing to the UGA PTC at two different concentrations (each cohort comprising three mice) was administered. Three mice were treated with PBS and served as negative control.

Intratracheal administration: six- to ten-week-old, female B6.Cg-Gt(ROSA)26Sortm14(CAG-tdTomato)Hze/J mice were purchased from Jackson Laboratories. LUNAR2 formulations were administered through intratracheal instillation in mice anaesthetized with 2% isoflurane, using a 27G × 1 inch blunt needle (B27-100, SAI) at 0.35 mg kg⁻¹ for wild-type $cre$ mRNA only or 0.7 mg kg⁻¹ of in vitro-transcribed PTC-$cre$ mRNA and in vitro-transcribed tSA1T5 (0.35 mg kg⁻¹ each); 50 µl per mouse was given on day 0 and day 2 (two doses total). Forty-eight hours after the last dose, lungs from anaesthetized mice were collected and insufflated with 10% neutral buffered formalin (NBF). Lungs were then fixed overnight in 10% NBF prior to processing for immunohistochemistry. In total, 24 mice were subjected to intratracheal administrations and animals were assigned to groups randomly. No statistical tests were used to predetermine sample size. Four mice were dosed with LUNAR2 co-formulated with PTC-$cre$ mRNA and tSA1T5 with anticodon pairing to UGA, four with LUNAR2 co-formulated with PTC-$cre$ mRNA and mismatch tSA1T5 with anticodon pairing to UAG, four with LUNAR2 co-formulated with PTC-$cre$ mRNA only, four with LUNAR co-formulated with wild-type $cre$ mRNA only, and four received PBS as a negative control.

Sections were cut at 5 µm, dewaxed and rehydrated into distilled water before antigen retrieval procedure. Sections were incubated with TrueBlack Lipofuscin Autofluorescence Quencher (Biotium) for 1 min, washed in PBS and blocked to suppress non-specific antibody biding using TrueBlack IF Background Suppressor System (Biotium). Sections were incubated with primary rabbit polyclonal RFP antibody (dilution 1:800; 600-401-379, Rockland Antibody), mouse monoclonal FOXJ1 antibody (dilution 1:1,000; 14-9965-82, Thermo Fisher Scientific) or rabbit polyclonal MUC5B antibody (dilution 1:1,000; PA5-82342, Thermo Fisher Scientific) overnight at 4 °C. A M.O.M. kit (Vector, BMK-2002) was used to remove any mouse-on-mouse Ig interference. After washing in PBS, donkey anti mouse AF488 (A-21202, Thermo Fisher Scientific) or donkey anti rabbit AF555 (A-21428, Thermo Fisher Scientific) secondary antibodies were incubated with slides for 1 h. Slides were subsequently counterstained in DAPI and mounted with VECTASHIELD Vibrance Antifade Mounting Medium (H-1700, Vector Laboratories).

### In vitro Cre-$loxP$ system
Stable expression system of Cre recombinase-dependent eGFP was established by transfecting HEK293T cells with pLV-CMV-LoxP-DsRed-LoxP-eGFP[57] (a gift from J. Rheenen), with puromycin selection following a protocol from the Genetic Perturbation Platform at Broad Institute. The selected cells were confirmed with lack of eGFP by EVOS Cell Imaging Systems (Thermo Fisher Scientific). Cells were seeded in 96-well plates at $1.6 \times 10^4$ cells per well and after 24 h transfected with in vitro-transcribed $cre$ mRNA (12.5 ng) and in vitro-transcribed tS variants (25 ng) pairing to different PTCs (tS::UGA, tS::UAG; tS::UAA)

using MessengerMax (0.2 µl per well). Every day within four days after transfection the fluorescence was recorded on Spark microplate reader (Tecan) and reported as relative fluorescence units. The expression of DsRed and eGFP were visualized by EVOS Cell Imaging Systems (Thermo Fisher Scientific).

### Culturing of patient-derived nasal epithelial cells at air−liquid interface
hNE (R1162X/R1162X) cells were seeded in T75 flasks coated with collagen IV (Sigma) or Purecol (Advanced Biomatrix) in 12 ml pre-warmed complete PneumaCult ALI Ex⁺ medium (StemCell kit) at 37 °C in 5% CO₂. To 500 ml medium the following supplements were added: 0.5 ml hydrocortisone (StemCell), 10 ml 50X Ex⁺ supplement (StemCell kit), and for some preparations 2 ml of amphotericin B (12.5 µg ml⁻¹; Sigma), 500 µl ceftazidime (100 mg ml⁻¹; Sigma), 500 µl vancomycin (100 mg ml⁻¹; Sigma), and 500 µl tobramycin (100 mg ml⁻¹; Sigma). Cells were expanded for 3−5 days, until they reached 70−80% confluency.

Cells were then detached by 0.05% trypsin-EDTA (Pan Biotech) or enzymatic (StemCell) treatment and seeded onto 12- or 24-ALI Transwells (0.4-µm pore polyethylene terephtalate membrane inserts, Corning) coated with collagen IV at a confluency of $1.5 \times 10^5$ to $2 \times 10^5$ per well, and Complete Ex⁺ medium (without antibiotics) was added as following, 0.5−0.6 ml (basolaterally) and ~0.5 ml (apically). Cells were grown for 3−4 days at 37 °C with 5% CO₂. Medium were changed every day on the apical and basolateral side. On day 4, the apical medium was removed, and basolateral bathing solution exchanged for ALI complete medium (StemCell), followed by additional exchanges three times per week for at least 21 days until reaching a fully differentiated state.

### Transfection with tRNA and treatment with NMD inhibitors
CFBE41o⁻ cells, 16HBEge $CFTR^{R1162X/-}$ or 16HBE41o− $CFTR^{WT}$ cells were seeded on 12-well cell culture plates at $1 \times 10^5$ cells per well and cultured in Minimum Essential Medium (MEM, Pan Biotech) supplemented with 10% FBS (Pan Biotech) and 2 mM L-glutamine (Thermo Fisher Scientific). The medium of 16HBEge cells was additionally supplemented with 1% penicillin/streptomycin (Gibco) and the culture plates were precoated with 1% human fibronectin (Thermo Fisher Scientific), 1% bovine collagen type I (Advanced BioMatrix), 1% bovine serum albumin (Gibco). At 24 h after seeding, CFBE41o⁻ cells were co-transfected with 400−500 ng PTC-$CFTR$ (S466X, R553X or R1162X variants) or wild-type $CFTR$ plasmids and 800−1,000 ng in vitro-transcribed tRNA variant using Lipofectamine 3000 (Thermo Fisher Scientific). In the experiments with the NMD inhibitor (immunoblot and quantitative PCR with reverse transcription (RT−qPCR) analysis), 16HBEge cells were pre-treated with 5 µM NMD14 (MedChemExpress) for 24 h and then transfected with 800 ng of in vitro-transcribed tRNA variants using Lipofectamine 3000 (Thermo Fisher Scientific). After 4−6 h of tRNA transfection, medium was replaced and cells grown for another 24 h.

Well-differentiated hNE (R1162X/R1162X) cells (see above) were transfected from the apical surface of monolayers with 800−1,000 ng in vitro-transcribed tRNA (tRT5 or mismatch tRNA) using Lipofectamine 3000 (Thermo Fisher Scientific). After 6 h, fresh medium was exchanged and cells were incubated for the next 24 h. Transfection efficiency with Lipofectamine was approximately 18−20% as estimated using co-transfection with fluorescent proteins. In the experiments with NMD inhibitor (NMD14 or SMG1[58,59]), drug was added on the basolateral side. Six hours following addition of the NMD inhibitor, cells were transfected from the apical surface of monolayers with in vitro-transcribed tRNA, while continuing the treatment with the NMD inhibitor. After an additional 6 h (total treatment with NMD14 or SMG1 was 12 h), the medium was replaced and cells grown for additional 24 h. We benchmarked the SMG1 concentration of 0.5 µM to allow mRNA stabilization as comparable to NMD14 at 5 µM. Two inhibitors were used to exclude an inhibitor-specific effect. It should be noted that 16HBEge cells were robust to NMD treatments, whereas donor-derived hNEs exhibited

alterations in viability after extended treatment with both NMD14 or SMG1, thus, NMD treatment should not exceed 12–15 h.

## CFTR immunoblot expression analysis

Cells (CFBE41o⁻ cells, 16HBEge *CFTR*^R1162X/−, 16HBE41o− *CFTR*^WT^ or hNE *CFTR*^R1162X/R1162X^) transfected with tRNA or mock-treated, were then lysed with 80 µl MNT buffer (10×; 300 mM Tris-HCl pH 7.5, 200 mM MES and 1 M NaCl) and lysates were subjected to immunoblotting with monoclonal CFTR-NBD2 antibody (1:100 dilution, 596, J. R. Riordan and T. Jensen) available through the Cystic Fibrosis Foundation Therapeutics Antibody Distribution Program. Analysis utilized the capillary electrophoresis system (Jess, ProteinSimple) as described previously[60]. The peak area corresponding to fully glycosylated CFTR (band C) was normalized to the total protein with the Jess quantification module and to the molecular weight marker (180 kDa peak) to minimize fluctuations between multiple Jess runs. Note that mock-transfected cells—those treated with Lipofectamine—were used as a control due to the slight adverse effect of Lipofectamine. Ataluren (PTC124) was added at a final concentration of 10 µM and 30 µM and incubated for 24 h.

## RT–qPCR

The steady-state mRNA expression of *CFTR* variants transfected with in vitro-transcribed tRNA variants (1,000 ng) or NMD inhibitor (5 µM) was measured using RT–qPCR. Cells were grown and treated the same way as described above for the immunoblot analysis, but to capture the mRNA expression, cells were lysed 6 h after the tRNA transfection (total treatment with NMD14 and tRNA was 12 h and 6 h, respectively). Total RNA was isolated using TRIzol (Invitrogen). 2 µg total RNA was reverse transcribed using random hexamers (Thermo Fisher Scientific) and RevertAid H Minus reverse transcriptase (Thermo Fisher Scientific) in 20 µl total volume. Quantitative PCR was performed using SensiMix SYBR Hi-ROX Kit (Thermo Fisher Scientific) on T Professional thermocycler (Biometra). The region spanning exon 23-exon 24 (nucleotides 3874–4001 in the *CFTR* mRNA) of the *CFTR* transcript was amplified with the following primer pair: forward 5′-GATCGATGGTGTGTCTTGGGA-3′ and reverse 5′-TCCACTGTTCATAGGGATCCAA-3′. *GUSB* transcript was used as a house-keeping expression control whose expression level ranges at the level of *CFTR*^WT^ expression and was amplified using the following primer pair: forward 5′-GACACGCTAGAGCATGAGGG-3′ and reverse 5′-GGGTGAGTGTGTTGTTGATGG-3′. The analysis was performed using $\Delta\Delta C_{T}$ approach. Technical duplicates of each biological replicate reaction were carried out for each sample.

## tRNA stability and tRNA-tailored microarrays

16HBEge *CFTR*^R553X/−^ cells were seeded at $6 \times 10^5$ cells per well into precoated 6-well cell culture plates and grown as described above. 24 h after seeding, cells were transfected with 1.5 µg tSA2T5 or water (as control) in triplicates using Lipofectamine 3000 according to the manufacturer's protocol. Cells were lysed at 5, 24, 36, 48 and 72 h post transfection, by adding 1 ml TRIzol per well (Invitrogen). Total RNA from cells was isolated using the TRIzol method according to manufacturer's instructions and RNA integrity was assessed by 10% denaturing polyacrylamide gel electrophoresis.

Four female Balb/C mice were intravenously dosed with LUNAR1 formulated with 0.6 mg kg⁻¹ in vitro-transcribed tSA1T5. At 6 h and 72 h post treatment, mice were anaesthetized with 3% isoflurane in a VetEquip inhalation anaesthesia system chamber. Livers from the anaesthetized mice were collected and flash frozen. Mice treated with PBS served as a negative control. The whole organs were pulverized in liquid nitrogen and lysed by grinding in 500 µl TRIzol (Invitrogen). Total RNA was isolated using the TRIzol method according to manufacturer's instructions and RNA integrity was assessed on 10% denaturing polyacrylamide gel electrophoresis.

tRNAs were analysed using tRNA-tailored microarrays as previously described[61,62], with some adjustments to measure the sup-tRNA. On the microarrays, tDNA probes covering the full-length tRNA sequence of the 41 cytoplasmic tRNA species complementary to 49 nuclear-encoding tRNA families are spotted, along with the tDNA complementary to tSA1T5 (5′-TGGCGTAGTCGACGGGATTCGAACCCGTGCGGGGAAAC CCCAATGGTTTTGAAGACCATCGCCTTAACCACTCGGCCACGACTAC-3′) or tDNA complementary to tSA2T5 (5′-TGGCGTAGTCGACGGGATT CGAACCCGTGCGGGGAAACCCCAACAGGTTTGAAGCCTGCCGCCTTA ACCACTCGGCCACGACTAC-3′). Each microarray consisted of 12 identical blocks, each containing 2 probes for each natural tRNA and 3 probes for tSA2T5 or tSA1T5 (36 signals in total for each sup-tRNA). To fully deacylate tRNAs, 5 µg of total RNA was incubated for 45 min at 37 °C in 100 mM Tris-HCl buffer (pH 9.0), followed by purification by precipitation with ethanol and 0.1 volume of 3 M sodium acetate (pH 5.5), supplemented with glycogen (20 mg ml⁻¹, Thermo Fisher Scientific). Cy3-labeled RNA:DNA hairpin oligonucleotide was ligated to deacylated 3′-NCCA ends of the tRNAs using T4 DNA ligase (NEB) for 1 h at room temperature. Total RNA from non-transfected cells was used as comparison and labelled with Atto647-labelled RNA:DNA hairpin oligonucleotide. For subsequent normalization of the arrays, each sample was spiked in with three in vitro-transcribed tRNAs (2 µM of each), which do not cross hybridize with any of the human tRNAs or the sup-tRNA. Detailed experimental protocol for tRNA microarrays is available at protocols.io (https://doi.org/10.17504/protocols. io.hetb3en). Scanned microarray slides were analysed using inhouse Python scripts. The median of the ratio of Cy3 to Atto647 signals was normalized to spike-ins whose ratio set to one. Thereafter, each single Cy3 signal from the sup-tRNA was normalized to the Cy3 signal of the spike-ins and represented as a ratio to the mean of the signal at 5 h (for 16HBEge cells) or 6 h (for mouse) which was set as 100%. The arrays were performed in two biological replicates for the samples withdrawn at 5 h, 24 h and 36 h, and in a single replicate for 48 h and 72 h samples. Due to a high reproducibility of the arrays (confidence intervals higher than 98%), following the normalization to the spike-ins the individual signals from the biological duplicates were merged.

## Short-circuit current $I_{sc}$ measurements

Transepithelial ion transport was measured in FRT cells, which represent a standard model for polarizing epithelia expressing apical CFTR and are viewed by the US Food and Drug Administration as informative for drug label expansion for CFTR modulators[63,64]. FRT cells stably expressing *CFTR*^R553X^, *CFTR*^R1162X^ or wild-type *CFTR* were seeded at 100,000–150,000 cells onto permeable supports (0.33 cm² per Transwell insert). Four days after seeding, cells form tight junctions and were transfected with 400 ng in vitro-transcribed tRNA (tR, tRT5 or mismatch tRNA) in 20 µl OptiMEM per insert using Lipofectamine 3000 (Thermo Fisher Scientific). Cells were maintained under air–liquid interface conditions at 37 °C in 5% CO₂ for 24 h. As described above, hNE (R1162X/R1162X) cells were also cultured and treated with NMD inhibitor as indicated, followed by $I_{sc}$ measurement.

Short-circuit current was monitored under voltage clamp conditions with an MC8 voltage clamp and P2300 Ussing chamber equipment (Physiologic Instruments). Cells grown on culture inserts (Corning) were bathed on both sides with identical Ringer's solutions containing (in mM): 115 NaCl, 25 NaHCO₃, 2.4 KH₂PO₄, 1.24 K₂HPO₄, 1.2 CaCl₂, 1.2 MgCl₂, and 10 D-glucose (pH 7.4). Solutions were aerated with 95% O₂:5% CO₂, and 1-s-long, 3-mV pulses imposed every 10 s to calculate resistance by Ohm's law. As indicated for the particular study, mucosal solutions were changed to a low chloride buffer (1.2 mM NaCl and 115 mM sodium gluconate, with other components as above). Amiloride (100 µM) was added (bilaterally) to block residual sodium current, followed by the CFTR agonist forskolin (10 µM) and CFTR potentiator VX-770 (5 µM). At the end of each experiment, Inh-172 (10 µM, apically) was employed to block CFTR-dependent $I_{sc}$. For analysis of Ussing chamber data, the ACQUIRE & ANALYZE 2.3 package (Physiologic Instruments) was run on Windows environment software to measure current,

voltage, conductance and resistance from 1 to 8 tissues simultaneously. For hNE measurements, a standard setting provided with the equipment software was used with 60 s data acquisition to monitor current changes from baseline.

## Functional assessment of airway surface by micro-optical coherence tomography

For assessment of the functional microanatomic parameters (such as ASL height) of hNE (R1162X/R1162X) transfected with tRT5 or mismatch tRNA, we used micro-optical coherence tomography (µOCT), a high-speed, high-resolution microscopic reflectance imaging approach, as described earlier[65,66]. In brief, this is a non-invasive method, without using exogenous dyes and particles, to image airway epithelia and the associated quantitative analysis, and the µOCT instrument provides cross-sectional images of the cell monolayers at a resolution of approximately 1 µm. Images were acquired 1 mm from the filter periphery with a scanning beam parallel to the tangent of the circumference of the filter membrane disc. Data were acquired at 20,480 Hz line rate, resulting in 40 frames per second at 512 lines per frame. Quantification of the ASL height was performed directly by geometric measurement of the corresponding layers by an investigator blinded to treatment using Image J software. Statistical analysis was performed by two-way ANOVA using Sidak's multiple comparisons.

## Ribosome profiling and data analysis

For ribosome profiling, 2 female wild-type mice were intravenously dosed with LUNAR1 formulated with 0.6 mg kg$^{-1}$ in vitro-transcribed tSA1T5, and 2 female mice were administered intratracheally twice 48 h apart (on day 0 and day 2) with LUNAR2 formulated with 0.35 mg kg$^{-1}$ in vitro-transcribed tSA1T5. Six hours after treatment the mice were anaesthetized with 3% isoflurane in a VetEquip inhalation anaesthesia system chamber. With mice still under anaesthesia inhaled through a nose cone, thoracotomy was performed for dissecting liver and lung tissues, and the tissues were immediately flash frozen. Mice treated with PBS for the same duration served as negative control. The whole organs were pulverized in liquid nitrogen and lysed by grinding in 360 µl 10 mM Tris-HCl (pH 7.4) supplemented with 5 mM MgCl$_2$, 100 mM KCl, 1% NP-40, 2 mM DTT, 100 µg ml$^{-1}$ cycloheximide topped with 40 µl 10% sodium deoxycholate. Lysate from each animal organ were used to produce an independent library.

Twenty-million CFBE41o$^-$ cells were co-transfected with 400 ng CFTR$^{R553X}$ plasmid and 400 ng in vitro-transcribed tRT5 or with 400 ng wild-type CFTR plasmid alone using Lipofectamine 3000 (Thermo Fisher Scientific). Twenty-four hours after transfection, cells were collected and lysed with lysis buffer (10 mM Tris-HCl pH 7.4, 5 mM MgCl$_2$, 100 mM KCl, 1% NP-40, 2 mM DTT).

After lysis, the lysates from mice organs or CFBE41o$^-$ cells were supplemented with cycloheximide (100 µg ml$^{-1}$) to additionally stabilize ribosome–mRNA complexes during RNase I digestion (1.5 µl of 5 U per OD$_{260}$ for 30 min). Sequencing libraries from the RNase I digestion-derived ribosome-protected fragments (RPFs) were prepared using a protocol for micro RNA with direct ligation of the adapters[67].

Sequenced reads were quality selected using the fastx-toolkit (0.0.13.2) with a threshold of 20. Adapter sequences were removed by cutadapt (1.8.3) with a minimal overlap of 1 nt. The libraries were depleted of reads mapping to rRNA reference sequences (bowtie 1.2.2; -y –un) and the reads were mapped to the human (GRCh38) and mouse (GRCm38) reference genomes, respectively. Mapping was performed using STAR[68] (2.5.4b) allowing maximum of one mismatch and filtering out reads mapping to multiple positions (–outFilterMismatchNmax 1 –outFilterMultimapNmax 1). In the reference annotation files, the longest annotated CDS for each transcript was selected. For two transcripts of the same CDS length, we selected the longest transcript including 5′ and 3′ untranslated regions. Uniquely mapped reads were normalized to RPM or RPKM.

To evaluate the stop codon readthrough in the mouse libraries, we used the procedure described in ref. 29. We plotted the middle nucleotide of RPFs or by odd-length reads the nucleotide upstream to the middle in the regions flanking the stop codon, 100 nt upstream and 100 nt downstream of the stop codon. Reads with a length of 25–32 nt were considered. For the CFBE41o$^-$ cell culture libraries, we first calibrated the RPFs to the A-site codon in each RPF following the described procedure along the scripts therein[69]. In these analyses, transcripts with expression higher than 0.1 RPKM were considered.

To select transcripts that have undergone readthrough, we used the ribosome readthrough score (RRTS) described in[26]. RRTS is a ratio of the mean read density over the CDS (reads per kilobases (RPK), normalized to the CDS length) and the mean read density between the natural termination codon and next in-frame stop codon (RPK, normalized to the length of the considered 3′ untranslated region between two stop codons) separated by at least 4 nt from the natural stop codon of the CDSs. The CFTR transcript (ENSEMBL: ENSG00000001626; ENST00000003084.11) coverage is represented as RPM.

## Reporting summary

Further information on research design is available in the Nature Portfolio Reporting Summary linked to this article.

## Data availability

Data from the Ribo-seq and tRNA microarrays were deposited in the Gene Expression Omnibus (GEO) under the accession numbers GSE191048, GSE192623 and GSE205660. Source data are provided with this paper.

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

**Acknowledgements** The authors thank J. van Rheenen for the gift of pLV-CMV-LoxP-DsRed-LoxP-eGFP; R. Trelles for the Cre-*loxP* HEK293T reporter cell line; P. Hartman for construction of some reporter mRNA constructs; H. Valley and M. Mense. for the 16HBEge and primary patient-derived hNE cells; and M. Sablad and T. Dam for technical guidance and assistance regarding mouse experiments. This research was supported by grants from the Cystic Fibrosis Foundation (IGNATO2010 to Z.I. and ROWE19R0 to S.M.R.), NIH (1R01HL136414-01 and 1R01HL136414-05 to Z.I. and E.J.S., and P30DK072482 and R35HL135816 to S.M.R.), the German Cystic Fibrosis Foundation muko e.V (2105 to S.A.), Hamburg Innovation C4T projects (C4T635 to Z.I. and S.A.), and a scholarship from ANII Uruguay (POS-EXT-2020-1-164944 to M.D.). Illustrations were created with BioRender.com.

**Author contributions** S.A., K.T., P.C., D.M. and Z.I. conceptualized the work and strategy. S.A., E.C.A., N.B., M.D., D.J., C.G.P.-G., M.A.D.-T., B.M. and D.M. performed the experiments and analysed the data. L.S. and Z.I. analysed the deep-sequencing data. E.C.A., K.K., K.J.-J.P., J.A.G. and M.A. performed the synthesis of tRNA and mRNA for in vivo formulations and G.A. performed the administrations in mouse. R.M., A.S. and P.K. made the various LUNAR–mRNA and LUNAR–tRNA formulations. S.E.B., G.J.T. and S.M.R. performed the µOCT measurements. J.S.H. produced the FRT cell line, stably expressing *CFTR*[R553X] and with C.M. and D.J. performed the channel activity measurements in primary cells. S.A., E.C.A., E.J.S., D.M., K.T., P.C. and Z.I. planned the experiments. S.A., E.C.A., N.B., M.D., D.J., C.G.P.-G., J.S.H., C.M., E.J.S., G.A., D.M. and Z.I. analysed the experiments and interpreted the data. S.A. and Z.I. supported by N.B., M.D., L.S., D.M. and E.J.S. wrote the manuscript. All authors supported the review of the final version of the manuscript.

**Funding** Open access funding provided by Universität Hamburg.

**Competing interests** Z.I., S.A., N.B. and M.D. are inventors on patents related to tRNA designs for PTC correction. Z.I. is also a scientific advisor for Tevard Biosciences. S.M.R. and G.J.T. are named on an unlicensed patent on the use of OCT for airway surface liquid measurements. S.M.R. is named on an unrelated patent on translational readthrough pharmacotherapy. E.J.S. is a non-voting board member of the Cystic Fibrosis Foundation. The LUNAR technology is proprietary to Arcturus Therapeutics. E.C.A., C.G.P.-G., R.M., B.M., K.K., M.A., K.J.-J.P., G.A., J.A.G., A.S., K.T., P.K., D.M. and P.C. are employees and have securities from Arcturus Therapeutics. The other authors declare no competing interests.

**Additional information**
**Correspondence and requests for materials** should be addressed to Eric J. Sorscher, Pad Chivukula or Zoya Ignatova.

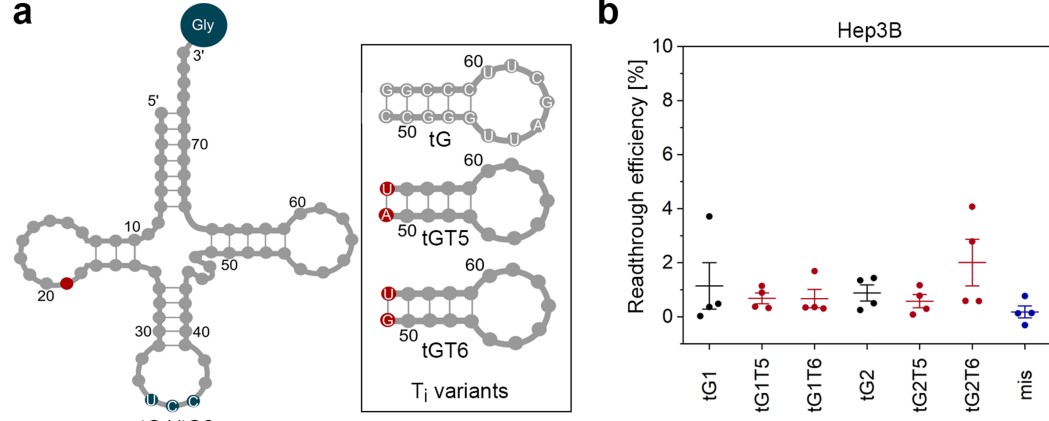

**Extended Data Fig. 1 | tRNA designs to suppress PTCs at Gly codons.**
**a**, Cloverleaf structure (left) of the human tRNA$^{Gly}$UCC which is represented by two isodecoders that share the same anticodon to decode GGA codon, but differ by a single nucleotide (red) in their sequence. The anticodon of both tRNA$^{Gly}$ isodecoders were mutated to pair to the UGA PTC yielding tG1 and tG2 (Supplementary Table 1); nucleotide substitutions in TΨC stem yielding the Ti variants (right) are highlighted in red. **b**, Suppression efficiency of tG variants at UGA PTC monitored with R208X-FLuc in Hep3B cells and normalized to wild-type FLuc expression. In vitro transcribed tG variants were co-transfected with plasmid-borne *FLuc$^{R208C}$* reporter construct (schematic Fig. 1b). Anticodon-edited tG1 and tG2, black; TΨC-stem edited variants of tG1 and tG2, red; mismatch tRNA that does not pair to UGA PTC, dark blue. Data are means ± s.e.m. (n = 4 biologically independent experiments).

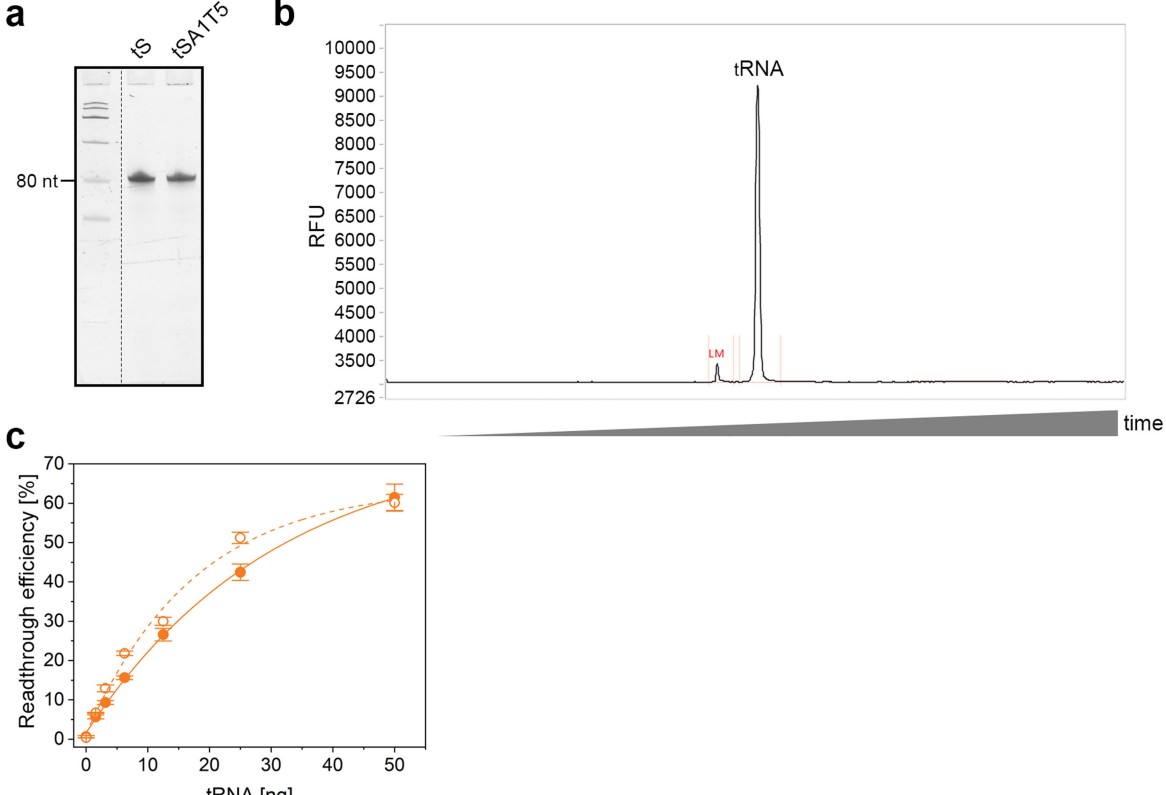

**Extended Data Fig. 2 | In vitro transcription generates high quality functional tRNAs with functionally homogenous 3′ ends in Hepa1-6 cells. a, b**, Purified in vitro transcribed tRNAs are homogenous as monitored by denaturing polyacrylamide gel (**a**) or capillary electrophoresis (**b**) as exemplified by tS and tSA1T5. Left lane on the polyacrylamide gel, RNA ladder (Low Range ssRNA ladder, NEB). LM, lower marker. Analysis performed once with a randomly chosen batch. **c**, Readthrough efficacy of tSA1T5 transcribed with conventional primer (closed circles) is comparable to that of the tSA1T5 produced using 3′-modified (2′-OMe) primer (open circles) which enables precise termination of in vitro tRNA transcription. Data are means ± e.e.m. (n = 3 biologically independent experiments). For gel source data to panel a see Supplementary Fig. 2.

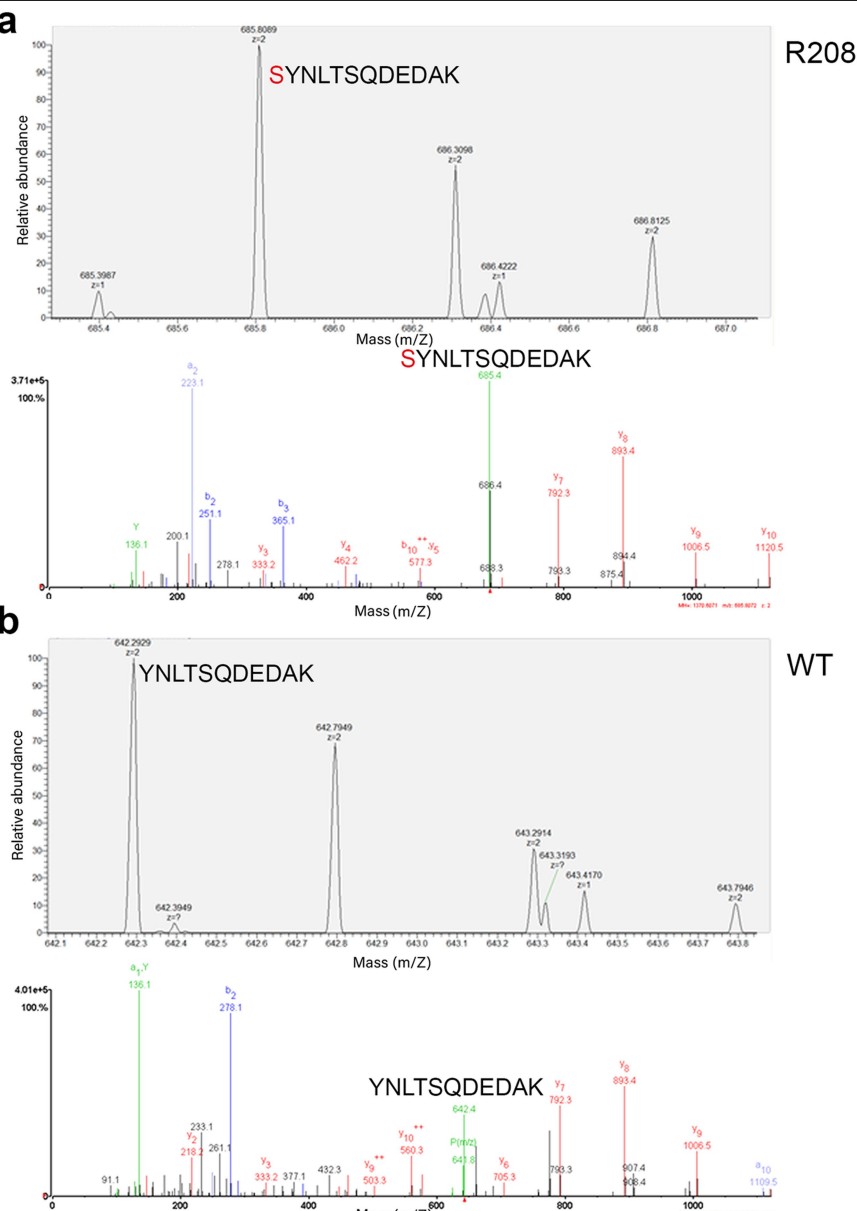

**Extended Data Fig. 3 | tSA1T5 incorporates only Ser in place of the UGA stop codon. a, b**, Hepa 1-6 cells co-transfected with in vitro transcribed tSA1T5 and *FLuc^R208X-RLuc* mRNA (**a**) or WT *FLuc^R208-RLuc* mRNA (**b**) and analyzed by 2D-nano PRM LC-MS/MS. Extracted Ion Chromatograms (EICs) (**a,b** upper panels) and MS/MS spectra (**a,b** lower panels). Peptides with m/z 685.8089 and 642.2929 correspond to the double-charged Luc-derived peptides. Peptide **S**YNLTSQDEDAK is identified in R208X-FLuc expressing cells transfected with tSA1T5, indicating the incorporation of Ser at the UGA stop codon. Sequence parts of tSA1T5 originate from tRNA^Thr GCU, yet they do not compromise the fidelity of seryl-tRNA-synthetase, as no threonine incorporation was detected (Supplementary Table 1). No tryptophan or selenocysteine was detected either despite the intrinsic tendency of tRNA^Trp or tRNA^Sec to pair to UGA codon. Wild-type FLuc-derived peptides from the FLuc-R208-RLuc reporter without PTC were cleaved directly after the Arg208 residue generating the peptide YNLTSQDEDAK. The experiment serves as a quality check and hence, was performed as a single experiment.

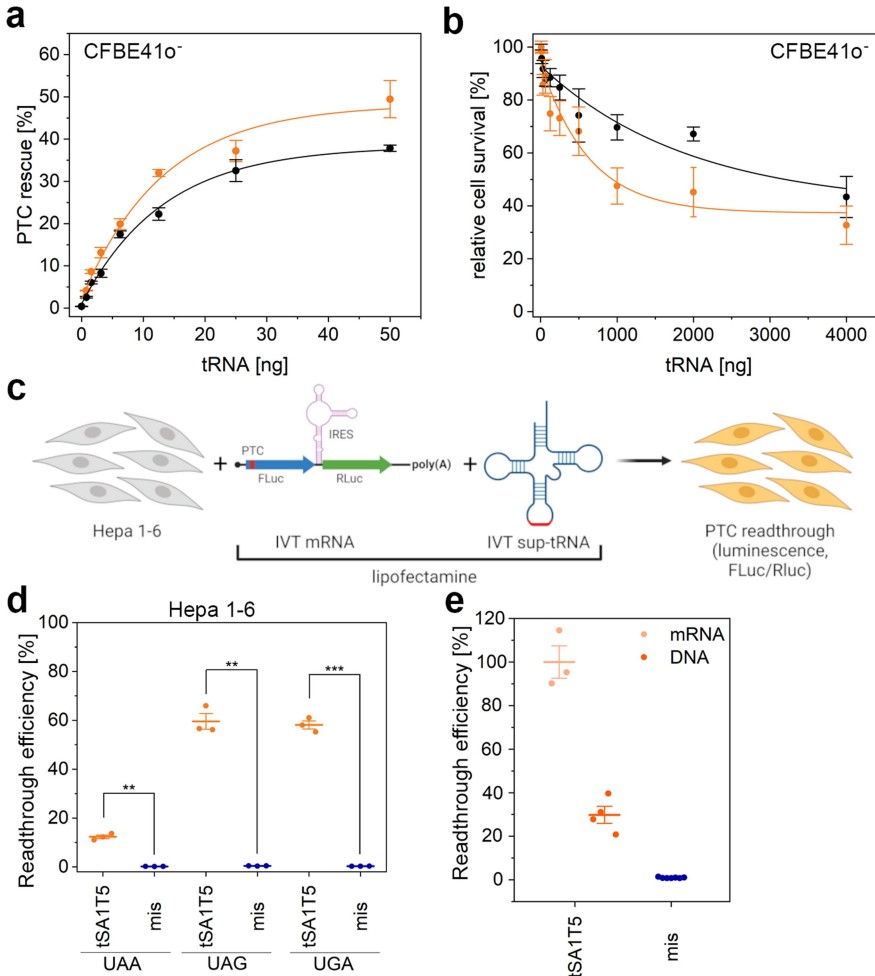

**Extended Data Fig. 4 | Dose-dependent suppression activity of sup-tRNA and cell viability. a**, Suppression activity monitored by co-transfecting different concentrations of in vitro transcribed tS (black) or tSA1T5 (orange) suppressor tRNAs . PTC suppression activity is represented as a ratio of the normalized PTC-FLuc reporter activity from *FLuc-R208X-RLuc* mRNA to that of the reporter without PTC (FLuc(R208S)-RLuc). Data are means ± s.e.m. (n = 3 biologically independent experiments). **b**, Cell viability assay 24 h post transfection of CFBE41o⁻ cells transfected with different tS (black) and tSA1T5 (orange) concentrations. The signal was normalized to mock-transfected cells and the signal with the lowest tRNA concentration was set as 100%. Data are means ± s.e.m (n = 3 biologically independent experiments). **c**, Schematic of screening by co-transfecting of in vitro transcribed (IVT) sup-tRNA and PTC-*FLuc-RLuc* reporter mRNA. FLuc signal was normalized on the RLuc,

both of which encoded within the same reporter but expressed from independent promoters. *RLuc* uses cap-independent IRES . **d**, tSA1T5 targets PTCs with different stop codon identities tested in murine Hepa 1-6 cells in the setup introduced in panel c. PTC suppression activity was calculated using the ratio of the normalized PTC-FLuc reporter activity from FLuc(R208X)-RLuc mRNA to that of a reporter without PTC (FLuc-R208S-RLuc), and represented as % readthrough efficiency at the PTC. Data are means ± s.e.m. (n = 3 biologically independent experiments). **P < 0.01, ***P < 0.001 (one-sided *t* test). **e**, Comparison of PTC suppression efficiency by co-transfecting tSA1T5 (orange) with the PTC (UGA) reporter as either mRNA (**c**), or DNA (schematic, Fig. 1b). mis (dark blue) designates mismatch tRNA that does not pair to any PTC. Data are means ± s.e.m. (for mRNA – n = 3, for DNA – n = 4, for mis combined n = 7 biologically independent experiments).

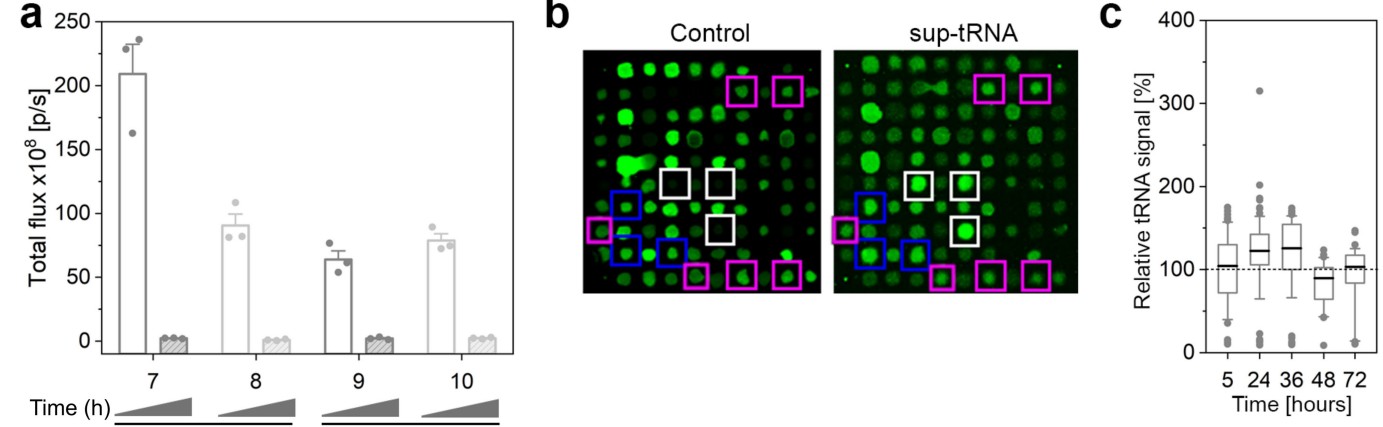

**Extended Data Fig. 5 | Stability of the administered mRNA and sup-tRNA.**
**a**, The amount of the control $aLuc^{R208S}$ mRNA inferred by aLuc activity in mice decreased by two orders of magnitude within 24 h. Quantification of aLuc activity expressed from $aLuc^{R208S}$ mRNA post i.v. injection from the IVIS images of the Balb/C mice (Fig. 2b) at 6 h (open bars) or 24 h (filled bars). $aLuc^{R208S}$ mRNA was co-encapsulated with tS (groups 7, 8) or tSA1T5 (groups 9, 10) at 0.6 mg/kg sup-tRNA (group 7, 9) or 1.2 mg/kg sup-tRNA (group 8, 10). Data are means ± s.e.m. (n = 3 mice in each group). **b**, Representative block from the tRNA-tailored microarray to detect sup-tRNA with high sensitivity. Full-length tDNA probes complementary to tSA2T5 were used to detect sup-tRNA (white squares). Each array consists of 12 such blocks, each of which contains a duplicate of three distinct probes recognizing all tRNA$^{Ser}$ isoacceptors, i.e. tRNA$^{Ser}$(wGA), a combined probe for tRNA$^{Ser}$AGA and tRNA$^{Ser}$UGA (recognizing UCU, UCC and UCA codons; both probes at the bottom right), tRNA$^{Ser}$GCU

(pairing to AGU and AGC codons; probes - upper right), and tRNA$^{Ser}$CGA (reading codon UCG, bottom left flanking the spike-ins). Blue squares, spike-ins or non-human tRNAs, used for signal normalization. tSA2T5 share some sequence similarity with tRNA$^{Ser}$AGA and tRNA$^{Ser}$UGA whose backbone was the starting sequence for tSA2T5 (Supplementary Table 1). In the control group, untreated with sup-tRNA, none of the natural tRNA$^{Ser}$ isoacceptors (purple squares) hybridize to tSA2T5 probe suggesting high specific of detection.
**c**, sup-tRNA stability in gene-edited 16HBEge cells (16HBEge$^{R1I62X/-}$) monitored with tRNA microarrays represented as box plot (n = 2 independent replicates) and normalized to the mean of the signal at first time point (5 h) post transfection which was set as 100 % (horizontal line). The center line indicates the median, box ends are 10%-90%, and the whiskers – the range of the remaining data without exclusion of outliers.

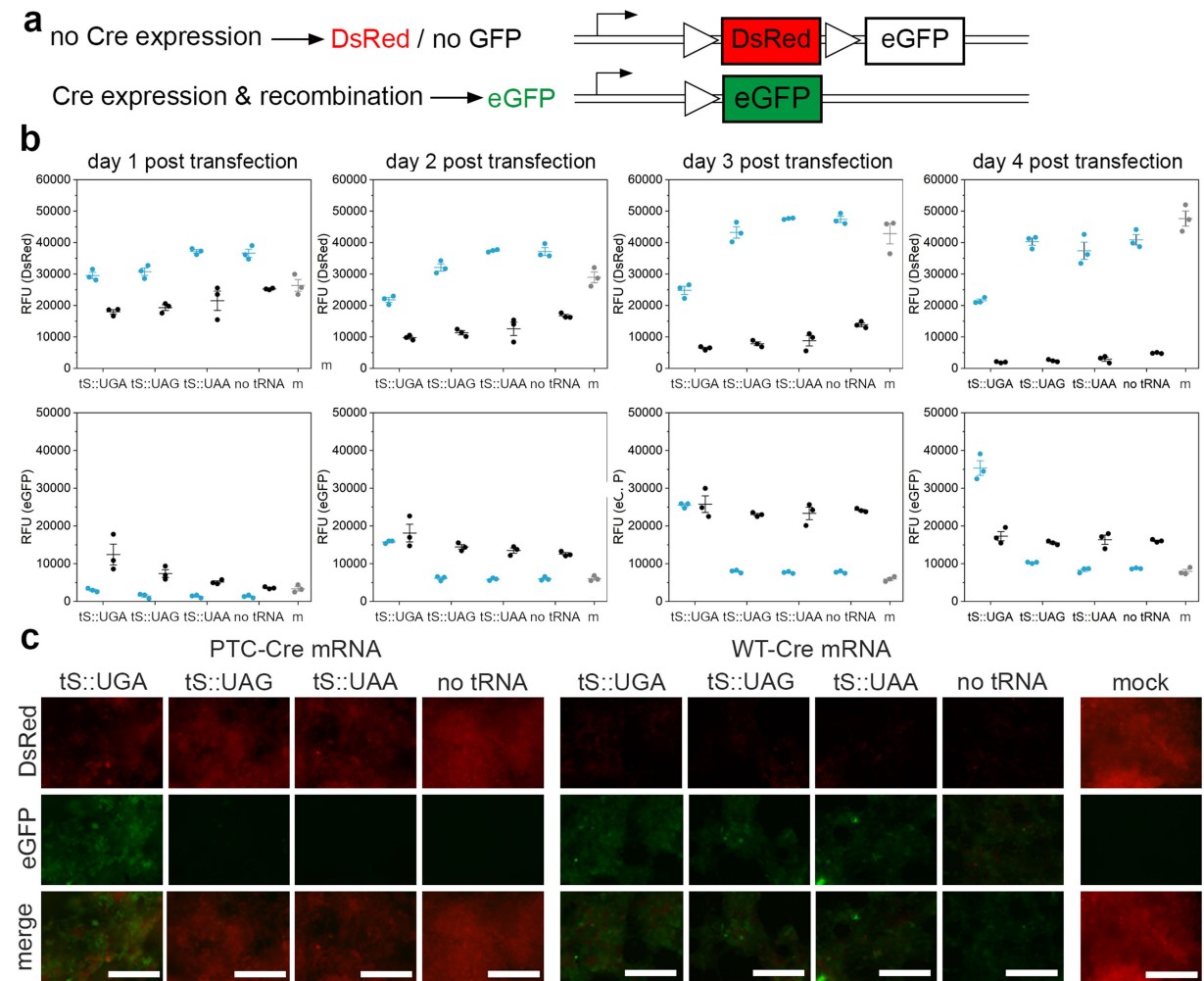

**Extended Data Fig. 6 | Cre-lox system is suitable to monitor PTC suppression. a**, Schematic of the Cre-lox system. eGFP will be only expressed if full-length Cre recombinase is expressed, which through subsequent recombination removes DsRed gene from the original cassette. Each fluorescent protein is color coded and captures the corresponding fluorescence pattern on the images. Arrow and triangle designate the CMV and *loxP* promoters, respectively. **b, c** Time course of DsRed and eGFP expression HEK293T cells stably expressing the floxed-DsRed-floxed-eGFP cassette and transfected with *Cre* mRNA and tRNA variants. Integrated fluorescence (**b**) and representative microscopy images at day 4 post transfection (**c**). A PTC-*Cre* mRNA with two UGA PTCs (S69X-S82X; blue) was used for co-transfection with tS variants designed to suppress UGA, UAA or UAG codons, respectively. Wild-type *Cre* mRNA without PTC (black) and mock (m) transfected cells (grey) served as positive and negative control of Cre-*lox* recombination, respectively. Notably, we observed elevation of eGFP signals along with simultaneous reduction of DsRed only in cells co-transfected with PTC-*Cre* mRNAs and PTC-matching tRNA (tS::UGA) suggesting efficient PTC correction and expression of full-length Cre recombinase. The eGFP signal decreased at day 4, likely because of degradation and/or decreased protein expression in general, since the DsRed signal also decreased. Data are means ± s.e.m. (n = 3 biologically independent experiments). Scale bar, 200 µM.

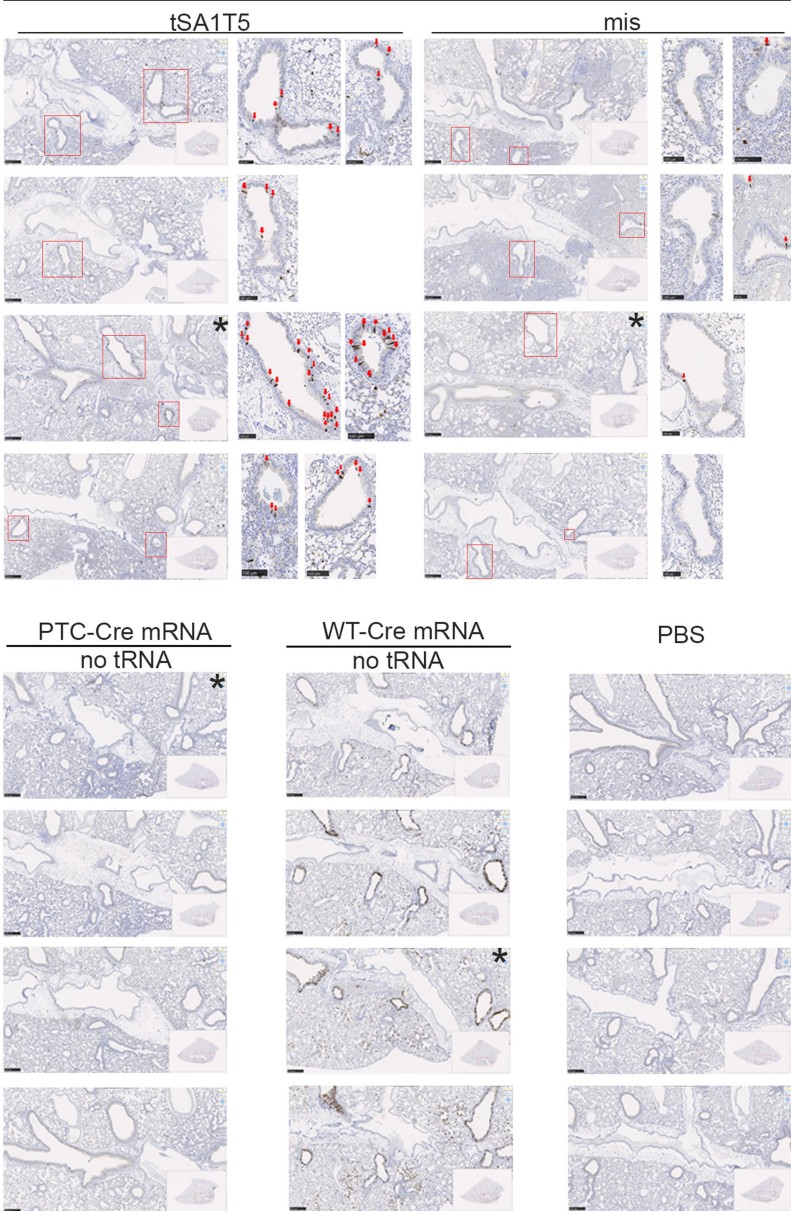

**Extended Data Fig. 7 | In vivo PTC suppression in lung by LUNAR-formulated tSA1T5 in a transgenic Flox-tdTomato mouse model.** Brightfield immunohistochemical images for tdTomato expression in lungs from mice (n = 4 mice in each group) intratracheally dosed with LUNAR2-LNPs that were formulated with WT-*Cre* mRNA (no PTC), PTC-*Cre* mRNA with UGA at S69X-S82X with or without tSA1T5 with an anticodon pairing to UGA or a mismatch tRNA (mis). Control mice with PTC-*Cre* mRNA administered only were used to establish any background tdTomato that might be present as reported by the transgenic line producer (Jackson Laboratories, USA). Mouse lung treated with PBS served as a negative control. Without a suppressor tRNA, PTC-*Cre*^S69X-S82X mRNA did not result in tdTomato expression and is indistinguishable from the negative control (PBS). tdTomato expression is detectable as dark brown spots and seen only in the WT-*Cre* mRNA or following suppression with tSA1T5. Representative image from each mouse of each group is included. Square in the whole image (bottom right) points location of each panel. Scale bar, 250 μM. Successful PTC suppression is indicated by the higher numbers of cells expressing tdTomato (red arrow on the magnified images, scale bar, 100 μM). Very low background of tdTomato expression from mice dosed with PTC-Cre^S69X-S82X mRNA and non-matching mismatch tRNA (mis) was observed (red arrow on the magnified images, scale bar, 100 μM), likely due to a low Cre-independent tdTomato background reported for these transgenic mice. Images designated with * in the right corner are the ones shown in Fig. 2d–g.

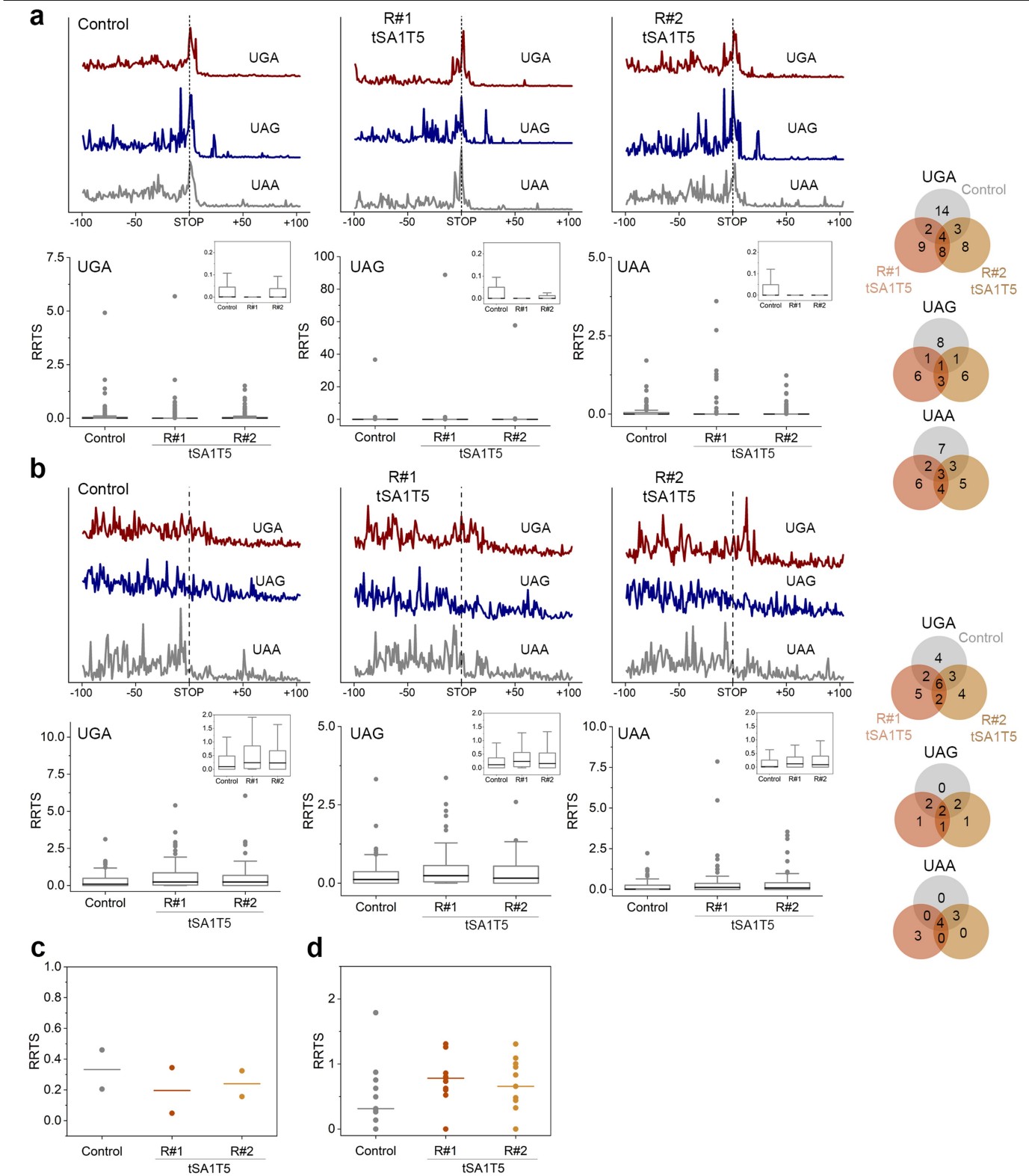

**Extended Data Fig. 8** | See next page for caption.

**Extended Data Fig. 8 | Suppressor tRNAs do not induce readthrough at native UGA codons in mice and cell culture. a-b**, Marginal but comparable readthrough at all three natural stop codons (including the unspecific ones UAA and UAG not targeted by the sup-tRNA) detected in the cumulative coverage plots (upper panels), the ribosome readthrough score (RRTS, lower panels) and Venn diagrams of the top 10% transcripts with the highest RRTS (on the right) in lung tissue of mice treated by i.t. microsprayer instillation with nanoparticulated tSA1T5 (**a**), liver tissue from mice treated by i.v. with nanoparticulated tSA1T5 (**b**). The center line indicates the median, the box ends – the first and third quartiles, and the whiskers – the range of the remaining data without exclusion of outliers. (**b**). Insets, box-plots as in the main plot without the outliers to visualize the median values. Control, mice treated with PBS; R#1 and R#2 indicate two mice used as independent replicates. The RRTS is a comparison between tRNA-treated samples compared to the corresponding tissue of a control mice (**a,b**). We noted that all detected transcript with a high RRTS (Venn diagrams) are within 10%ile of genes with highest expression level, suggesting a high proportion of false positives. **c,d**, Suppressor tRNA do not enhance the readthrough at internal UGA stop codons. RRTS in lung tissue of mice treated by i.t. with nanoparticulated tSA1T5 (**c**) and liver tissue from mice treated by i.v. with nanoparticulated tSA1T5 (**d**). The RRTS is a comparison between tRNA-treated samples compared to the corresponding tissue of a control mice (**c,d**). Note that different number of genes with internal UGA stop codons were detected as expressed in different tissues (i.e. 11 internal stop codons in mouse liver, 2 in lung tissue).

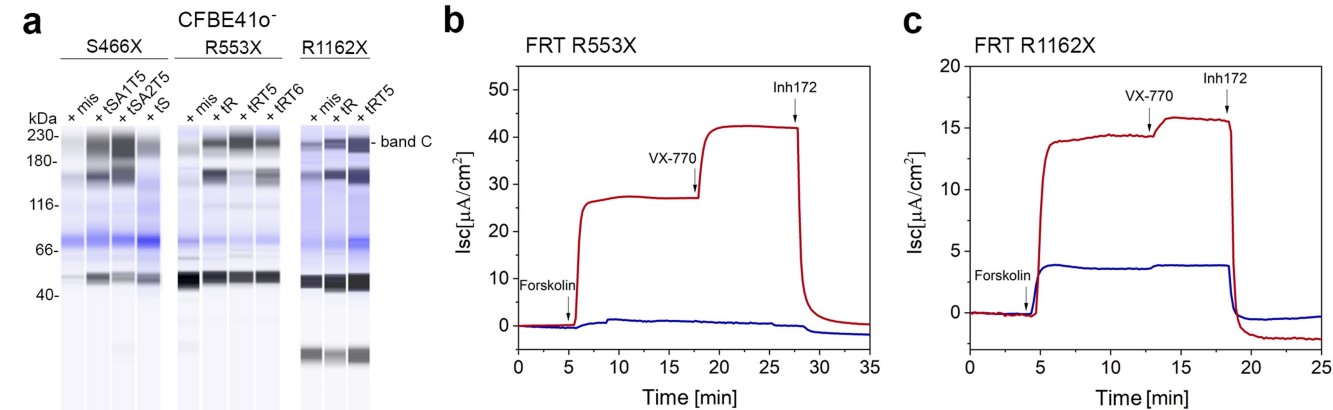

**Extended Data Fig. 9 | Efficiency of tS and tR variants in suppressing PTCs in CFTR-expressing immortalized cell lines. a,b,** Suppression activity in full-length CFTR was monitored by automated immunoblotting (JESS system) using monoclonal CFTR-NBD2 antibody (1:100 dilution, #596) in CFBE41o⁻ cells co-transfected with plasmid-encoded *CFTR^{S466X}*, *CFTR^{R553X}*, *CFTR^{R1162X}* and the corresponding in vitro transcribed tRNA variant, using lipofectamine as transfection reagent. The immunofluorescence signal of the mature fully glycosylated CFTR (band C) was normalized against signal from the total protein staining (blue). mis, PTC-CFTR variants co-transfected with mismatch tRNA which does not pair to the PTC. Lower molecular weight bands are likely CFTR degradation or incompletely synthesized products. **b, c,** Representative Ussing chamber tracings obtained from FRT^{R553X} (**b**) and FRT^{R1162X} (**c**) cells transfected with tRT5 (red) or mismatch tRNA (blue) using lipofectamine. Forskolin, CFTR activator; VX-770, potentiator, Inh-172, CFTR inhibitor. Statistically insignificant (minimally negative) VX-770 deflections were noted in some tracings after peak CFTR stimulation by forskolin. Gel source data to panel a included in Supplementary Fig.1.

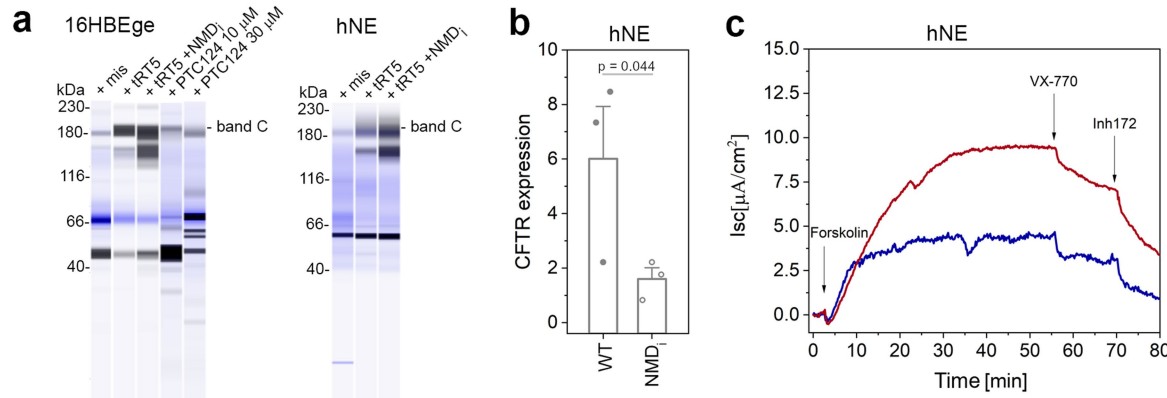

**Extended Data Fig. 10 | Restoration of CFTR expression and activity by tRT5 or by co-treatment with NMD inhibitor in patient-derived hNE (R1162X/R1162X) cells. a**, Representative immunoblots (automated immunoblotting, JESS system) using monoclonal CFTR-NBD2 antibody (1:100 dilution, #596) in 16HBEge$^{R1162X/-}$ and hNE$^{R1162X/R1162X}$ transfected with sup-tRNA alone using lipofectamine or treated with 5μM NMD14. PTC124 (ataluren). **b**, Wild-type (WT) CFTR expression (band C normalized to total protein) in hNE derived from healthy individuals with and without treatment with NMD$_i$, (NMD14 (5μM)) for 24h. Data are means ± s.e.m. (n = 3 biologically independent replicates). Statistics, one-sided t test. NMD alone decreases the WT-CFTR expression most likely owing to alterations of hNE viability following NMD treatment. **c**, Representative short-circuit current traces obtained from hNE$^{R1162X/R1162X}$ cells transfected with tRT5 (red) or mismatch tRNA (blue). Forskolin, CFTR activator; VX-770, potentiator, Inh-172, CFTR inhibitor. Gel source data to panel a included in Supplementary Fig. 1.

# Reporting Summary

## Statistics

For all statistical analyses, confirm that the following items are present in the figure legend, table legend, main text, or Methods section.

| n/a | Confirmed | |
|---|---|---|
| ☐ | ☒ | The exact sample size (*n*) for each experimental group/condition, given as a discrete number and unit of measurement |
| ☐ | ☒ | A statement on whether measurements were taken from distinct samples or whether the same sample was measured repeatedly |
| ☐ | ☒ | The statistical test(s) used AND whether they are one- or two-sided *Only common tests should be described solely by name; describe more complex techniques in the Methods section.* |
| ☒ | ☐ | A description of all covariates tested |
| ☐ | ☒ | A description of any assumptions or corrections, such as tests of normality and adjustment for multiple comparisons |
| ☐ | ☒ | A full description of the statistical parameters including central tendency (e.g. means) or other basic estimates (e.g. regression coefficient) AND variation (e.g. standard deviation) or associated estimates of uncertainty (e.g. confidence intervals) |
| ☐ | ☒ | For null hypothesis testing, the test statistic (e.g. *F*, *t*, *r*) with confidence intervals, effect sizes, degrees of freedom and *P* value noted *Give P values as exact values whenever suitable.* |
| ☒ | ☐ | For Bayesian analysis, information on the choice of priors and Markov chain Monte Carlo settings |
| ☒ | ☐ | For hierarchical and complex designs, identification of the appropriate level for tests and full reporting of outcomes |
| ☒ | ☐ | Estimates of effect sizes (e.g. Cohen's *d*, Pearson's *r*), indicating how they were calculated |

*Our web collection on statistics for biologists contains articles on many of the points above.*

## Software and code

Policy information about availability of computer code

| Data collection | Bulk activity (luciferase/renilla) measurements: Spark microplate reader, SparkControl Version 2.3 (Tecan). |
|---|---|
| | Mass spectrometry: 2D nano PRM LC-MS/MS by Jade Bio, Inc (San Diego, CA) |
| | qRT-PCR: T professional thermocycler (Biometra) |
| | Immunoblots: Jess capillary electrophoresis system (Jess, ProteinSimple) |
| | Cell fluorescence imaging: EVOS Cell Imaging Systems (Thermo Fisher Scientific) |
| | Live animal bioluminescence imaging: IVIS Lumina III (Perkin Elmer, USA) |
| | Ussing chamber for CFTR channel activity measurements (Psysiologic Instruments, Inc.) |

| Data analysis | Immunoblot quantification: Compass for SW Version 6.0.0 (Jess, ProteinSimple) |
|---|---|
| | Live animal bioluminescence imaging quantification: Living Image Software, version 4.7.4. (Perkin Elmer) |
| | CFTR channel activity - ACQUIRE & ANALYZE v.2.3 package provided by Ussing chamber vendor (Physiologic Instruments, Inc.) and measures current, voltage, conductance and rsistance from up to 8 tissues simultaneously. |
| | µOCT measuremnts - Image J (version 1.49) |

For manuscripts utilizing custom algorithms or software that are central to the research but not yet described in published literature, software must be made available to editors and reviewers. We strongly encourage code deposition in a community repository (e.g. GitHub). See the Nature Portfolio guidelines for submitting code & software for further information.

## Data

Policy information about availability of data

All manuscripts must include a data availability statement. This statement should provide the following information, where applicable:

- Accession codes, unique identifiers, or web links for publicly available datasets
- A description of any restrictions on data availability
- For clinical datasets or third party data, please ensure that the statement adheres to our policy

All data are available in the main text or the supplementary materials.
Ribosome profiling data (Ribo-seq) from mouse organs and human CFBE41o- cells and tRNA microarray generated in this study are available at the Gene Omnibus (GEO) under the accession numbers GSE191048, GSE 192623 and GSE205660, respectively. For mapping of the Ribo-seq data sets human (GRCh38) and mouse (GRCm38) reference genomeswere used, respectively.

# Field-specific reporting

Please select the one below that is the best fit for your research. If you are not sure, read the appropriate sections before making your selection.

☒ Life sciences      ☐ Behavioural & social sciences      ☐ Ecological, evolutionary & environmental sciences

For a reference copy of the document with all sections, see nature.com/documents/nr-reporting-summary-flat.pdf

# Life sciences study design

All studies must disclose on these points even when the disclosure is negative.

| | |
|---|---|
| Sample size | Sample size is stated in figure legends. For readthrough efficiency, 3 mice were used and for Ribo-seq 2 mice. Experiments were designed to detect differences greater than 20% at a significance of p<0.05. |
| Data exclusions | No data were excluded. |
| Replication | Experiments were reproduced in multiple (2-10) independent biological replicates each time using independent experimental methods. In each figure legends the number of independent biological replicates is stated. The majority of the assays wered cunducted three times, which demonstrates the high reproducibility of the data. |
| Randomization | Animals were chosen randomly and assigned either as control group or for treatment. For immunofluorescence and immunohistochemistry, the fields of view were randomly chosen. For other experiments, where the samples were not randomly assigned, there was no need for randomization, because all samples were independently measured in a controlled manner |
| Blinding | For the majority of the experiments no blinding was performed as blinding is not possible or not applicable for the experiments; only in the µOCT assays the geometric measurement of the corresponding layers was perfomed by an investigator blinded to the treatment. The in vivo reserachers were not blinded, as they performed the treatments. In cell culture experiments, blinding is not applicable because the researchers need to verify samples and controls for each experiment.<br>However, in almost all experiemnts, a second researcher confirmed the results. |

# Reporting for specific materials, systems and methods

We require information from authors about some types of materials, experimental systems and methods used in many studies. Here, indicate whether each material, system or method listed is relevant to your study. If you are not sure if a list item applies to your research, read the appropriate section before selecting a response.

### Materials & experimental systems

| n/a | Involved in the study |
|---|---|
| ☐ | ☒ Antibodies |
| ☐ | ☒ Eukaryotic cell lines |
| ☒ | ☐ Palaeontology and archaeology |
| ☐ | ☒ Animals and other organisms |
| ☒ | ☐ Human research participants |
| ☒ | ☐ Clinical data |
| ☒ | ☐ Dual use research of concern |

### Methods

| n/a | Involved in the study |
|---|---|
| ☒ | ☐ ChIP-seq |
| ☒ | ☐ Flow cytometry |
| ☒ | ☐ MRI-based neuroimaging |

## Antibodies

| | |
|---|---|
| Antibodies used | 1) primary rabbit polyclonal RFP antibody (#600-401-379, Lot #42896, dilution 1:800, Rockland Antibody) |

| Antibodies used | 2) mouse monoclonal FoxJ1 (#14-9965-82, Lot #2262272, dilution 1:100, ThermoFisher Scientific)<br>3) rabbit polyclonal Muc5B (#PA5-82342, Lot # VL31536898, dilution 1:600, ThermoFisher Scientific)<br>4) anti CFTR-NBD2 antibody (#596, Lot #596TJ03182012, 1:100 dilution, John R. Riordan and Tim Jensen, University of North Carolina, Chapel Hill, USA)<br>5) Donkey anti mouse AF488 (#A-21202, dilution 1:1000 ThermoFisher Scientific).<br>6) Donkey anti rabbit AF555 (#A-21428, dilution 1:1000, ThermoFisher Scientific) |
|---|---|
| Validation | For Jess-Immunoblot experiments, we tested different dilutions in the range 1:10-1:250 which is recommended by the manufacturer. The optimal dilution was 1:100.<br><br>The primary rabbit polyclonal RFP antibody (#600-401-379, Rockland Antibody) was internally validated for immunohistochemistry on transgenic models expressing TdTomato protein and also by the vendor. Used in Fig. 2 and Extended Data Fig. 7. Antibody profile webpage: https://rockland-inc.com/store/Antibodies-to-GFP-and-Antibodies-to-RFP-600-401-379-O4L_24299.aspx<br><br>For mouse monoclonal FoxJ1 (#14-9965-82, ThermoFisher Scientific) and rabbit polyclonal Muc5B (#PA5-82342, ThermoFisher Scientific) the vendors provided validation reports for tests in different organisms. The suitability of the rabbit polyclonal Muc5B for immunohistochemistry on mouse tissues was tested internally. Both antibodies were used in Fig. 2. Vendors antibody profile webpages: Muc5B - i. https://www.thermofisher.com/antibody/product/MUC5B-Antibody-Polyclonal/PA5-82342, and FoxJ1 - i. https://www.thermofisher.com/antibody/product/FOXJ1-Antibody-clone-2A5-Monoclonal/14-9965-82<br><br>Donkey anti-mouse AF488: https://www.thermofisher.com/antibody/product/Donkey-anti-Mouse-IgG-H-L-Secondary-Antibody-Polyclonal/R37114<br><br>Donkey anti rabbit AF555: https://www.thermofisher.com/antibody/product/Donkey-anti-Rabbit-IgG-H-L-Highly-Cross-Adsorbed-Secondary-Antibody-Polyclonal/A-31572 |

# Eukaryotic cell lines

Policy information about cell lines

| Cell line source(s) | Human Hep3B (HB-8064) and murine Hepa 1-6 (CRL-1830) were obained form ATCC/DSM.<br>HEK293XL TLR7 and TLR8 cells from Invivogen.<br>Cre-lox HEK293T (CRL-3216) reporter cell line was received from Dr. Ramon Trelles (Assoc. Director of Drug Discovery at Arcturus Therpaeutics).<br>CFBE41o- laboratory collection; originally received from Dr. Karl Kunzelmann, University of Regensburg, Germany) and Dr. Dieter Gruenert (University of California, San Francisco, CA).<br>Stably transfected R553X-CFTR and R1162X-CFTR FRT cell lines were created by Jeong S. Hong, a co-author of this study.<br>16HBEge cells ( R1162X/-): received from the Cystic Fibrosis Foundation Therapeutics Lab.<br>16HBE14o- WT-CFTR obtained from Dr. Michael Lalk (Univ. Greifswald, Germany)<br>Primary human nasal epithelial cells (hNE R1162X/R1162X): received from the Cystic Fibrosis Foundation Therapeutics Lab.<br>Primary human nasal epithelial cells fron non-CF individual expressing wildtype CFTR: received from the Cystic Fibrosis Foundation Therapeutics Lab. |
|---|---|
| Authentication | For the purchased cell lines the vendor provided the authentication protocol and no additional authentification was used.<br>CFBE41o- was tested for the deletion of WT-CFTR (by sequencing and immunoblot).<br>The presence of R553X mutation in CFTR and the CFTR integration were verified by sequencing. |
| Mycoplasma contamination | Cell lines directly purchased from ATCC/DSM for the study arrived with a certificate for mycoplasma test and were not tested at arrival. All other cells were tested at arrival for mycoplasma contamination using Venor GeM PCR-based detection kit (Merck) and the tests were negative. Tested aliquots at low passage number were stored in liquid nitrogen. Cell lines in culture are regularly (every 6 mo. or by indication) tested for mycoplasma contamination and they all tested negative. |
| Commonly misidentified lines<br>(See ICLAC register) | No commonly misidentified cell lines were used. |

# Animals and other organisms

Policy information about studies involving animals; ARRIVE guidelines recommended for reporting animal research

| Laboratory animals | 10-weeks old female Balb/C mice were purchased from Charles River Laboratories and 7-8 weeks old female B6.Cg-Gt(ROSA)26Sortm14(CAG-tdTomato)Hze/J were purchased from Jackson Laboratories. All mice were purpose bred and experimentally naïve at the start of the study. Mice were housed 5 per cage in a pathogen-free environment in Innovive disposable IVC rodent caging system with a 12h light/dark cycle, at temperature between 19-22°C and humidity 50-60%. Ad libitum access to standard diet (2018, Global 18% protein rodent diet from Envigo+++, San Diego, CA, USA) and pre-filled acidified water from Innovive (pH 2.5-3.0) were used throughout the study period. The bedding material was hardwood chips (Sani-Chips, Cat# 7115, Envigo++++, CA, USA) and cages were changed biweekly. |
|---|---|
| Wild animals | No wild animals were used in this study. |
| Field-collected samples | No field collected samples were used in this study. |
| Ethics oversight | All studies were performed at Arcturus Therapeutics in accordance with the animal use protocols and policies approved by the Institutional Animal Care and Use Committee (IACUC) by an AAALAC approved vendor Explora BioLabs. The IACUC issuing organization is Explora BioLabs, protocol number EB17-004-003 from 2/1/17 and latest amendment from 6/17/21. |

Note that full information on the approval of the study protocol must also be provided in the manuscript.

