## [Peer Review File · Nature]

Manuscript Title: Engineered tRNAs suppress nonsense mutations in cells and in vivo

Reviewer Comments & Author Rebuttals

Reviewer Reports on the Initial Version:

Referee #1 (Remarks to the Author):

Albers et al.,

The authors have shown that they can efficiently suppress PTCs using suppressor tRNAs, and the manuscript certainly has merit (aside from the many issues raised because the writing has uncertain meaning).

That noted, as the reader becomes aware as early as line 45 with the sentence “A targeted strategy to correct nonsense mutations and alleviate translation...”, the authors are not considering that most PTCs trigger nonsense-mediated mRNA decay (NMD). This sentence would more accurately read “A targeted strategy to correct nonsense mutations and alleviate the premature termination of translation and the resulting nonsense-mediated mRNA decay...”. It is one accomplishment to suppress a PTC so that it encodes an amino acid, which the authors have done. However, since most PTCs trigger NMD once recognized by the translating ribosome, a second requisite accomplishment is for suppression to be sufficiently efficient during the time that NMD occurs. The authors cannot assess this efficiency since their reporter genes/mRNAs lack a PTC-distal intron/EJC and, thus, are immune to NMD.

The authors are set to assess the effect of tRNA-mediated suppression on NMD. To do this, they must test cells and mice expressing a full-length PTC-containing disease gene and compare the level of full-length protein deriving from this gene to that deriving from its normal (i.e. PTC-lacking) counterpart. Otherwise, the utility of their approach is not fully realized as a therapeutic.

Major Comments

1) The largest criticism of this study is the lack of using PTC-containing target transcripts that exhibit nonsense mediated decay (NMD). A significant part of the mechanism of nonsense-associated diseases that makes them so severe is the loss of target gene transcript expression through NMD. Indeed, the major advancement of this study is the delivery of sup-tRNAs as RNA using LUNAR2 LNP formulation to cultured cells, lung and liver. However, suppression of PTCs within the mRNAs chosen, i.e., those deriving from FLuc, Cre and CFTR genes that do not harbor at least one intron downstream of the PTC, does not provide insight into the ability of sup-tRNAs to inhibit NMD. Determining the ability of sup-tRNAs to suppress PTCs encoded in endogenous transcripts that undergo proper post-transcriptional processing is an important aspect not addressed in this study.

2) One of the significant issues/concerns with use of sup-tRNA RNAs as therapeutics is their relatively short half-lives (PMID: 33567469) due to their degradation. All of the experiments in this study were performed over a short time-frame. The time-course of sup-tRNA activity should be

determined in vivo and in cell culture as this provides crucial insight into the therapeutic promise of sup-tRNAs.

3) It should be stated how the original tRNA sequences that were modified in this study were identified. Examining the tRNAscan-SE Genomic tRNA Database (Cite: PMID: 26673694 & 18984615), there appears to be many tRNA sequences one could choose from to generate active sup-tRNAs. Furthermore, a previously published study (cite: PMID: 30778053) generated a library of sup-tRNAs from all unique tRNA sequences (23 serine, 29 Arg, 20 Gly) and tested their suppression activity to define those with the most activity. While it appears that the authors were able to identify active sup-tRNA sequences for each tRNA family tested, determining how these sequences were chosen from the many isoacceptors would be helpful to the reader. Notably, the GtRNAdb Gene Symbol and/or tRNAscan-SE ID should be used for each tRNA sequence listed.

4) It is unclear why the specific anticodon stem and t-stem were chosen. While there have been several t-stem (example PMID: 21402928) and anticodon-stem mutations (example PMID: 2428035) that have been made to enhance sup-tRNA activity, relatively few variants were tested here. It would be beneficial to the reader to understand what methods were used to predict how specific variants were chosen to test.

5) Comparisons of sup-tRNA variant PTC rescue are shown in Figure 4. These should be compared to the original sup-tRNA sequences to show the benefit of variants over original sequences. Furthermore, a dose-dependence of the original sup-tRNA should be included in Extended Data Figure 3a and b, for comparison to the tSA1T5 variant. It would be expected to have a rightward shift of PTC suppression efficiency and toxicity.

6) The mass spec performed to ensure that the correct amino-acid is incorporated is insufficient. These experiments do not report on the percent incorporation of serine, only that it was incorporated. The absolute fidelity of PTC suppression should be determined. An important aspect of sup-tRNAs is that they predominately incorporate the correct amino acid (cite: PMID: 30778053) and not the near-cognate as a result of small molecule-dependent readthrough (PMID: 27702906). However, modifications to the anticodon stem and t-stem could impart reduced aminoacyl-tRNA synthetase fidelity. Comparing the amino acid incorporation fidelity of the original and variant sup-tRNAs is important for the therapeutic promise of the presented sup-tRNA variants.

7) It is peculiar that only two experiments are reported for WB results in Fig 4b. This makes it impossible to perform statistics.

8) Representative short-circuit currents should be included in the supplemental data for Fig. 4c and WT CFTR currents should be presented for PTC suppression efficiency comparison. As with Fig 4a and 4b, the original non-mutated sup-tRNA should be included in this analysis to present the benefit of sup-tRNA variants.

9) There is general lack of detail of cDNA constructs used to collect data for Figure 1. What cDNA constructs are used for the sup-tRNAs? Were the target FLuc construct and sup-tRNA delivered on different or the same cDNA construct? Furthermore, the 5' and 3' sequence of the sup-tRNA

expression cassette should be included as a supplemental figure.

10) Diagrams depicting t-stem and anticodon-stem variants are only somewhat useful to the reader. These diagrams should also include the nucleotide substitutions made to show preservation or elimination of specific bonds (as example, Fig. 1 PMID: 2428035 and PMID: 34158503).

11) Representative RNA PAGE gels should be presented in supplemental data for clarification of tRNA synthesis methods. T7 polymerase is known to have high processivity and therefore have non-template nucleotide addition (PMID: 10496227) that can greatly impact tRNA activity.

Other issues

Regarding the choice of wording, the authors would do well to go through the manuscript with an eye to writing what they mean. Here are examples:

Line 32-33. “repurpose sense-codon decoding tRNA into efficient suppressors of the three nonsense mutation induced-PTCs” would be better as “repurpose each of three sense-codon decoding tRNAs into an efficient suppressor of UGA, UAG or UAA PTCs.”

Line 35. The meaning of “fine-tuning them for the cognate amino acids” is unclear.

Line 37. The meaning of “administered” is unclear.

Line 53. “at the UGA stop codon” – presumably this is in bacteria, but this should be clarified.

Line 48. The sentence beginning “One impediment is the relatively...”. After several readings, this sentence still makes little sense.

Line 54. “tRNASer UGA” should be “tRNASer UCA”.

Referee #2 (Remarks to the Author):

The manuscript entitled “Repurposed lipid-nanoparticulated tRNAs suppress pathogenic nonsense mutations “ by Ignatova and co-workers, presents exciting results that methodically describe the engineering of suppressor tRNA for use in the treatment of disease caused by nonsense mutations. The authors present a foundationally sound strategy to repurpose human tRNAs to decode premature stop codons by modifying tRNA identity elements to create three human suppressor tRNAs capable of efficient charging with the amino acids serine, arginine and glycine. They accurately describe the vast literature describing the identity elements found in tRNA that promote the interaction of cognate aminoacyl-tRNA synthetase for efficient aminoacylation with high fidelity. They present data describing the extensive sequence changes which led to discovery of tRNAs with the ability to decode nonsense codons in a reporter (FLuc) model for a common nonsense allele in the tripeptidyl peptidase I (TPP1) gene which causes the disease late infantile neuronal ceroid lipofuscinosis (CLN2). They further describe the effect of tRNA changes on eEFA1, which shares a binding site with the aminoacyl-tRNA synthetase. They describe how increased interaction with eEF1A is critical for stabilization of the amino acid on tRNASer, a relatively unstable amino-acyl conjugate.

The experiments, results and description in this manuscript represent a beautiful, clear and deep

understanding of interplay of aminoacyl synthetases with cognate tRNA and translation factors, critical for high fidelity, efficient decoding by tRNA during translation initiation, elongation and termination.

Based on this, the authors continue to demonstrate the ability of these engineered suppressor tRNAs to decode nonsense codons (UGA, UAG, UAA) derived from a disease allele in CFTR (S466X). They uncover a role for context of the stop codon, a principle that has been seen for other readthrough agents.

From these experiments, the authors present experiments that directly demonstrate the ability of the engineered suppressor tRNAs to restore expression in vivo of both a reporter mouse model (R208X-aLuc, a surrogate for CLN2) in the liver and through intra-tracheal injection, full-length CFTR harboring pathological mutation S466X or R553X, by utilizing a previously described lipid nanoparticle encapsulation delivery system, LUNAR. The results are clear and convincing and show great promise for this system as a therapeutic for genetic diseases caused by nonsense mutation.

The manuscript is clear and well-written. Congratulations to the team on a very nice addition to the landscape of potential therapies for patients with rare diseases caused by nonsense mutations.

Referee #3 (Remarks to the Author):

This manuscript aims to help address a very important unsolved problem in human medicine – how to treat diseases caused by loss of protein function secondary to nonsense mutations. The strengths of this study include a set of innovative approaches that collectively aim to improve the capacity for tRNAs to act as suppressors of nonsense mutations. These include optimization of the tRNAs at sites other than the anticodon regions (likely tapping into very interesting combinatorial allosteric effects that help achieve the subtle biophysical dance required for nonsense suppression), further optimizations of the tRNAs to maximize compatibility with the cognate amino acid, and the use of lipid nanoparticle-based delivery technology. The concept of using tRNAs to suppress pathogenic nonsense mutations has been pursued for over four decades, but clinical translation of this approach to provide clinical benefit has not yet been achieved. Thus, the level of impact for this paper rests on whether the data shows that the reported approaches for optimizing tRNA suppressors substantially advance the field toward increased likelihood of achieving therapeutic impact. The data in this current manuscript do not support the conclusion that this important goal has been achieved.

There are many previous reports showing that tRNA-based suppression of nonsense mutations can lead to increased expression of the corresponding proteins in cells, both in vitro and in vivo. The same was true with ataluren, which showed increased expression of CFTR in many in vitro and in vivo studies, and also showed increased expression of CFTR in people, and yet this drug failed to show clinical benefit in multiple clinical trials in people with cystic fibrosis, and then failed again to show benefit in clinical trials in people with Duchenne's muscular dystrophy. A critical limitation of this manuscript in its present form is that it only shows increased protein expression/function. It does not show recovery of physiology in clinically meaningful assays. There many reasons why just

showing increased protein expression/function is not sufficient. For example... The thresholds for extent and duration of increased expression/function required to meaningfully recover physiology are unknown; The likely and potentially competitive consequences of non-selective PTC suppression on cellular, tissue, and organismal physiology are unknown; There are likely toxicities of high concentrations of tRNAs that are not detectable with the rudimentary cell viability assays that were used in this study. Etc.

More specifically, the authors focus some of their key studies on cystic fibrosis. They demonstrate increased expression of CFTR in transfected CFBE41o- cells and changes in short circuit current in transfected Fischer rat thyroid monolayers. These assays are not sufficient to probe whether these new tRNAs have moved the needle. Alternatively, there are now several well-established assays to probe for recovery of airway host defenses in primary cultured airway epithelia from people with CF (ASL antibacterial activity, ASL pH, ASL viscosity, ASL height), and the Cystic Fibrosis Foundation has recently made primary airway epithelial cells from people with CF caused by nonsense mutations available to researchers via the RARE initiative. Moreover, some of these same assays performed in cultured airway epithelia derived from people with CF with the corresponding sensitive mutations have also been validated to respond to clinically approved small molecule modulators that are also known to provide clinical benefit to people with CF. Many of these same assays could also be performed with ataluren, which is not clinically effective, as an important comparator.

There are also several recently reported animal models for CF nonsense mutations. These include a rat model that is genetically engineered to bear the CFTR G542X nonsense mutation in the endogenous locus (Front Physiol 2020; 11: 611294). Importantly, and consistent with the arguments presented above, although single readthrough agent therapy caused functional restoration of CFTR in G542X rat tracheal epithelial cells, therapeutic efficacy was not observed in the G542X rats in vivo.

Thus, there is an accessible opportunity in CF to use a range of in vitro and in vivo assays to more rigorously and effectively probe whether these new tRNAs can restore airway host defenses, and whether they can really outperform state of the art tRNAs, or other approaches such as readthrough compounds, in their ability to restore physiology.

This approach might represent a significant step forward in this important area of biomedicine, but based on the data provided in this manuscript, it is currently not clear if this is the case. I am thus unable to support its publication in Nature.

Author Rebuttals to Initial Comments:

Point-by-point responses to the Referees' comments

Referee #1:

The authors have shown that they can efficiently suppress PTCs using suppressor tRNAs, and the manuscript certainly has merit (aside from the many issues raised because the writing has uncertain meaning).

We are thankful to the Reviewer for their overall positive assessment of our manuscript and gratefully acknowledge their critical comments, which we addressed in the revision with additional experiments.

That noted, as the reader becomes aware as early as line 45 with the sentence "A targeted strategy to correct nonsense mutations and alleviate translation...", the authors are not considering that most PTCs trigger nonsense-mediated mRNA decay (NMD). This sentence would more accurately read "A targeted strategy to correct nonsense mutations and alleviate the premature termination of translation and the resulting nonsense-mediated mRNA decay...". It is one accomplishment to suppress a PTC so that it encodes an amino acid, which the authors have done. However, since most PTCs trigger NMD once recognized by the translating ribosome, a second requisite accomplishment is for suppression to be sufficiently efficient during the time that NMD occurs. The authors cannot assess this efficiency since their reporter genes/mRNAs lack a PTC-distal intron/EJC and, thus, are immune to NMD.

The authors are set to assess the effect of tRNA-mediated suppression on NMD. To do this, they must test cells and mice expressing a full-length PTC-containing disease gene and compare the level of full-length protein deriving from this gene to that deriving from its normal (i.e. PTC-lacking) counterpart. Otherwise, the utility of their approach is not fully realized as a therapeutic.

Major Comments

1) The largest criticism of this study is the lack of using PTC-containing target transcripts that exhibit nonsense mediated decay (NMD). A significant part of the mechanism of nonsense-associated diseases that makes them so severe is the loss of target gene transcript expression through NMD. Indeed, the major advancement of this study is the delivery of suptrRNAs as RNA using LUNAR2 LNP formulation to cultured cells, lung and liver. However, suppression of PTCs within the mRNAs chosen, i.e., those deriving from FLuc, Cre and CFTR genes that do not harbor at least one intron downstream of the PTC, does not provide insight into the ability of suptrRNAs to inhibit NMD. Determining the ability of sup-tRNAs to suppress PTCs encoded in endogenous transcripts that undergo proper post-transcriptional processing is an important aspect not addressed in this study.

The Reviewer is correct that NMD may diminish mRNA abundance and thus, decrease tRNA suppressor-driven mRNA utilization (i.e. in a manner that it could outcompete the readthrough). Reflecting of the comments of the Reviewer, we included a series of experiments to address the effect of NMD decay on the tRNA-dependent readthrough. For this, we used two different systems: (1) primary nasal epithelial cells derived from a cystic fibrosis (CF) patient who carries a homozygous nonsense mutation

(R1162X/R1162X), and (2) 16HBE14o- cells, the immortalized version of human bronchial epithelial cells in which the endogenous WT-CFTR locus has been gene edited using CRISPR/Cas9 to create isogenic cell lines 16HBEge CFTR R1162X/- cells. Both cell systems express full-length CFTR protein with all its native introns and exons. While introns downstream of PTCs are considered to majorly driving the NMD, increasing evidence suggests that also flanking introns upstream of the PTC are contributory to the effect. Thus, we believe that the natural full-length gene context is more relevant system compared to constructs containing single/few introns. The cell systems were obtained from the Cystic Fibrosis Foundation Therapeutics Laboratory (Lexington MA, USA) within the RARE program. The primary nasal epithelia from homozygous patients are extremely rare and this was the genotype we could obtain as a genotypic match between both primary patient-derived cells and the isogenic 16HBEge cells.

The results are summarized in Figure 4. We are greatly thankful to the Reviewer for this suggestion. Briefly, our results show that the tRNA suppression successfully competes with the NMD and the tRNA suppressor alone establishes a sentinel round of full-length peptide translation. As we elaborate in the manuscript, although NMD inhibitors greatly stabilize the CFTR mRNA, they only modestly enhance protein production and do not synergize the tRNA effect on channel activity. Moreover, a long treatment with NMD inhibitors caused some adverse effects on cell vitality, as also already reported in the literature (PMID: 33396210 and PMID: 31831337). In sum, our results suggest that in the case we tested (i.e., tRT5) addition of NMD inhibitors as adjuvants may not be necessary.

The Reviewer suggests expressing intron-containing constructs using human cell lines or mice. Our ultimate goal is optimizing tRNA therapeutics for application in humans, and we have chosen one specific disease model – cystic fibrosis – to test the applicability of our tRNA suppressors in disease-related settings. For CF, there are only few animal models available for nonsense variants, which however are limited in their application mainly because of their limited efficacy as a tool for human drug development (i.e. no pharmacologic rescue has yet been shown). For example, Drs. Birket, Rowe, Sorscher and Hong (co-authors on our paper) are also co-developers of the first and most widely used CF rat models. As noted by the Reviewer, the highly valuable G542X rat is still being established as a reliable tool for predicting therapeutic efficacy in humans. For example, no pharmacologic rescue has yet been demonstrated in the G542X CF rat. Moreover, this model utilizes a rat - not human- CFTR sequence, making it less useful for testing human-directed tRNAs). In the present studies we concentrated on optimizing tRNA suppressors for human use, and because human patient-derived primary cells are among the best available indicators of clinical benefit (and in some cases represent an FDA endorsed system for this purpose), we have included this model (i.e., primary nasal cells obtained from a CF patient with homozygous PTC genotype) as relevant for evaluating the overall strategy.

2) One of the significant issues/concerns with use of sup-tRNA RNAs as therapeutics is their relatively short half-lives (PMID: 33567469) due to their degradation. All of the experiments in this study were performed over a short time-frame. The time-course of sup-tRNA activity should be determined in vivo and in cell culture as this provides crucial insight into the therapeutic promise of sup-tRNAs.

We have included evidence for the stability of the tRNA suppressor (Fig. 3a). We used the gene-edited HBE16ge cells as they well mimic the natural CFTR expression environment. Our tRNAs are stable for at least 72 hours after a pulsed single transfection of in vitro transcribed tRNAs.

The mentioned publication (PMID: 33567469) is a review on tRNA applications as therapeutics, in which we could not find any data (including in the primary literature cited therein) on the half-life time of tRNAs administered or transfected directly as a mature in vitro transcribed tRNA. To the best of our

knowledge, this is a unique strategy we describe here – nanoparticulated administering of mature tRNAs – and data on the stability are not yet available in the published literature.

3) It should be stated how the original tRNA sequences that were modified in this study were identified. Examining the tRNAscan-SE Genomic tRNA Database (Cite: PMID: 26673694r & 18984615), there appears to be many tRNA sequences one could choose from to generate active sup-tRNAs. Furthermore, a previously published study (cite: PMID: 30778053) generated a library of sup-tRNAs from all unique tRNA sequences (23 serine, 29 Arg, 20 Gly) and tested their suppression activity to define those with the most activity. While it appears that the authors were able to identify active sup-tRNA sequences for each tRNA family tested, determining how these sequences were chosen from the many isoacceptors would be helpful to the reader. Notably, the GtRNAdb Gene Symbol and/or tRNAscan-SE ID should be used for each tRNA sequence listed.

We have included the tRNAscan-SE IDs of the tRNA^{Ser}, tRNA^{Arg} and tRNA^{Gly} isoacceptors whose backbone was used to produce the first generation tRNAs with only exchanged anticodons, i.e. tS, tR and tG, respectively (Extended Data Table 1). Previous work (PMID: 20026070 for tRNA^{Ser} and PMID: 30778053 for tRNA^{Arg} and tRNA^{Gly}) suggested that among the isoacceptors of one tRNA family (that are tRNAs charged with the same amino acid), the ones we selected were among those with the highest readthrough following the exchange of their natural anticodon to decode a stop codon.

The general strategy of repurposing tRNAs along with the choice of tRNAs encoding Ser, Arg or Gly codons is mentioned in the main text (l. 56-84). We also elaborate on the tRNA variants and the design strategy in a dedicated subsection ‘tRNA design’ in the Methods section.

4) It is unclear why the specific anticodon stem and t-stem were chosen. While there have been several t-stem (example PMID: 21402928) and anticodon-stem mutations (example PMID: 2428035) that have been made to enhance sup-tRNA activity, relatively few variants were tested here. It would be beneficial to the reader to understand what methods were used to predict how specific variants were chosen to test.

To increase the readthrough efficiency of tS, tR and tG, we subjected them to a comprehensive set of sequence changes targeting two regions that are crucial for tRNA function in translation: (1) the anticodon (AC)-stem and loop which modulates accuracy of decoding and (2) the T Ψ C-stem that determines binding affinity to the elongation factor (ref. 15-18). For the changes in the AC stem, we have chosen basepairs preferred in natural suppressor tRNAs (e.g. U31-U39 (PMID: 22068346) or in the tRNA^{Sec} as it naturally decodes UGA codon).

The eukaryotic (eEF1A) and bacterial (EF-Tu) elongation factors share conserved sites of aminoacyl-tRNAs binding (PMID: 11106763, ref. 19). In the T Ψ C stem the first three basepairs are the most important for binding to the EF-Tu (PMID:19452597), thus, we mainly focused on mutating these three basepairs. The nucleotide identities of the basepairs were chosen to cover a variety of binding affinities, but preferably such that have higher $\Delta\Delta G^\circ$ values, because naturally such tRNAs exhibit higher efficacy in recoding stop codons (PMID:34158503). As explained in the text, to calculate the binding affinities of the T Ψ C stem-modified variants to eEFA1 (Fig. 1a, table) we used the energy contribution of each basepairs

that has been determined for the binding of the homologous prokaryotic counterpart (EF-Tu and published in PMID:19452597, ref. 37). We have extended the explanation on the tRNA designs in a designated subsection in the Methods section.

5) Comparisons of sup-tRNA variant PTC rescue are shown in Figure 4. These should be compared to the original sup-tRNA sequences to show the benefit of variants over original sequences. Furthermore, a dose-dependence of the original sup-tRNA should be included in Extended Data Figure 3a and b, for comparison to the tSA1T5 variant. It would be expected to have a rightward shift of PTC suppression efficiency and toxicity.

As suggested by the Reviewer, we have included data on the efficacy of the first-generation tRNAs, i.e. those with changed anticodon but with the body of an intact sense-codon decoding tRNA (tR, tS). These include dose-dependence of the readthrough activity (black symbols, Extended data Fig. 4a, b) and readthrough efficiency on PTC-CFTR transcripts (Fig. 3, former Fig.4). These data emphasize on the marked improvement of suppression efficiency achieved in our studies by modulating different regions of tRNA.

6) The mass spec performed to ensure that the correct amino-acid is incorporated is insufficient. These experiments do not report on the percent incorporation of serine, only that it was incorporated. The absolute fidelity of PTC suppression should be determined. An important aspect of sup-tRNAs is that they predominately incorporate the correct amino acid (cite: PMID: 30778053) and not the near-cognate as a result of small molecule-dependent readthrough (PMID: 27702906). However, modifications to the anticodon stem and t-stem could impart reduced aminoacyl-tRNA synthetase fidelity. Comparing the amino acid incorporation fidelity of the original and variant sup-tRNAs is important for the therapeutic promise of the presented sup-tRNA variants.

As the Reviewer notes, despite strictly preserving the recognition signals of the seryl-tRNA-synthetase, still some of the sequence modifications of tSA1T5 could compromise the aminoacylation. To prove for the fidelity of incorporation of Ser and indirectly the aminoacylation fidelity, we also looked for incorporation of threonine, tryptophan or selenocysteine, for the following reasons: (1) sequence resemblance with some parts of the TΨC-stem and acceptor stem of tRNA^{Thr}GCU; (2) the ability of tRNA^{Sec} (in conjunction with secondary structural elements) to decode UGA codons; and (3) the highest intrinsic ability of tRNA^{Trp} among natural tRNAs to sporadically decode an UGA codon. None of these amino acids has been incorporated suggesting a very high fidelity of the seryl-tRNA-synthetase in aminoacylating tSA1T5. Also, the lack of 'near-cognate' sporadic decoding (e.g. by tRNA^{Sec} or tRNA^{Trp}) support the high efficacy of our tRNA designs. We have mentioned this in the manuscript (Extended Data Fig. 3). We apologize for not having including this in the previous version.

7) It is peculiar that only two experiments are reported for WB results in Fig 4b. This makes it impossible to perform statistics.

We have included more replicates and statistical analysis. The number of the independent replicates along with the statistical method is indicated in each figure, for each panel.

8) Representative short-circuit currents should be included in the supplemental data for Fig. 4c and WT CFTR currents should be presented for PTC suppression efficiency comparison. As with Fig 4a and 4b, the original non-mutated sup-tRNA should be included in this analysis to present the benefit of sup-tRNA variants.

A representative example of the primary short-circuit current tracings from the FRT (Extended Data Fig. 10) and primary patient-derived cells are included (Extended Data Fig. 11). As suggested by the Reviewer, we show wild-type CFTR data and first-generation tRNA (tR) results (Fig. 3c).

9) There is general lack of detail of cDNA constructs used to collect data for Figure 1. What cDNA constructs are used for the sup-tRNAs? Were the target FLuc construct and sup-tRNA delivered on different or the same cDNA construct? Furthermore, the 5' and 3' sequence of the sup-tRNA expression cassette should be included as a supplemental figure.

We have included a schematic of the constructs for clarity (Fig. 1d,e and Extended Data Fig.4a). We would like to emphasize that all tRNAs were directly delivered (i.e., co-transfected in cell culture models or primary cells, or administered as nanoparticulated formulations in mice) as ready-to-use, translationally competent tRNAs – a novel and unique element of our tRNA-based treatment strategy. Prior to transfection or administration, all tRNAs were in vitro transcribed and purified. We have included clarification notes, where suitable to emphasize that tRNAs are not plasmid-borne but in vitro transcribed. Our approach shares resemblance with the administrations of the mRNA-based vaccines whose recent successful applications brings to the fore nanoparticulated RNA therapeutics.

10) Diagrams depicting t-stem and anticodon-stem variants are only somewhat useful to the reader. These diagrams should also include the nucleotide substitutions made to show preservation or elimination of specific bonds.

We have included the nucleotide identities in the diagrams (Fig. 1a and Extended Data Fig. 1).

11) Representative RNA PAGE gels should be presented in supplemental data for clarification of tRNA synthesis methods. T7 polymerase is known to have high processivity and therefore have non-template nucleotide addition (PMID: 10496227) that can greatly impact tRNA activity.

tRNAs are produced with homogenous 3' ends (a new figure added, Extended Data Fig. 2). Reflecting on the Reviewer's question, we included an additional experiment: we in vitro transcribed a tRNA variant with 5'-modified 2'-methoxy (2'-OMe) primer which enables precision in the T7-polymerase synthesis. The activity of tRNA from the 2'-OMe-modified template is nearly identical to that transcribed using conventional primers (Extended Data Fig. 2c), suggesting the uniform 3' ends of our tRNA batches.

Other issues

Regarding the choice of wording, the authors would do well to go through the manuscript with an eye to writing what they mean. Here are examples:

Line 32-33. “repurpose sense-codon decoding tRNA into efficient suppressors of the three nonsense mutation induced-PTCs” would be better as “repurpose each of three sense-codon decoding tRNAs into an efficient suppressor of UGA, UAG or UAA PTCs.”

Line 35. The meaning of “fine-tuning them for the cognate amino acids” is unclear.

Line 37. The meaning of “administered” is unclear.

Line 53. “at the UGA stop codon” – presumably this is in bacteria, but this should be clarified.

Line 48. The sentence beginning “One impediment is the relatively...”. After several readings, this sentence still makes little sense.

Line 54. “tRNA^{Ser} UGA” should be “tRNA^{Ser} UCA”. It is correct here. We mean the natural tRNA^{Ser}, not the one with the exchanged anticodon to decode a PTC. The original sequence is now introduced in Extended data Table 1.

The text is edited.

Referee #2:

The manuscript entitled “Repurposed lipid-nanoparticulated tRNAs suppress pathogenic nonsense mutations” by Ignatova and co-workers, presents exciting results that methodically describe the engineering of suppressor tRNA for use in the treatment of disease caused by nonsense mutations. The authors present a foundationally sound strategy to repurpose human tRNAs to decode premature stop codons by modifying tRNA identity elements to create three human suppressor tRNAs capable of efficient charging with the amino acids serine, arginine and glycine. They accurately describe the vast literature describing the identity elements found in tRNA that promote the interaction of cognate aminoacyl-tRNA synthetase for efficient aminoacylation with high fidelity. They present data describing the extensive sequence changes which led to discovery of tRNAs with the ability to decode nonsense codons in a reporter (FLuc) model for common nonsense alleles in the tripeptidyl peptidase I (TPP1) gene which causes the disease late infantile neuronal ceroid lipofuscinosis (CLN2). They further describe the effect of tRNA changes on eEFA1, which shares a binding site with the aminoacyl-tRNA synthetase. They describe how increased interaction with eEF1A is critical for stabilization of the amino acid on tRNA^{Ser}, a relatively unstable amino-acyl conjugate.

The experiments, results and description in this manuscript represent a beautiful, clear and deep understanding of interplay of aminoacyl synthetases with cognate tRNA and translation factors, critical for high fidelity, efficient decoding by tRNA during translation initiation, elongation and termination.

Based on this, the authors continue to demonstrate the ability of these engineered suppressor tRNAs to decode nonsense codons (UGA, UAG, UAA) derived from a disease allele in CFTR (S466X). They uncover a role for context of the stop codon, a principle that has been seen for other readthrough agents.

*From these experiments, the authors present experiments that directly demonstrate the ability of the engineered suppressor tRNAs to restore expression in vivo of both a reporter mouse model (R208X-*aLuc*, a surrogate for *CLN2*) in the liver and through intra-tracheal injection, full-length CFTR harboring pathological mutation S466X or R553X, by utilizing a previously described lipid nanoparticle encapsulation delivery system, LUNAR. The results are clear and convincing and show great promise for this system as a therapeutic for genetic diseases caused by nonsense mutation.*

The manuscript is clear and well-written. Congratulations to the team on a very nice addition to the landscape of potential therapies for patients with rare diseases caused by nonsense mutations.

We were very pleased to read the comment of the Reviewer who emphasized on our successful achievement with regard to design of tRNAs for suppression by simultaneously considering the chemical nature of the amino acid and the auxiliary interactions of tRNAs with both translation factors and ribosome, thus enhancing its therapeutic significance and readthrough performance.

Referee #3:

This manuscript aims to help address a very important unsolved problem in human medicine – how to treat diseases caused by loss of protein function secondary to nonsense mutations. The strengths of this study include a set of innovative approaches that collectively aim to improve the capacity for tRNAs to act as suppressors of nonsense mutations. These include optimization of the tRNAs at sites other than the anticodon regions (likely tapping into very interesting combinatorial allosteric effects that help achieve the subtle biophysical dance required for nonsense suppression), further optimizations of the tRNAs to maximize compatibility with the cognate amino acid, and the use of lipid nanoparticle-based delivery technology.

We thank the Reviewer for emphasizing the importance of this study and our approach to successfully suppress nonsense mutations, both using in vitro model systems and in vivo in mice.

The concept of using tRNAs to suppress pathogenic nonsense mutations has been pursued for over four decades, but clinical translation of this approach to provide clinical benefit has not yet been achieved. Thus, the level of impact for this paper rests on whether the data shows that the reported approaches for optimizing tRNA suppressors substantially advance the field toward increased likelihood of achieving therapeutic impact. The data in this current manuscript do not support the conclusion that this important goal has been achieved.

We agree with the Reviewer that demonstration in a primary cell culture model would provide a crucial support of potential clinical utility. Thus, in the revised manuscript, we included tests in CF-patient-derived primary cells (human nasal epithelial, hNE), which are viewed by many in the field together with the extant literature as predictors of clinically beneficial effects. The correction of hNE shown in the

revised paper (4-6 $\mu\text{A}/\text{cm}^2$) is in a range that suggests clinical benefit. In addition, FRT cell data indicate strong PTC readthrough nearing 10% of wildtype CFTR levels – an important threshold interpreted by FDA as clinically meaningful for the rescue of CFTR and sufficient to lead to drug approval. While the FRT model does not fully account for loss of CFTR transcript due to NMD, when coupled with the significant improvement in primary cells, the data suggest the potential for clinical benefit.

In the revised paper, we also present ribosome profiling showing reestablishment of translation at a specific PTC of interest. Furthermore, the absence of off-target effects (e.g., at canonical stop codons) provides evidence for a PTC rescue mechanism capable of substantial safety. Moreover, this new data supports an important and highly impactful tRNA-based intervention with specific, cutting-edge LUNAR LNPs optimized for clinical use. Together, these new data furnish compelling evidence for strong clinical promise of an innovative tRNA-based therapeutic strategy, which is predicated on tailoring suppressor tRNAs to individual PTCs to provide relevant amino acid replacement and foster precision in the individualized therapy of hereditary diseases caused by nonsense mutations.

There are many previous reports showing that tRNA-based suppression of nonsense mutations can lead to increased expression of the corresponding proteins in cells, both in vitro and in vivo. The same was true with ataluren, which showed increased expression of CFTR in many in vitro and in vivo studies, and also showed increased expression of CFTR in people, and yet this drug failed to show clinical benefit in multiple clinical trials in people with cystic fibrosis, and then failed again to show benefit in clinical trials in people with Duchenne’s muscular dystrophy. A critical limitation of this manuscript in its present form is that it only shows increased protein expression/function. It does not show recovery of physiology in clinically meaningful assays. There many reasons why just showing increased protein expression/function is not sufficient. For example... The thresholds for extent and duration of increased expression/function required to meaningfully recover physiology are unknown; The likely and potentially competitive consequences of non-selective PTC suppression on cellular, tissue, and organismal physiology are unknown; There are likely toxicities of high concentrations of tRNAs that are not detectable with the rudimentary cell viability assays that were used in this study. Etc.

It is critical to demonstrate safety of this more efficacious approach at PTC suppression to demonstrate potential clinical utility. In response to Reviewer’s comment, we performed ribosome profiling of mice following administration of suppressor tRNA by intratracheal microsyringe instillation in lungs or intravenously to liver to determine with nucleotide precision any adverse or off-target effects. The analysis shows that our tRNA suppressors do not augment readthrough at native stop codons or at internal stop codons (i.e., intrinsically more prone to readthrough than stop codons designating terminus of a coding sequence) beyond the natural stochastic background readthrough level. Among other PTC readthrough strategies, therefore, our technology offers a unique mechanism for highly efficient PTC suppression at cellular, tissue and consequently organismal level.

In the revised version, we also included tRNA stability data using tRNA-tailored microarrays to specifically capture and quantify tRNA levels in human cells. The measured high stability of tRNA supports fairly low frequency of administration, although true dosing frequency and schedule can be only defined in the clinical trial setting and might be disease-specific.

Response concerning the Reviewer's important points related to ataluren are provided in the sections below.

More specifically, the authors focus some of their key studies on cystic fibrosis. They demonstrate increased expression of CFTR in transfected CFBE41o- cells and changes in short circuit current in transfected Fischer rat thyroid monolayers. These assays are not sufficient to probe whether these new tRNAs have moved the needle. Alternatively, there are now several well-established assays to probe for recovery of airway host defenses in primary cultured airway epithelia from people with CF (ASL antibacterial activity, ASL pH, ASL viscosity, ASL height), and the Cystic Fibrosis Foundation has recently made primary airway epithelial cells from people with CF caused by nonsense mutations available to researchers via the RARE initiative. Moreover, some of these same assays performed in cultured airway epithelia derived from people with CF with the corresponding sensitive mutations have also been validated to respond to clinically approved small molecule modulators that are also known to provide clinical benefit to people with CF. Many of these same assays could also be performed with ataluren, which is not clinically effective, as an important comparator.

Reflecting on Reviewers comment, we performed additional studies of tRNA suppressor (tRT5) efficacy for restoring airway homeostasis in hNE cells derived from CF patient homozygous for R1162X/R1162X. In addition to restoration of CFTR-specific currents noted above, we also demonstrated improved downstream physiological function using ASL depth. An adequate ASL plays a major role during mucus clearance and respiratory defense against infection. Thus, we measured the ASL height using state-of-the-art optical coherence tomography and show a strong and time dependent enhancement in airway surface liquid height (Figure 4f), complementing electrophysiological findings in the same donor.

The Reviewer also mentions several other assays (e.g., ASL pH or viscosity, ASL antibacterial activity). There are a comparatively small number of studies and laboratories that perform measurements to probe ASL homeostasis in this manner. Moreover, to our knowledge only one group has published a comprehensive set of such measurements (e.g. PMID: 22763554). We contacted Dr. J. Zabner at University of Iowa (a leader in this area) who is among the developers of such approaches. He felt that even with evidence of CFTR PTC rescue in primary cells (e.g., 4-6 $\mu\text{A}/\text{cm}^2$ as shown here in hNEs) which might otherwise suggest clinical benefit in CF individuals, it is not yet known whether ASL composition (pH, antibacterial properties) can effectively serve as surrogate or correlative metrics for clinical improvement. While the field has not yet delineated clinical endpoints for thresholds such as hNE pH and viscosity, both I_{sc} and ASL depth have proven themselves useful in this respect. Please also note that the RARE initiative of the Cystic Fibrosis Foundation and CFFT Laboratory had no available primary hBE cells homozygous for nonsense mutations (hBEs have higher expression than hNEs) that could be effectively targeted by our optimized tRNAs.

As also suggested by the Reviewer, we included PTC124 (ataluren) at two different concentrations as an important comparator. Ataluren was ineffective in rescuing CFTR protein expression (Figure 4c compare to the control, dashed line), although it modestly stabilized the CFTR mRNA (Figure 4b), corroborating published literature showing no effect of ataluren in correcting clinically relevant PTCs. Note that the usefulness of ataluren was questioned by CF experts while the drug was still being developed (e.g., PMID: 19208811). We feel that clinical failure of ataluren (which works by a mechanism completely distinct from tRNA treatment) should not preclude advancement of other promising therapies for PTCs.

In summary, our results from multiple (also FDA-utilized) cell systems for testing and approving new CF therapies (e.g., FRT, gene-edited bronchial cells and primary hNE cells), together with clear establishment of strong CFTR functional rescue, biochemical improvement of band C and increase of airway surface liquid depth provide compelling evidence for expected therapeutic benefit and, given the unique PTC specificity of the approach, justify advancement to clinical testing.

There are also several recently reported animal models for CF nonsense mutations. These include a rat model that is genetically engineered to bear the CFTR G542X nonsense mutation in the endogenous locus (Front Physiol 2020; 11: 611294). Importantly, and consistent with the arguments presented above, although single readthrough agent therapy caused functional restoration of CFTR in G542X rat tracheal epithelial cells, therapeutic efficacy was not observed in the G542X rats in vivo. Thus, there is an accessible opportunity in CF to use a range of in vitro and in vivo assays to more rigorously and effectively probe whether these new tRNAs can restore airway host defenses, and whether they can really outperform state of the art tRNAs, or other approaches such as readthrough compounds, in their ability to restore physiology.

The Reviewer suggests using a rat model in vivo or cultured airway epithelia derived from CF rats to provide further support for human efficacy. Drs. Birket, Rowe, Sorscher and Hong (co-authors on our paper) are also co-developers of the first and most widely used CF rat models. As noted by the Reviewer, the highly valuable G542X rat is still being established as a reliable tool for predicting therapeutic efficacy in humans. For example, no pharmacologic rescue has yet been demonstrated in the G542X CF rat. Moreover, this model utilizes a rat - not human- CFTR sequence, making it less useful for testing human-directed tRNAs). Finally, in the present studies we concentrated on optimizing tRNA suppressors for human use, and because human patient-derived primary cells are among the best available indicators of clinical benefit (and in some cases represent an FDA endorsed system for this purpose), we have included this model (i.e., primary nasal cells obtained from a CF patient with homozygous PTC genotype) as relevant for evaluating the overall strategy (Figure 4).

This approach might represent a significant step forward in this important area of biomedicine, but based on the data provided in this manuscript, it is currently not clear if this is the case. I am thus unable to support its publication in Nature.

We believe that the additional experiments included in the revised manuscript rigorously probe the ability of novel suppressor tRNAs to provide a significant and highly impactful advance. We present new and important evidence for: 1) an innovative strategy to optimize the tRNA suppression efficacy and specificity to PTCs of interest, 2) novel and robust LNP-based delivery mechanisms for use against refractory human diseases caused by PTCs in target tissues, and 3) activity against several different nonsense alleles – supporting broad spectrum of activity and a platform technology with potential relevance to a large number of inherited disease states. Our results with disease-associated protein (CFTR) rescue in physiologically relevant systems (primary airway epithelia, CF cell lines, well-established FRT models, CRISPR-Cas modified airway epithelia) and restoration of airway surface liquid depth, establish a crucial step forward in the important area of translational research for patients with CF and no other available molecular intervention.

Reviewer Reports on the First Revision:

Referee #1 (Remarks to the Author):

The authors of the manuscript did an excellent job addressing both the major and minor concerns raised in the first review. However, there are still some comments that need to be addressed which mostly pertain to the new results presented in Figure 4.

Comments:

1) This statement is confusing, or not correct, "Among the three amino acids, Ser, Arg and Gly, the most destabilizing is Gly followed by Arg and Ser, thus mutations in TΨC-stem stabilizing the interactions with eEF1A (tGT6) enhanced the tG readthrough activity (Extended Data Fig. 1). Looking at Extended Data Fig. 1, t-stem substitutions did not significantly enhance tG suppression efficiency as stated. Therefore, tG cannot be added to the list of sup-tRNAs enhanced by this study.

2) It is peculiar in Fig. 2a and b, only tS was used. Then in Fig. 2e, only tSA1T5 was used. Is there a reason why the comparison of tS and tSA1T5 was not made in vivo? It seems there is a missed opportunity to show how t-stem and anti-codon stem sequence changes enhance sup-tRNA activity in vivo, with the use of the therapeutic delivery agent. Results presented in Fig. 4 also lack the comparison of the original tRNA to the one with nucleotide substitutions.

3) It would be informative to show 6 and 24hr aLuc rescue in the main Fig. 2b. With sup-tRNAs delivered as RNA, the real concern is durability of effect. At 6hrs, the higher concentration of tS is detrimental; however, at 24hr, it is beneficial, which is an important finding. One thing to take into consideration is that it appears that the durability of sup-tRNA suppression is confounded by the degradation of the target aLuc mRNA. Therefore, analysis of percent drop in WT aLuc and R208X + tS would be helpful between 6 and 24hr. Further, there is significant spurious readthrough of aLuc R208X at 24hr. Thus, the interpretation of results stated here, "In the liver, 24 h post i.v. we detected tRNA dose-dependent rescue of R208X-aLuc expression reaching up to 33% of the non-PTC aLuc (R208S-aLuc) expression (Fig. 2 and Extended Data Fig. 5). Even at 6 h post dosing, the tS-corrected R208X-aLuc expression reached 13% (Fig. 2a,b)," is misleading. If the spurious readthrough were to be subtracted, percent rescue would be ~28%, which is still impressive.

4) Why was a Cre transcript with two UGA PTCs used in experiments outlined in Fig. 2? Was this because there is too much spurious readthrough of one PTC in Cre that resulted in high background activity? This implies an exceptionally "easy" PTC containing transcript to target. This issue paired with a Cre mediated event that is permanent, integrates sup-tRNA suppression activity over time. With a disease causing PTC, the endogenous transcripts and proteins are subject to degradation over time, and therefore the activity of the delivered sup-tRNA needs to be durable. Do the authors suggest daily inhalation of sup-tRNAs to maintain a steady state level of PTC readthrough?

5) The addition of results in Fig. 4 significantly adds to the study. These are very impressive Western blots. However, the amount of CFTR mRNA expression in panel b, CFTR protein expression in panel c and d, short circuit current in panel e and ASL height in panel f should all be compared to WT levels.

Comparison to WT levels of all of these readouts is what is relevant for a proposed therapeutic, not fold-change over control.

6) Why was lipofectamine 3000, rather than the lipid-nanoparticle, used to transfect 16HBEge- and hNE cells in Fig. 4? A high delivery efficiency with the repurposed lipid-nanoparticulated tRNAs to primary human cells would significantly add to the findings of this paper. Further, it is not understood how hNE and 16HBEge- cells were transfected in Transwells. It is well known that these cell types are resistant to transfection, especially once they have formed tight junctions and have polarized. Therefore, it would be reasonable to conclude that low transfection efficiency greatly influenced the results.

7) This statement “Finally, we addressed the efficacy of the nonsense suppressor tRNAs to restore expression of the full-length CFTR protein harboring different pathogenic mutations in cell models and patient-derived primary epithelial cells. PTC-tailored suppressor tRNAs were specific to the PTC identity and context (Extended Data Fig. 10a) and rescued full-length S466X-CFTR (X=UGA or UAA) or R553X-CFTR (X=UGA) transiently transfected in CFBE41o- cells (Fig. 3a and Extended Data Fig. 10b).”, is very misleading. The way this sentence is worded leads one to believe that a PTC was suppressed in a CFTR transcript that was expressed from the endogenous gene, i.e. a transcript that experiences post-transcriptional processing. Here, the CFTR mRNA with CF-causing PTCs was introduced into cells using lipofectamine 3000, which notably is an mRNA that is not targeted for NMD. Moreover, it is unclear how this is different or more impactful than transfecting these components into HEK293 cells.

8) Overall, in Fig. 3 and 4, there are different suptRNAs used in different cell types and with different CFTR mutations in different CFTR transcript types. Therefore, it is difficult to really compare all of the results. It will be hard to follow what seems to be patch-work results for many readers. For comparison of a CF PTC expressed from a transgene with no post-transcriptional regulation and an endogenous CFTR transcript, where post-transcriptional modification occurs, R1162X in 16HBEge- and FRT cells can be studied.

9) The panels in Fig. 4 should be labeled as 16HBEge- or hNE cells so it is clear to the reader which data came from what.

10) The right y-axis in Fig. 3 panel c should indicate percent WT rescue and the difference in CFTR functional rescue between tR and tRT5 should be highlighted.

Minor

11) When the two different NMD inhibitors (NMD14 and SMG1) are used, it should be made clearer in the manuscript which one is used and when. Also, why are two different NMD inhibitors used in this study?

12) It is stated that short circuit currents shown in Figure 4 are area under the curve. However, this cannot be possible as it appears both F&I and Inh172 responses are plotted.

Referee #3 (Remarks to the Author):

The authors have satisfactorily responded to my comments. I am happy to support publication of the revised manuscript in Nature

Author Rebuttals to First Revision:

Point-by-point responses to the Referee #1 comments

Referee #1:

The authors of the manuscript did an excellent job addressing both the major and minor concerns raised in the first review. However, there are still some comments that need to be addressed which mostly pertain to the new results presented in Figure 4.

We are very pleased about this overall positive assessment of the Referee who acknowledges our thorough revisions and feels that we satisfactorily addressed their previous comments.

The outstanding comments we have addressed in the following way (edits are designated red in the manuscript):

1) This statement is confusing, or not correct, "Among the three amino acids, Ser, Arg and Gly, the most destabilizing is Gly followed by Arg and Ser, thus mutations in TΨC-stem stabilizing the interactions with eEF1A (tGT6) enhanced the tG readthrough activity (Extended Data Fig. 1). Looking at Extended Data Fig. 1, t-stem substitutions did not significantly enhance tG suppression efficiency as stated. Therefore, tG cannot be added to the list of sup-tRNAs enhanced by this study.

We have edited the text about tG variants to clearly state that alterations in TΨC-stem did not enhance suppression activity owing to the intrinsic hyper accuracy of the natural tRNAs^{Gly}. However, we do like to keep this example, as it brings an important issue to consider when repurposing different tRNA families to suppress PTCs. Briefly, our reasoning for choosing the tRNA families that bear Ser, Arg or Gly, was driven by the different thermodynamic contribution of the cognate amino acid on the tRNA interactions with elongation factor and hence, on their decoding efficacy as suppressor. These three amino acids span the whole spectrum of thermodynamic effect, from Gly being the most destabilizing, followed by Arg and Ser. Considering these different thermodynamic contributions, we discuss different strategies tailored repurposing of tRNAs. Mutations to stabilize the TΨC-stem interactions with elongation factor were the least effective with tG, suggesting that for largely destabilizing amino acids, as Gly, additional mutations (outside the TΨC-stem) are needed to repurpose this naturally hyper accurate tRNA backbone. Thus, although the least efficient, we believe this example emphasizes one of the major messages of our work that unique design principles should be established for each tRNA family.

Reflecting on the Referee's comment we agree that the rationale behind the tRNA families' selection was not clear. We have thoroughly reorganized and edited the text to highlight the differences among three tRNA families (pages 4-7) and have pointed out at the low efficacy in tRNA^{Gly}.

2) It is peculiar in Fig. 2a and b, only tS was used. Then in Fig. 2e, only tSA1T5 was used. Is there a reason why the comparison of tS and tSA1T5 was not made in vivo? It seems there is a missed opportunity to show how t-stem and anti-codon stem sequence changes enhance sup-tRNA activity in vivo, with the use

of the therapeutic delivery agent. Results presented in Fig. 4 also lack the comparison of the original tRNA to the one with nucleotide substitutions.

In the revised version, we included a comparison between tS and tSA1T5 in mice (see new Fig. 2). Simultaneously modulating TΨC-stem and AC-stem in tS largely improves the tSA1T5 efficiency in vivo. We agree with the Referee that this data provides a direct in vivo comparison of the efficacy of the different tRNA^{SUP} generations and supports the in vitro optimization studies.

In general, based on the Reviewer's comment about Fig. 4, in all experiments we have included the comparison to the first generation tRNAs (tR or tS), i.e the one with exchanged anticodon only. [Please note, that in the revision the figures' and panels' designations have changed.]

3) It would be informative to show 6 and 24hr aLuc rescue in the main Fig. 2b. With sup-tRNAs delivered as RNA, the real concern is durability of effect. At 6hrs, the higher concentration of tS is detrimental; however, at 24hr, it is beneficial, which is an important finding. One thing to take into consideration is that it appears that the durability of sup-tRNA suppression is confounded by the degradation of the target aLuc mRNA. Therefore, analysis of percent drop in WT aLuc and R208X + tS would be helpful between 6 and 24hr. Further, there is significant spurious readthrough of aLuc R208X at 24hr. Thus, the interpretation of results stated here, "In the liver, 24 h post i.v. we detected tRNA dose-dependent rescue of R208X-aLuc expression reaching up to 33% of the non-PTC aLuc (R208S-aLuc) expression (Fig. 2 and Extended Data Fig. 5). Even at 6 h post dosing, the tS-corrected R208X-aLuc expression reached 13% (Fig. 2a,b)," is misleading. If the spurious readthrough were to be subtracted, percent rescue would be ~28%, which is still impressive.

We have added a plot to compare the activity of the control R208S-aLuc at 6h and 24h (new Extended Fig. 5a and revised Fig. 2b). To support the notion that the aLuc signal decrease is due to the less stable R208X-aLuc mRNA, we also measured the tRNA stability in the liver using tRNA-tailored microarrays (new panel, Fig. 2c and Extended Fig. 5b; see also comment #4). Together, the data implies – as also mentioned by the Referee – that the decrease of the rescue effect at 24h is confounded by the much lower mRNA stability. We have edited the text accordingly.

4) Why was a Cre transcript with two UGA PTCs used in experiments outlined in Fig. 2? Was this because there is too much spurious readthrough of one PTC in Cre that resulted in high background activity? This implies an exceptionally "easy" PTC containing transcript to target. This issue paired with a Cre mediated event that is permanent, integrates sup-tRNA suppression activity over time. With a disease causing PTC, the endogenous transcripts and proteins are subject to degradation over time, and therefore the activity of the delivered sup-tRNA needs to be durable. Do the authors suggest daily inhalation of sup-tRNAs to maintain a steady state level of PTC readthrough?

When designing the Cre transcript, we were led by the idea of creating a stringently terminating construct to minimize any false positives. The use of termination codon tandems has been proposed (<https://www.ncbi.nlm.nih.gov/pmc/articles/PMC521343/>) as an accurate way to study readthrough and avoid any context-dependent effects based on the used reporter. Furthermore, as mentioned in the manuscript, tdTomato alone exhibits a low Cre-independent background expression for yet unknown reasons (also stated by the manufacturer). Thus, to have a stringent reporter, we used a reporter with two PTCs. We did not test any constructs with single codons.

In the revised manuscript, we included tRNA stability data in vivo (Fig. 2c), which along with the in-cell stability (Extended Data Fig. 5c) illustrates the high stability of administered tRNA, which largely exceeds the stability of mRNA. The high stability of the suppressor tRNA supports a fairly low frequency of administration, although true dosing frequency and schedule can be only defined in a clinical trial setting (please see: <https://www.nature.com/articles/s41578-021-00358-0>). Furthermore, the dosing would be disease-specific, i.e. depending also on the stability of the disease protein in the corresponding tissue. Thus, tRNA administration may range from once every two weeks for some diseases with faster turnover of the underlying protein, or once a month or even longer for others with slower protein turnover. A phase I/II clinical trial with mRNA-replacement therapy launched by Translate Bio shows maximal effects on the lung function on day 8, though the in-lung stability of the administered mRNA is approx 24h (<https://www.nature.com/articles/s41578-021-00358-0>). Thus, considering the much higher in-tissue stability of tRNAs, in CF therapeutic settings the administration might be every 4-6 weeks.

5) The addition of results in Fig. 4 significantly adds to the study. These are very impressive Western blots. However, the amount of CFTR mRNA expression in panel b, CFTR protein expression in panel c and d, short circuit current in panel e and ASL height in panel f should all be compared to WT levels. Comparison to WT levels of all of these readouts is what is relevant for a proposed therapeutic, not fold-change over control.

We followed the suggestion of the Referee and in the revised manuscript, we induced the requested comparison to wildtype CFTR (Fig. 3 and 4) or wildtype control (Fig. 1 and 2), and where suitable included additional y-axes representing the signal as a percentage of the wildtype signal. We also discuss the numbers in the text.

Please note that in the microOCT measurements (Fig. 4e) we measured the ASL heights of a non-CF, healthy individual expressing wildtype CFTR, but do not present them as percentage of WT for the following reasons: (1) The data are time-based recordings, which prevents setting a value at maximum. (2) Our co-authors, Drs. Birket, Tearney and Rowe, who are amongst the developers of this approach, and within drug programs of CF foundation have established it as a reliable tool for predicting therapeutic efficacy, suggest to not directly calculate any percentage. The ASL is a complex tissue parameter, which does not have a zero value for a damaged organ (i.e. see the ASL with mismatch tRNA) neither a maximal value for a non-diseased individual (i.e. values among human individuals largely vary), which would be a prerequisite to calculate percentage, and this basal mucosal clearance varies between patients. As shown in Fig. 4c and 4d, and also discussed in the text, the CFTR protein expression and consequently the activity vary over many folds. For these two parameters, since nonsense causes no protein production (i.e. well defined zero value), the data can be presented as a percentage of wildtype (Fig. 4c and d). Similar to the ASL heights data, since nonsense mutation does not completely abolish transcript production, the mRNA expression data are not calculated as percentage of wildtype; the mRNA expression of wildtype CFTR is included for comparison (Fig. 4a).

6) Why was lipofectamine 3000, rather than the lipid-nanoparticle, used to transfect 16HBEge- and hNE cells in Fig. 4? A high delivery efficiency with the repurposed lipid-nanoparticulated tRNAs to primary human cells would significantly add to the findings of this paper. Further, it is not understood how hNE and 16HBEge- cells were transfected in Transwells. It is well known that these cell types are resistant to

transfection, especially once they have formed tight junctions and have polarized. Therefore, it would be reasonable to conclude that low transfection efficiency greatly influenced the results.

Lipofectamine is a routinely used transfection reagent and at pre-clinical stage a comparison with lipofectamine is endorsed in in vitro studies (e.g. cell culture and primary cells) for drug label expansion. As commented by the Referee, the transfection of primary differentiated cells is not as trivial as transfection of immortalized cells lines, yet it is well documented in the literature and many of the co-authors of the paper (Drs. Sorscher, Hong and our laboratory) routinely use such models and mode of transfection. We have included details on the transfection protocol (see the dedicated subsection “Transfection with tRNA and treatment with NMD inhibitor” in the Methods section), in which we acknowledge the relatively low lipofectamine transfection efficiency in primary cells of appr. 20%. We also mention in the text, that this low transfection efficiency is likely underestimating the effect of tRNA^{Sup} and in vivo, in patients, much higher efficacy in restoring CFTR function might be achieved.

The Referee suggests treating primary hNEs with LUNAR-LNPs to directly compare to our in vivo systemic administration. We considered indeed this suggestion and performed the experiment, however, administering amounts comparable with the in vivo experiments we detected no full length CFTR. By increasing the concentration of the formulation by two to three orders of magnitude over those used in vivo, we detected low amount of full length CFTR with no obvious concentration dependence of its expression. At such high concentrations, the LNPs adhered at the cell membrane and altered cell resistance and shape, thus preventing ion channel activity measurement. The transfection efficiency with LUNARs was also very poor reaching only 5-7%, which corroborates published data (see below). Because of the high concentrations, that are incompatible with a drug pipeline, this experiment suggests to us, that LNPs cannot be applied on isolated cells.

While experiencing these experimental difficulties, we performed a thorough literature search and were surprised to find many examples of incompatibility of LNPs for in vitro use (with immortalized or primary cells), corroborating our observations. Systematic analysis of more than 85 LNP types, shows an extremely poor in vitro delivery contrasting a highly efficient delivery in animals (PMID: 29489381). Comparison of lipofectamine transfection with LNP-mediated transfection in 14 immortalized cell types shows a poor LNP-based transfection efficiency in cells (PMID: 30788952). For example, in 9 out of 14 cell lines, the LNP transfection did not work; even in the most robust laboratory HEK293 cell line the efficiency with LNPs was 20% compared to 95% with lipofectamine. Likely reasons for this are: (i) vital interactions of serum proteins with LNP surface – a process termed ‘passive targeting’ or ‘endogenous targeting’ which is absent in cell culture (PMID: 23334168, PMID: 35173043), and (ii) affected endocytosis by gene expression alterations that occur when cells are removed from their natural tissue microenvironment (in particularly relevant to primary cells) (PMID: 30788952). Many published studies – including successful pre-clinical precursors of current clinical trials – have been performed following the scheme: in in vitro experiments (i.e. with immortalized cell lines and primary cells) using commercial, optimized transfection approaches or electroporation, while in vivo work (e.g. in mouse) is performed using LNPs (to mention some examples: PMID: 35356682; PMID: 35624092; PMID: 35356683; PMID: 35755289; PMID: 35688311; PMID: 34880218...). Importantly, this experimental setup is not limited to nanoparticulated RNA-based therapeutics. Also, AAV-based delivery systems do not efficiently work in cell culture system or primary patient-derived cells. Thus, AAV-systems are used mainly for in vivo studies, while in vitro work is done with classic lentiviral vectors, as also exemplified in a recent example of AAV-based tRNA therapeutics (PMID: 35322228).

In summary, we believe that using a common and established transfection reagent (here lipofectamine) allows for disjoining the effect of tRNA^{sup} entity from that of the LNPs. In addition, in primary cell lines a comparable transfection to immortalized cell lines is required by NDA as a part of pre-clinical data.

7) This statement “Finally, we addressed the efficacy of the nonsense suppressor tRNAs to restore expression of the full-length CFTR protein harboring different pathogenic mutations in cell models and patient-derived primary epithelial cells. PTC-tailored suppressor tRNAs were specific to the PTC identity and context (Extended Data Fig. 10a) and rescued full-length S466X-CFTR (X=UGA or UAA) or R553X-CFTR (X=UGA) transiently transfected in CFBE41o- cells (Fig. 3a and Extended Data Fig. 10b).”, is very misleading. The way this sentence is worded leads one to believe that a PTC was suppressed in a CFTR transcript that was expressed from the endogenous gene, i.e. a transcript that experiences post-transcriptional processing. Here, the CFTR mRNA with CF-causing PTCs was introduced into cells using lipofectamine 3000, which notably is an mRNA that is not targeted for NMD. Moreover, it is unclear how this is different or more impactful than transfecting these components into HEK293 cells.

This sentence is edited and displaced to the section that discusses only 16HBEge and primary patient-derived hNE cells, both of which expressing endogenous full-length R1162X-CFTR with the complete set of introns and exons. By separating the cell systems (Fig. 3 expressing CFTR from cDNA (no introns) and Fig 4 endogenous mRNA expression undergoing mRNA processing) along with the extensive text editing to explain each experiment, we believe that we reached a clarity of the systems and which purpose it serves (please see also the response to comment #7).

8) Overall, in Fig. 3 and 4, there are different suptRNAs used in different cell types and with different CFTR mutations in different CFTR transcript types. Therefore, it is difficult to really compare all of the results. It will be hard to follow what seems to be patch-work results for many readers. For comparison of a CF PTC expressed from a transgene with no post-transcriptional regulation and an endogenous CFTR transcript, where post-transcriptional modification occurs, R1162X in 16HBEge- and FRT cells can be studied.

Reflecting on the Referee’s comment, we substantially edited the text discussing the results of Fig. 3 and Fig. 4 to clearly state the purpose of each analysis and emphasize on the strategic resemblance with CF-related drug-screening pipelines. Namely, those include:

- (1) Testing several mutations, including such of the same type (e.g. here R553X and R1162X) for context-dependent effects in model systems with a simple quantifiable readout from intron-less DNA (i.e. CFBE41o- for full-length protein expression and FRT for activity). Both systems are standard cellular models viewed by the US Food and Drug Administration as informative for drug label expansion of CFTR modulator compounds.
- (2) Dependent on the available genotypes, testing selected compounds in primary patient-derived cells to establish the effect in the naturally regulated gene (i.e. endogenous expression with full set of introns and exons), including analysis predictive to clinical outcomes. Thereby, mutations tested with a patient material (e.g. here R1162X) must be tested in cell culture settings as well, hence considering the R1162X mutation in CFBE41o- and FRT cells.

In the revision of the manuscript, we have adhered to this strategical resemblance with CF-related drug-screening pipelines and by presenting the data in this view, we hope it is clear why more mutations were tested in CFBE410- and FRT cells. We have included the requested activity measurements with R1162X FRT cells.

9) The panels in Fig. 4 should be labeled as 16HBEge- or hNE cells so it is clear to the reader which data came from what.

In the revision, we reorganized the figures, so that Fig. 3 summarizes all experiments in cell culture models expressing intron-less CFTR variants (i.e. from cDNA for testing only effects on protein and activity), while Fig. 4 contains data on systems endogenously expressing the natural CFTR open-reading frames (please see also our response to the above comment #8).

10) The right y-axis in Fig. 3 panel c should indicate percent WT rescue and the difference in CFTR functional rescue between tR and tRT5 should be highlighted.

This is included and as mentioned in the response to comment #5. We present all data as a comparison to wildtype or corresponding control.

11) When the two different NMD inhibitors (NMD14 and SMG1) are used, it should be made clearer in the manuscript which one is used and when. Also, why are two different NMD inhibitors used in this study?

We have designated the inhibitors and their concentration in the figure legends. We also have included an explanation in the Results and Methods sections on the reasoning for using two inhibitors.

12) It is stated that short circuit currents shown in Figure 4 are area under the curve. However, this cannot be possible as it appears both F&I and Inh172 responses are plotted.

We apologize for any confusion regarding this point. The values ($\mu\text{A}/\text{cm}^2$) are correct and were obtained using a standard software setting provided with the equipment supplier (Physiologic Instruments); we used a window to record changes over 60s intervals. This is now clarified in the revised Methods section.

Reviewer Reports on the Second Revision:

Referee #1 (Remarks to the Author):

The authors of the manuscript did an excellent job addressing the concerns of the previous review. However, the new results raise additional questions and also seem to have made the overall reading of the manuscript difficult. Furthermore, there are some troubling statements and choice of words as noted below.

Issues are not numbered by importance, but rather by the order in which they arise in the manuscript.

- 1) The title of the manuscript is not only misleading but also incorrect. This study does not “repurpose tRNAs”. Rather, it uses engineered suppressor tRNAs to suppress PTCs. Repurposing traditionally in this context would mean that the natural tRNAs, as they stand, are used to suppress nonsense mutations. Further, the wording is confusing. The tRNAs are delivered with lipid nanoparticles and do not act as lipid nanoparticles in vivo.

- 2) Lines 55-59: The authors state that “the impediment of using anti-codon repurposed native tRNAs as sup-tRNAs drugs is their relatively low suppression efficacy....” This has recently been shown to be untrue (PMID: 35664697; PMID: 35322228). In PMID: 35664697, 70-90% of CFTR channel function was rescued in 16HBEge-cells following cDNA sup-tRNA delivery. In PMID: 35322228, 20-30% of Idua protein expression was rescued with sup-tRNAs encoded by cDNA and virus. There are many more examples, but these are two recent publications.

- 3) Results in Figure 1C and D. It is unclear why two cell types are used here for two different PTCs that are encoded in the exogenous transgene FLuc. Also, “mis” is not defined in the figure legend.

- 4) Lines 91-94: It is claimed that the exchange of nucleotides in the anticodon of tRNA-Ser, tRNA-Arg, and tRNA-Gly has generated first generation sup-tRNAs. However, all of these sup-tRNAs were previously generated in PMID: 30778053 and tested for their suppression efficiency. Further, PMID: 10757796 used sup-tRNA-Ser in vivo, and sup-tRNA-Ser was defined in yeast in PMID: 762155. Notably, Mario Capecchi used sup-tRNA-Ser in 1965 (PMID: 17809404).

- 5) Lines 103-104: Stating that sup-tRNA-Arg has the highest suppression efficiency because of the parent tRNA having a high amount of miscoding is unfounded. Miscoding and suppression efficiency are not thought to be linked mechanistically. This same explanation was used in Line 136 for sup-tRNA-Gly. A more likely explanation is that the interaction of the tRNA body and anticodon with the aminoacyl-tRNA synthetase results in less efficient acylation. Miscoding would mean that the fidelity of suppression is influenced. It has been shown in PMID: 35322228 and in PMID: 30778053 that the fidelity of most sup-tRNAs is quite high, as has been shown in results from these authors.

- 6) Line 112: The authors need to be careful with the statement “efficiency in decoding” as this imparts possibly unintended meaning about decoding fidelity. PTC suppression efficiency is commonly used.

7) Line 130: The authors need to take better care in using the term “readthrough” when referring to sup-tRNAs. Readthrough is usually reserved for drugs that interact with the translation and or termination machinery (e.g. aminoglycosides).

8) The authors use “PTC rescue” throughout the manuscript (line 132) and on many of the y-axes. This is not correct, as the PTC is not corrected/rescued. Rather, it is suppressed, which allows for full-length protein to be made. Examples of technologies that correct PTCs are pseudouridine modification of mRNA and base-editing of genomic DNA.

9) Extended Data Figure 5b and c: At what timepoint is the quantitation in panel b made? The measurements recorded in panel c are confusing. The average amount of tRNA appears to go up at 24 and 36hrs from the 5hr-timepoint, down at 48hrs and then back up at 72hrs. Since it was tRNA that was delivered to cells, therefore no way for the cell to generate more of this tRNA, begging the question: how is this possible? This issue is again seen in Fig 2c, where more tRNA is present at 72hrs relative to 6hrs. Further, how is this finding reconciled with recent published findings in PMID: 35664697 that the half-life of sup-tRNA in cells is ~6hrs?

10) Figure 3 a. The rescue efficiency of CFTR protein in CFBE41o cells was determined by western blotting (WB). Why was CFTR functional rescue then measured in FRT cells? CFTR function deriving from stably introduced cDNA is altogether different than when encoded from the endogenous transcripts that undergo post-transcriptional processing. These results are disjointed and the panels should be labeled with cell-types used to remove confusion by the reader. Further, how much CFTR protein derived from introduced cDNA in the FRT-cell experiments, as determined by WB? Does this parallel the functional rescue?

11) Figure 4a: The ability of sup-tRNAs to stabilize PTC-containing mRNAs through inhibition of NMD has been previously shown and therefore should be referenced. PMID: 35664697 demonstrated this to a much higher degree than reported here for CFTR-R1162X in 16HBEge cells, and PMID: 35322228 determined this in vivo for iuda transcripts using sup-tRNA-Tyr rAAV. Given that tR5+NMD inhibitor demonstrated an additive effect, it is unclear why tR+NMD inhibitor resulted in less steady-state cftr mRNA expression when compared to tR alone (and the level of expression is also below that obtained using NMD inhibitor alone). A mismatched control needs to be included in these ddPCR experiments to compare the degree of rescue.

12) Figure 4c and d: the right y-axes are confusing, and WT CFTR expression (Fig. 4c) and function (Fig. 4d) should be separated from the sup-tRNA result panels.

13) Line 340-341 and Figure 4 title. Is it appropriate to say that sup-tRNAs outcompete mRNA surveillance mechanisms or NMD decay? This reviewer believes that “outcompete” should be replaced by “inhibit”, as mechanistically it is more accurate.

14) Line 381-382: The findings that sup-tRNAs alone can inhibit NMD in cftr transcripts harboring a PTC was previously described in PMID: 35322228 and commented on in PMID: 36035751. It was also demonstrated here PMID: 35322228.

Minor:

- 1) Standard nomenclature for suppressor tRNAs should be followed, i.e. "sup-tRNA". It is a concern that using the terminology "anti-codon repurposed tRNAs^{Sup}" will add confusion to readers and the field.
- 2) Line 56: Calling sup-tRNAs "drugs" will be viewed by many as inappropriate. Sup-tRNAs are more akin to a gene therapy approach. They are not a small molecule.

Referee #2 (Remarks to the Author):

NA

Referee #3 (Remarks to the Author):

The authors have meaningfully responded to all of the concerns/suggestions raised in the additional round of review, and the manuscript has been further strengthened as a result. I remain supportive of its publication in Nature. Error bars are included and the statistical analyses employed are reasonable. To increase clarity, I recommend adding specific labels to the top of each dataset in Figs 1-4 to indicate which type of biological sample was used for each experiment, e.g., "CF primary human epithelia" etc.

Author Rebuttals to Second Revision:

Point-by-point responses to the Referee #1 comments

Referee #1

The authors of the manuscript did an excellent job addressing the concerns of the previous review. However, the new results raise additional questions and also seem to have made the overall reading of the manuscript difficult. Furthermore, there are some troubling statements and choice of words as noted below.

We are very pleased about this overall positive assessment of the Referee.

The outstanding comments, which seem more of textual editing and terminology clarification, we have addressed in the following way:

Issues are not numbered by importance, but rather by the order in which they arise in the manuscript. 1) The title of the manuscript is not only misleading but also incorrect. This study does not “repurpose tRNAs”. Rather, it uses engineered suppressor tRNAs to suppress PTCs. Repurposing traditionally in this context would mean that the natural tRNAs, as they stand, are used to suppress nonsense mutations. Further, the wording is confusing. The tRNAs are delivered with lipid nanoparticles and do not act as lipid nanoparticles in vivo.

The comment it on the semantics of the word ,repurpose’. The Referee suggests ‘engineered’ instead, which we are glad to consider. In the former title we use nanoparticulations, which implies LNP-sup-tRNA formulations - a term, which might be specific for the field. Thus, we shortened the title to: Engineered tRNAs suppress nonsense mutations in cells and in vivo

2) Lines 55-59: The authors state that “the impediment of using anti-codon repurposed native tRNAs as sup-tRNAs drugs is their relatively low suppression efficacy....” This has recently been shown to be untrue (PMID: 35664697; PMID: 35322228). In PMID: 35664697, 70-90% of CFTR channel function was rescued in 16HBEge-cells following cDNA sup-tRNA delivery. In PMID: 35322228, 20-30% of Idua protein expression was rescued with sup-tRNAs encoded by cDNA and virus. There are many more examples, but these are two recent publications.

We have the feeling that the sentence has been misunderstood by the Referee. It was our intention to emphasize on the lack of clinical trials, although the concept of using suppressor tRNAs has been proposed back in 1979 by Kan and co-authors (PMID 492326 – ref. 3 in the reference list), with preclinical tests in NDA-approved systems to show clinical benefit. Thereby, we meant the general applicability of the concept of Kan and co-workers in the context of any mutation-based disease. To clarify this point, we have edited this part of the sentence (l. 54-56) to:

“An impediment is that not every native tRNA can be engineered into sup-tRNA by altering its anticodon...”

Both articles are cited at various places in the paper (ref. 7 and 35).

3) Results in Figure 1C and D. It is unclear why two cell types are used here for two different PTCs that are encoded in the exogenous transgene FLuc. Also, “mis” is not defined in the figure legend.

To establish the broad applicability of our sup-tRNAs, as early as at the screening level, we considered different disease-related mutations, using thereby the recommended cell system for each mutation; the cell system most closely match the natural cell/tissue of onset of the corresponding disease in humans. We first used R208X – a common mutation in tripeptidyl peptidase 1 gene associated with lysosomal storage disorder using liver-derived laboratory cell line. For one sup-tRNA, to test its efficiency for another disease-mutation context, we considered S466X in CFTR. Since this mutation is implicated in CF disease with the strongest phenotype in bronchial tissues, we considered human bronchial epithelial CFBE41o⁻ cell line (please see l. 120-128). Indeed, the efficacy of tSA1T5 at the UGA PTC was greater for the S466X than for R208X. We believe this is an important result, implying that the PTC sequence context also modulates sup-tRNA efficacy – an effect that has been reported for aminoglycosides-stimulated readthrough at natural termination codons.

The term ‘mis’ was defined only in panel c; we have also added it to the legend of Fig. 1d.

4) Lines 91-94: It is claimed that the exchange of nucleotides in the anticodon of tRNA-Ser, tRNA-Arg, and tRNA-Gly has generated first generation sup-tRNAs. However, all of these sup-tRNAs were previously generated in PMID: 30778053 and tested for their suppression efficiency. Further, PMID: 10757796 used sup-tRNA-Ser in vivo, and sup-tRNA-Ser was defined in yeast in PMID: 762155. Notably, Mario Capecchi used sup-tRNA-Ser in 1965 (PMID: 17809404).

We do not claim that we are the first to use natural tRNA^{Ser}, tRNA^{Gly}, tRNA^{Arg} with altered anticodon to decode PTC. In the introduction, we acknowledge that the idea of using such sup-tRNAs was first proposed in the late 70ies. In a similarity to the terminology used in drug screening pipelines (i.e. generation 1, generation 2 etc.), we use the term ‘generation’ to designate the successive changes we made into the same tRNA body. Reflecting on the Referee’s comment, we realize that this might be misunderstood in a sense of invention, thus, we deleted ‘generation’ and use the sup-tRNA abbreviations instead.

5) Lines 103-104: Stating that sup-tRNA-Arg has the highest suppression efficiency because of the parent tRNA having a high amount of miscoding is unfounded. Miscoding and suppression efficiency are not thought to be linked mechanistically. This same explanation was used in Line 136 for sup-tRNA-Gly. A more likely explanation is that the interaction of the tRNA body and anticodon with the aminoacyl-tRNA synthetase results in less efficient acylation. Miscoding would mean that the fidelity of suppression is influenced. It has been shown in PMID: 35322228 and in PMID: 30778053 that the fidelity of most sup-tRNAs is quite high, as has been shown in results from these authors.

We use the term ‘miscoding’ (l. 85, formerly l. 103) and refer to a mechanism described in ref. 23 for the natural tRNA^{Arg}UCU, and not to the properties of our sup-tRNAs. We write: “..likely because the natural tRNA^{Arg}UCU – the precursor of tR – is intrinsically prone to miscoding²².” Since ‘miscoding’ is the molecular explanation provided in ref. 22, we would like to adhere to the terminology used in the primary publication. We use the term ‘hyper accuracy’ for the natural sense-codon decoding tRNA^{Gly} and not for sup-tRNA. Again by using this term we correctly refer to the published primary literature (ref 22), which describes the properties of the natural sense-codon decoding tRNA^{Gly} as a hyper accurate tRNA.

6) Line 112: The authors need to be careful with the statement “efficiency in decoding” as this imparts possibly unintended meaning about decoding fidelity. PTC suppression efficiency is commonly used.

We have carefully inspected all places, where we use decoding, to only use it for natural tRNAs and not in a context of sup-tRNA and PTC suppression.

7) Line 130: The authors need to take better care in using the term “readthrough” when referring to sup-tRNAs. Readthrough is usually reserved for drugs that interact with the translation and or termination machinery (e.g. aminoglycosides).

Here, we respectfully disagree with the Referee. Suppression is a much broader term and refers to the biological effect of sup-tRNAs, while readthrough describes the molecular action at the codon (here PTC). Readthrough is an established molecular term to designate the stochastic spontaneous readthrough of native stop codons (i.e. natural tRNA-mediated readthrough at native stop codons, which in human cells may reach up to 1% for some native stop codons). Because of mechanistic similarities, it is also widely used for sup-tRNAs (PMID: 35322228; PMID: 23812587; PMID:36631608; PMID: 24302569). We have edited the manuscript and adhere to two terms, readthrough when referring to the molecular mechanism and suppression when describing the process. This is accordance with the recently published article describing rAAV-administered sup-tRNAs (PMID: 35322228).

8) The authors use “PTC rescue” throughout the manuscript (line 132) and on many of the y-axes. This is not correct, as the PTC is not corrected/rescued. Rather, it is suppressed, which allows for full-length protein to be made. Examples of technologies that correct PTCs are pseudouridine modification of mRNA and base-editing of genomic DNA.

We used ‘rescue’ as a widely used term for phenotypic correction of disease-related proteins, in particularly in the CF field, which indeed might be a field-specific term. Thus, we have edited the term ‘rescue’ to more specific terms, i.e. readthrough, suppress, restore, augment (see the above response to comment #8). In accordance with the recently published article describing rAAV-administered sup-tRNAs (PMID: 35322228), we also edited the axes of the plots to ‘Readthrough efficiency’.

9) Extended Data Figure 5b and c: At what timepoint is the quantitation in panel b made? The measurements recorded in panel c are confusing. The average amount of tRNA appears to go up at 24 and 36hrs from the 5hr-timepoint, down at 48hrs and then back up at 72hrs. Since it was tRNA that was delivered to cells, therefore no way for the cell to generate more of this tRNA, begging the question: how is this possible? This issue is again seen in Fig 2c, where more tRNA is present at 72hrs relative to 6hrs. Further, how is this finding reconciled with recent published findings in PMID: 35664697 that the half-life of sup-tRNA in cells is ~6hrs?

Panel b in Extended Data Figure 5b is a representative single block from one single microarray (each containing 12 such blocks) to illustrate the high sensitivity to the target sup-tRNA without any cross hybridization to the parental tRNA^{Ser}. All arrays are deposited in GEO.

The tRNA-tailored arrays are based on fluorescence and single hyperfluorescent spots are commonly observed in arrays (the spot at position B5 on the exemplified microarray block is an example for such hyperfluorescent spot, Extended Data Figure 5b). Thus, commonly probes of the same kind are

present in multiple copies (here 36 identical signals for each sup-tRNA, see Methods section). The signals are presented as box plots containing all data points (without omitting any points with an outlier test) to visualize the spread of the signals and the 90%ile that majorly influence the mean. Despite the oscillation of the mean and visually being over the mean of the 5h data point which is set at 100%, these fluctuations are statistically insignificant and thus, cannot be interpreted as an increase, but rather as unchanged values. The spread of the signal we receive is very typical for microarray (please see

We cannot comment on the findings in PMID: 35664697 and the much shorter half life of the tRNA in the mentioned study, since this is a result obtained with different tRNA(s), under different experimental conditions (e.g. transfection protocol) and with a tRNA detection method with different sensitivity.

10) Figure 3 a. The rescue efficiency of CFTR protein in CFBE41o cells was determined by western blotting (WB). Why was CFTR functional rescue then measured in FRT cells? CFTR function deriving from stably introduced cDNA is altogether different than when encoded from the endogenous transcripts that undergo post-transcriptional processing. These results are disjointed and the panels should be labeled with cell-types used to remove confusion by the reader. Further, how much CFTR protein derived from introduced cDNA in the FRT-cell experiments, as determined by WB? Does this parallel the functional rescue?

Figure 3 uses cellular systems (FRT and CFBE41o- cells) expressing CFTR cDNA, which does not undergo post-transcriptional processing. As we state in the text (subsection 'sup-tRNA efficacy on full-length protein') these models were utilized to establish the efficacy of the tS and tR variants in restoring translation and expression of a full-length disease protein (i.e. with intron-less cDNA). Systems with endogenously encoded PTC-CFTR transcripts that undergo processing are summarized in Figure 4. Following the recommendation of the Referee, the panels have now been labeled to identify cell types being tested.

In the CF field, CFBE41o cells are commonly used for expression studies based on measurable levels of CFTR protein, whereas FRT cells are employed for parallel activity assays. In particular, short circuit current findings in FRTs comprise a standard cellular model viewed by the US Food and Drug Administration as informative for drug label expansion of CFTR modulator compounds. Functional thresholds of this type are not available for CFBE cells. Because CFTR undergoes complex maturational processing, there is no direct correlation between C-band protein expression and activity in FRTs. Functional CFTR rescue has generally been difficult to correlate with CFTR protein expression levels in any cell-based system, hence the FDA-recommendation for the different cell line for biochemical and functional assay. In sum, as we explained in the previous reviewing round, we use multiple cell systems (also FDA-utilized) for testing and approving new CF therapies (e.g., FRT for activity, gene-edited bronchial cells (16HBE) and CFBE41o- for biochemical or expression analysis).

11) Figure 4a: The ability of sup-tRNAs to stabilize PTC-containing mRNAs through inhibition of NMD has been previously shown and therefore should be referenced. PMID: 35664697 demonstrated this to a much higher degree than reported here for CFTR-R1162X in 16HBEge cells, and PMID: 35322228 determined this in vivo for iuda transcripts using sup-tRNA-Tyr rAAV. Given that tR5+NMD inhibitor demonstrated an additive effect, it is unclear why tR+NMD inhibitor resulted in less steady-state cftr mRNA expression when compared to tR alone (and the level of expression is also below that obtained

using NMD inhibitor alone). A mismatched control needs to be included in these ddPCR experiments to compare the degree of rescue.

Both references have been cited in the NMD section (l. 237-239).

The reason for the decrease of CFTR mRNA by combined treatment with tR and NMD inhibitor is unknown, but could be due to the general adverse effects on NMD inhibitors. As we discuss in the text, we noted NMD inhibitor-driven effects on cell viability.

The mismatch tRNA control has been included in Figure 4a, which as expected did not stabilize CFTR mRNA, suggesting that the effect is specific to sup-tRNA.

12) Figure 4c and d: the right y-axes are confusing, and WT CFTR expression (Fig. 4c) and function (Fig. 4d) should be separated from the sup-tRNA result panels.

Figure 4c and 4d have double y-axes on the right. We made a wider separation of the wild type plot (i.e. a cut along the x-axis), and hope that the two consecutive plots are well distinguishable now.

13) Line 340-341 and Figure 4 title. Is it appropriate to say that sup-tRNAs outcompete mRNA surveillance mechanisms or NMD decay? This reviewer believes that “outcompete” should be replaced by “inhibit”, as mechanistically it is more accurate.

The term ‘inhibition’ would refer to a direct effect or a direct binding of sup-tRNA on NMD machinery. The sup-tRNA interacts solely with the elongation factor and the ribosome, and by increasing the utilization of PTC-mRNA in translation indirectly decreases the NMD decay. We believe that ‘outcompete’ is still the correct term. Also, other publications (PMID: 35322228 (ref. 7 and 36 in our reference list, respectively) use term as ‘antagonize’, which similarly to outcompete emphasize the indirect effect of suppressor tRNAs. Thus, we use both words to emphasize the indirect effect of sup-tRNA on NMD and use the word inhibition in the context of NMD inhibitors which directly inhibit NMD machinery components.

14) Line 381-382: The findings that sup-tRNAs alone can inhibit NMD in cfr transcripts harboring a PTC was previously described in PMID: 35322228 and commented on in PMID: 36035751. It was also demonstrated here PMID: 35322228.

We have difficulties understanding this comment. The cited text (l. 296-297, formerly l. 381-382) summarizes our results and not a published work. To emphasize this we added “We show that” so that it reads “We show that at CF disease-causing PTCs, an optimized tRNA suppressor alone restored protein expression, function, and airway volume homeostasis in a manner relevant to CF clinical benefit – thus, in this case addition of NMD inhibitors as adjuvants may not be necessary.”

In this context, we refer to the results from publication PMID: 35322228, when summarizing published work reporting that efficient readthrough with chemical compounds or sup-tRNAs antagonize NMD on PTC-containing mRNAs. (l. 237-239).

Minor:

1) Standard nomenclature for suppressor tRNAs should be followed, i.e. “sup-tRNA”. It is a concern that using the terminology “anti-codon repurposed tRNAsup” will add confusion to readers and the field.

By previously naming the suppressor tRNAs (tRNA^{sup}) we adhered to the standard nomenclature of tRNAs, namely adding the functionality for the encoded peptide as a superscript. Because for suppressor tRNAs there is no established nomenclature, we were glad to edit it to sup-tRNA which has recently been used in PMID: 35322228.

We changed 'anticodon-repurposed' to 'anticodon-altered'.

2) Line 56: Calling sup-tRNAs "drugs" will be viewed by many as inappropriate. Sup-tRNAs are more akin to a gene therapy approach. They are not a small molecule.

We edited this place in the text and do not use 'drug' in the context of sup-tRNAs; we use it only for low-molecular weight pharmacological compounds.

Referee #2:

NA

Referee #3:

The authors have meaningfully responded to all of the concerns/suggestions raised in the additional round of review, and the manuscript has been further strengthened as a result. I remain supportive of its publication in Nature. Error bars are included and the statistical analyses employed are reasonable. To increase clarity, I recommend adding specific labels to the top of each dataset in Figs 1-4 to indicate which type of biological sample was used for each experiment, e.g., "CF primary human epithelia" etc.

We were very please to see the recommendation of this Referee.

Following their recommendation too (also in the comment #10 of Referee #1), the panels have now been labeled to identify cell types being tested.

Reviewer Reports on the Third Revision:

Referee #1 (Remarks to the Author):

The authors have meaningfully responded to the concerns and suggestions raised in the additional round of review, and the manuscript has been significantly strengthened as a result.

We are pleased to read the very supportive comment of Referee #1, who feels that we have addressed all their concerns.